# Unifying PAC and Regret: Uniform PAC Bounds for Episodic Reinforcement Learning

**Christoph Dann**
Machine Learning Department
Carnegie-Mellon University
cdann@cdann.net

**Tor Lattimore**[*]
tor.lattimore@gmail.com

**Emma Brunskill**
Computer Science Department
Stanford University
ebrun@cs.stanford.edu

## Abstract

Statistical performance bounds for reinforcement learning (RL) algorithms can be critical for high-stakes applications like healthcare. This paper introduces a new framework for theoretically measuring the performance of such algorithms called *Uniform-PAC*, which is a strengthening of the classical Probably Approximately Correct (PAC) framework. In contrast to the PAC framework, the uniform version may be used to derive high probability regret guarantees and so forms a bridge between the two setups that has been missing in the literature. We demonstrate the benefits of the new framework for finite-state episodic MDPs with a new algorithm that is Uniform-PAC and simultaneously achieves optimal regret *and* PAC guarantees except for a factor of the horizon.

## 1 Introduction

The recent empirical successes of deep reinforcement learning (RL) are tremendously exciting, but the performance of these approaches still varies significantly across domains, each of which requires the user to solve a new tuning problem [1]. Ultimately we would like reinforcement learning algorithms that simultaneously perform well empirically and have strong theoretical guarantees. Such algorithms are especially important for high stakes domains like health care, education and customer service, where non-expert users demand excellent outcomes.

We propose a new framework for measuring the performance of reinforcement learning algorithms called Uniform-PAC. Briefly, an algorithm is Uniform-PAC if with high probability it simultaneously for all $\varepsilon > 0$ selects an $\varepsilon$-optimal policy on all episodes except for a number that scales polynomially with $1/\varepsilon$. Algorithms that are Uniform-PAC converge to an optimal policy with high probability and immediately yield both PAC and high probability regret bounds, which makes them superior to algorithms that come with only PAC or regret guarantees. Indeed,

(a) Neither PAC nor regret guarantees imply convergence to optimal policies with high probability;

(b) $(\varepsilon, \delta)$-PAC algorithms may be $\varepsilon/2$-suboptimal in every episode;

(c) Algorithms with small regret may be maximally suboptimal infinitely often.

---

[*]Tor Lattimore is now at DeepMind, London

Uniform-PAC algorithms suffer none of these drawbacks. One could hope that existing algorithms with PAC or regret guarantees might be Uniform-PAC already, with only the analysis missing. Unfortunately this is not the case and modification is required to adapt these approaches to satisfy the new performance metric. The key insight for obtaining Uniform-PAC guarantees is to leverage time-uniform concentration bounds such as the finite-time versions of the law of iterated logarithm, which obviates the need for horizon-dependent confidence levels.

We provide a new optimistic algorithm for episodic RL called UBEV that is Uniform PAC. Unlike its predecessors, UBEV uses confidence intervals based on the law of iterated logarithm (LIL) which hold uniformly over time. They allow us to more tightly control the probability of failure events in which the algorithm behaves poorly. Our analysis is nearly optimal according to the traditional metrics, with a linear dependence on the state space for the PAC setting and square root dependence for the regret. Therefore UBEV is a Uniform PAC algorithm with PAC bounds and high probability regret bounds that are near optimal in the dependence on the length of the episodes (horizon) and optimal in the state and action spaces cardinality as well as the number of episodes. To our knowledge UBEV is the first algorithm with both near-optimal PAC and regret guarantees.

**Notation and setup.** We consider episodic fixed-horizon MDPs with time-dependent dynamics, which can be formalized as a tuple $M = (\mathcal{S}, \mathcal{A}, p_R, P, p_0, H)$. The statespace $\mathcal{S}$ and the actionspace $\mathcal{A}$ are finite sets with cardinality $S$ and $A$. The agent interacts with the MDP in episodes of $H$ time steps each. At the beginning of each time-step $t \in [H]$ the agent observes a state $s_t$ and chooses an action $a_t$ based on a policy $\pi$ that may depend on the within-episode time step ($a_t = \pi(s_t, t)$). The next state is sampled from the $t$th transition kernel $s_{t+1} \sim P(\cdot|s_t, a_t, t)$ and the initial state from $s_1 \sim p_0$. The agent then receives a reward drawn from a distribution $p_R(s_t, a_t, t)$ which can depend on $s_t, a_t$ and $t$ with mean $r(s_t, a_t, t)$ determined by the reward function. The reward distribution $p_R$ is supported on $[0, 1]$.[2] The value function from time step $t$ for policy $\pi$ is defined as

$$V_t^\pi(s) := \mathbb{E}\left[\sum_{i=t}^H r(s_i, a_i, i)\middle| s_t = s\right] = \sum_{s' \in \mathcal{S}} P(s'|s, \pi(s, t), t)V_{t+1}^\pi(s') + r(s, \pi(s, t), t).$$

and the optimal value function is denoted by $V_t^\star$. In any fixed episode, the quality of a policy $\pi$ is evaluated by the *total expected reward* or *return*

$$\rho^\pi := \mathbb{E}\left[\sum_{i=t}^H r(s_i, a_i, i)\middle| \pi\right] = p_0^\top V_1^\pi,$$

which is compared to the *optimal return* $\rho^\star = p_0^\top V_1^\star$. For this notation $p_0$ and the value functions $V_t^\star$, $V_1^\pi$ are interpreted as vectors of length $S$. If an algorithm follows policy $\pi_k$ in episode $k$, then the optimality gap in episode $k$ is $\Delta_k := \rho^\star - \rho^{\pi_k}$ which is bounded by $\Delta_{\max} = \max_\pi \rho^\star - \rho^\pi \leq H$. We let $N_\varepsilon := \sum_{k=1}^\infty \mathbb{I}\{\Delta_k > \varepsilon\}$ be the number of $\varepsilon$-errors and $R(T)$ be the regret after $T$ episodes: $R(T) := \sum_{k=1}^T \Delta_k$. Note that $T$ is the number of episodes and not total time steps (which is $HT$ after $T$ episodes) and $k$ is an episode index while $t$ usually denotes time indices within an episode. The $\tilde{O}$ notation is similar to the usual $O$-notation but suppresses additional polylog-factors, that is $g(x) = \tilde{O}(f(x))$ iff there is a polynomial $p$ such that $g(x) = O(f(x)p(\log(x)))$.

## 2 Uniform PAC and Existing Learning Frameworks

We briefly summarize the most common performance measures used in the literature.

- $(\varepsilon, \delta)$-*PAC:* There exists a polynomial function $F_{\text{PAC}}(S, A, H, 1/\varepsilon, \log(1/\delta))$ such that
$$\mathbb{P}\left(N_\varepsilon > F_{\text{PAC}}(S, A, H, 1/\varepsilon, \log(1/\delta))\right) \leq \delta.$$

- *Expected Regret:* There exists a function $F_{\text{ER}}(S, A, H, T)$ such that $\mathbb{E}[R(T)] \leq F_{\text{ER}}(S, A, H, T)$.

- *High Probability Regret:* There exists a function $F_{\text{HPR}}(S, A, H, T, \log(1/\delta))$ such that
$$\mathbb{P}\left(R(T) > F_{\text{HPR}}(S, A, H, T, \log(1/\delta))\right) \leq \delta.$$

- *Uniform High Probability Regret:* There exists a function $F_{\text{UHPR}}(S, A, H, T, \log(1/\delta))$ such that

$$\mathbb{P}\left(\text{exists } T : R(T) > F_{\text{UHPR}}(S, A, H, T, \log(1/\delta))\right) \leq \delta \,.$$

In all definitions the function $F$ should be polynomial in all arguments. For notational conciseness we often omit some of the parameters of $F$ where the context is clear. The different performance guarantees are widely used (e.g. PAC: [2, 3, 4, 5], (uniform) high-probability regret: [6, 7, 8]; expected regret: [9, 10, 11, 12]). Due to space constraints, we will not discuss Bayesian-style performance guarantees that only hold in expectation with respect to a distribution over problem instances. We will shortly discuss the limitations of the frameworks listed above, but first formally define the Uniform-PAC criteria

**Definition 1** (Uniform-PAC). *An algorithm is Uniform-PAC for $\delta > 0$ if*

$$\mathbb{P}\left(\text{exists } \varepsilon > 0 : N_\varepsilon > F_{UPAC}\left(S, A, H, 1/\varepsilon, \log(1/\delta)\right)\right) \leq \delta \,,$$

*where $F_{UPAC}$ is polynomial in all arguments.*

All the performance metrics are functions of the distribution of the sequence of errors over the episodes $(\Delta_k)_{k \in \mathbb{N}}$. Regret bounds are the integral of this sequence up to time $T$, which is a random variable. The expected regret is just the expectation of the integral, while the high-probability regret is a quantile. PAC bounds are the quantile of the size of the superlevel set for a fixed level $\varepsilon$. Uniform-PAC bounds are like PAC bounds, but hold for all $\varepsilon$ simultaneously.

**Limitations of regret.** Since regret guarantees only bound the integral of $\Delta_k$ over $k$, it does not distinguish between making a few severe mistakes and many small mistakes. In fact, since regret bounds provably grow with the number of episodes $T$, an algorithm that achieves optimal regret may still make infinitely many mistakes (of arbitrary quality, see proof of Theorem 2 below). This is highly undesirable in high-stakes scenarios. For example in drug treatment optimization in healthcare, we would like to distinguish between infrequent severe complications (few large $\Delta_k$) and frequent minor side effects (many small $\Delta_k$). In fact, even with an optimal regret bound, we could still serve infinitely patients with the worst possible treatment.

**Limitations of PAC.** PAC bounds limit the number of mistakes for a given accuracy level $\varepsilon$, but is otherwise non-restrictive. That means an algorithm with $\Delta_k > \varepsilon/2$ for all $k$ almost surely might still be $(\varepsilon, \delta)$-PAC. Worse, many algorithms designed to be $(\varepsilon, \delta)$-PAC actually exhibit this behavior because they explicitly halt learning once an $\varepsilon$-optimal policy has been found. The less widely used TCE (total cost of exploration) bounds [13] and KWIK guarantees [14] suffer from the same issueand for conciseness are not discussed in detail.

**Advantages of Uniform-PAC.** The new criterion overcomes the limitations of PAC and regret guarantees by measuring the number of $\varepsilon$-errors at every level simultaneously. By definition, algorithms that are Uniform-PAC for a $\delta$ are $(\varepsilon, \delta)$-PAC for all $\varepsilon > 0$. We will soon see that an algorithm with a non-trivial Uniform-PAC guarantee also has small regret with high probability. Furthermore, there is no loss in the reduction so that an algorithm with optimal Uniform-PAC guarantees also has optimal regret, at least in the episodic RL setting. In this sense Uniform-PAC is the missing bridge between regret and PAC. Finally, for algorithms based on confidence bounds, Uniform-PAC guarantees are usually obtained without much additional work by replacing standard concentration bounds with versions that hold uniformly over episodes (e.g. using the law of the iterated logarithms). In this sense we think Uniform-PAC is the new 'gold-standard' of theoretical guarantees for RL algorithms.

## 2.1 Relationships between Performance Guarantees

Existing theoretical analyses usually focus exclusively on either the regret or PAC framework. Besides occasional heuristic translations, Proposition 4 in [15] and Corollary 3 in [6] are the only results relating a notion of PAC and regret, we are aware of. Yet the guarantees there are not widely used[3]

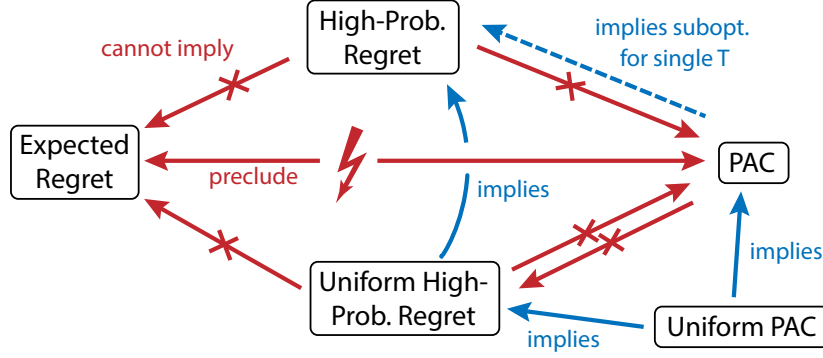

Figure 1: Visual summary of relationship among the different learning frameworks: Expected regret (ER) and PAC preclude each other while the other crossed arrows represent only a *does-not-implies* relationship. Blue arrows represent *imply* relationships. For details see the theorem statements.

unlike the definitions given above which we now formally relate to each other. A simplified overview of the relations discussed below is shown in Figure 1.

**Theorem 1.** *No algorithm can achieve*

- *a sub-linear expected regret bound for all $T$ and*
- *a finite $(\varepsilon, \delta)$-PAC bound for a small enough $\varepsilon$*

*simultaneously for all two-armed multi-armed bandits with Bernoulli reward distributions. This implies that such guarantees also cannot be satisfied simultaneously for all episodic MDPs.*

A full proof is in Appendix A.1, but the intuition is simple. Suppose a two-armed Bernoulli bandit has mean rewards $1/2 + \varepsilon$ and $1/2$ respectively and the second arm is chosen at most $F < \infty$ times with probability at least $1 - \delta$, then one can easily show that in an alternative bandit with mean rewards $1/2 + \varepsilon$ and $1/2 + 2\varepsilon$ there is a non-zero probability that the second arm is played finitely often and in this bandit the expected regret will be linear. Therefore, sub-linear expected regret is only possible if each arm is pulled infinitely often almost surely.

**Theorem 2.** *The following statements hold for performance guarantees in episodic MDPs:*

(a) *If an algorithm satisfies a $(\varepsilon, \delta)$-PAC bound with $F_{PAC} = \Theta(1/\varepsilon^2)$ then it satisfies for a specific $T = \Theta(\varepsilon^{-3})$ a $F_{HPR} = \Theta(T^{2/3})$ bound. Further, there is an MDP and algorithm that satisfies the $(\varepsilon, \delta)$-PAC bound $F_{PAC} = \Theta(1/\varepsilon^2)$ on that MDP and has regret $R(T) = \Omega(T^{2/3})$ on that MDP for any $T$. That means a $(\varepsilon, \delta)$-PAC bound with $F_{PAC} = \Theta(1/\varepsilon^2)$ can only be converted to a high-probability regret bound with $F_{HPR} = \Omega(T^{2/3})$.*

(b) *For any chosen $\varepsilon, \delta > 0$ and $F_{PAC}$, there is an MDP and algorithm that satisfies the $(\varepsilon, \delta)$-PAC bound $F_{PAC}$ on that MDP and has regret $R(T) = \Omega(T)$ on that MDP. That means a $(\varepsilon, \delta)$-PAC bound cannot be converted to a sub-linear uniform high-probability regret bound.*

(c) *For any $F_{UHPR}(T, \delta)$ with $F_{UHPR}(T, \delta) \to \infty$ as $T \to \infty$, there is an algorithm that satisfies that uniform high-probability regret bound on some MDP but makes infinitely many mistakes for any sufficiently small accuracy level $\varepsilon > 0$ for that MDP. Therefore, a high-probability regret bound (uniform or not) cannot be converted to a finite $(\varepsilon, \delta)$-PAC bound.*

(d) *For any $F_{UHPR}(T, \delta)$ there is an algorithm that satisfies that uniform high-probability regret bound on some MDP but suffers expected regret $\mathbb{E}R(T) = \Omega(T)$ on that MDP.*

For most interesting RL problems including episodic MDPs the worst-case expected regret grows with $O(\sqrt{T})$. The theorem shows that establishing an optimal high probability regret bound does not imply any finite PAC bound. While PAC bounds may be converted to regret bounds, the resulting bounds are necessarily severely suboptimal with a rate of $T^{2/3}$. The next theorem formalises the claim that Uniform-PAC is stronger than both the PAC and high-probability regret criteria.

**Theorem 3.** *Suppose an algorithm is Uniform-PAC for some $\delta$ with $F_{UPAC} = \tilde{O}(C_1/\varepsilon + C_2/\varepsilon^2)$ where $C_1, C_2 > 0$ are constant in $\varepsilon$, but may depend on other quantities such as S, A, H, $\log(1/\delta)$, then the algorithm*

(a) *converges to optimal policies with high probability:* $\mathbb{P}(\lim_{k\to\infty} \Delta_k = 0) \geq 1 - \delta$.

(b) *is $(\varepsilon, \delta)$-PAC with bound $F_{PAC} = F_{UPAC}$ for all $\varepsilon$.*

(c) *enjoys a high-probability regret at level $\delta$ with $F_{UHPR} = \tilde{O}(\sqrt{C_2 T} + \max\{C_1, C_2\})$.*

Observe that stronger uniform PAC bounds lead to stronger regret bounds and for RL in episodic MDPs, an optimal uniform-PAC bound implies a uniform regret bound. To our knowledge, there are no existing approaches with PAC or regret guarantees that are Uniform-PAC. PAC methods such as MBIE, MoRMax, UCRL-$\gamma$, UCFH, Delayed Q-Learning or Median-PAC all depend on advance knowledge of $\varepsilon$ and eventually stop improving their policies. Even when disabling the stopping condition, these methods are not uniform-PAC as their confidence bounds only hold for finitely many episodes and are eventually violated according to the law of iterated logarithms. Existing algorithms with uniform high-probability regret bounds such as UCRL2 or UCBVI [16] also do not satisfy uniform-PAC bounds since they use upper confidence bounds with width $\sqrt{\log(T)/n}$ where $T$ is the number of observed episodes and $n$ is the number of observations for a specific state and action. The presence of $\log(T)$ causes the algorithm to try each action in each state infinitely often. One might begin to wonder if uniform-PAC is too good to be true. Can *any* algorithm meet the requirements? We demonstrate in Section 4 that the answer is yes by showing that UBEV has meaningful Uniform-PAC bounds. A key technique that allows us to prove these bounds is the use of finite-time law of iterated logarithm confidence bounds which decrease at rate $\sqrt{(\log\log n)/n}$.

## 3 The UBEV Algorithm

The pseudo-code for the proposed UBEV algorithm is given in Algorithm 1. In each episode it follows an optimistic policy $\pi_k$ that is computed by backwards induction using a carefully chosen confidence interval on the transition probabilities in each state. In line 8 an optimistic estimate of the Q-function for the current state-action-time triple is computed using the empirical estimates of the expected next state value $\hat{V}_{\text{next}} \in \mathbb{R}$ (given that the values at the next time are $\tilde{V}_{t+1}$) and expected immediate reward $\hat{r}$ plus confidence bounds $(H-t)\phi$ and $\phi$. We show in Lemma D.1 in the appendix that the policy update in Lines 3–9 finds an optimal solution to $\max_{P',r',V',\pi'} \mathbb{E}_{s\sim p_0}[V_1'(s)]$ subject to the constraints that for all $s \in \mathcal{S}, a \in \mathcal{A}, t \in [H]$,

$$V_t'(s) = r(s, \pi'(s,t), t) + P'(s, \pi'(s,t), t)^\top V_{t+1}' \quad \text{(Bellman Equation)} \quad (1)$$

$$V_{H+1}' = 0, \quad P'(s,a,t) \in \Delta_S, \quad r'(s,a,t) \in [0,1]$$

$$|[(P' - \hat{P}_k)(s,a,t)]^\top V_{t+1}'| \leq \phi(s,a,t)(H-t)$$

$$|r'(s,a,t) - \hat{r}_k(s,a,t)| \leq \phi(s,a,t) \quad (2)$$

where $(P' - \hat{P}_k)(s,a,t)$ is short for $P'(s,a,t) - \hat{P}_k(s,a,t) = P'(\cdot|s,a,t) - \hat{P}_k(\cdot|s,a,t)$ and

$$\phi(s,a,t) = \sqrt{\frac{2\ln\ln\max\{e, n(s,a,t)\} + \ln(18SAH/\delta)}{n(s,a,t)}} = O\left(\sqrt{\frac{\ln(SAH\ln(n(s,a,t))/\delta)}{n(s,a,t)}}\right)$$

is the width of a confidence bound with $e = \exp(1)$ and $\hat{P}_k(s'|s,a,t) = \frac{m(s',s,a,t)}{n(s,a,t)}$ are the empirical transition probabilities and $\hat{r}_k(s,a,t) = l(s,a,t)/n(s,a,t)$ the empirical immediate rewards (both at the beginning of the $k$th episode). Our algorithm is conceptually similar to other algorithms based on the optimism principle such as MBIE [5], UCFH [3], UCRL2 [6] or UCRL-$\gamma$ [2] but there are several key differences:

- Instead of using confidence intervals over the transition kernel by itself, we incorporate the value function directly into the concentration analysis. Ultimately this saves a factor of $S$ in the sample complexity, but the price is a more difficult analysis. Previously MoRMax [17] also used the idea of directly bounding the transition and value function, but in a very different algorithm that required discarding data and had a less tight bound. A similar technique has been used by Azar et al. [16].

**Algorithm 1:** UBEV (**U**pper **B**ounding the **E**xpected Next State **V**alue) Algorithm

**Input**: failure tolerance $\delta \in (0, 1]$

1   $n(s, a, t) = l(s, a, t) = m(s', s, a, t) = 0; \quad \tilde{V}_{H+1}(s') := 0 \quad \forall s, s' \in \mathcal{S}, a \in \mathcal{A}, t \in [H]$

2   **for** $k = 1, 2, 3, \ldots$ **do**

     /* Optimistic planning                                                    */

3      **for** $t = H$ **to** $1$ **do**

4          **for** $s \in \mathcal{S}$ **do**

5              **for** $a \in \mathcal{A}$ **do**

6                  $\phi := \sqrt{\frac{2 \ln\ln(\max\{e, n(s,a,t)\}) + \ln(18SAH/\delta)}{n(s,a,t)}}$   // confidence bound

7                  $\hat{r} := \frac{l(s,a,t)}{n(s,a,t)}; \quad \hat{V}_{\text{next}} := \frac{m(\cdot, s, a, t)^\top \tilde{V}_{t+1}}{n(s,a,t)}$   // empirical estimates

8                  $Q(a) := \min\{1, \hat{r} + \phi\} + \min\left\{\max \tilde{V}_{t+1}, \hat{V}_{\text{next}} + (H - t)\phi\right\}$

9              $\pi_k(s, t) := \arg\max_a Q(a), \quad \tilde{V}_t(s) := Q(\pi_k(s, t))$

     /* Execute policy for one episode                                     */

10      $s_1 \sim p_0;$

11      **for** $t = 1$ **to** $H$ **do**

12          $a_t := \pi_k(s_t, t), \; r_t \sim p_R(s_t, a_t, t)$ and $s_{t+1} \sim P(s_t, a_t, t)$

13          $n(s_t, a_t, t)++; \quad m(s_{t+1}, s_t, a_t, t)++; \quad l(s_t, a_t, t) += r_t$   // update statistics

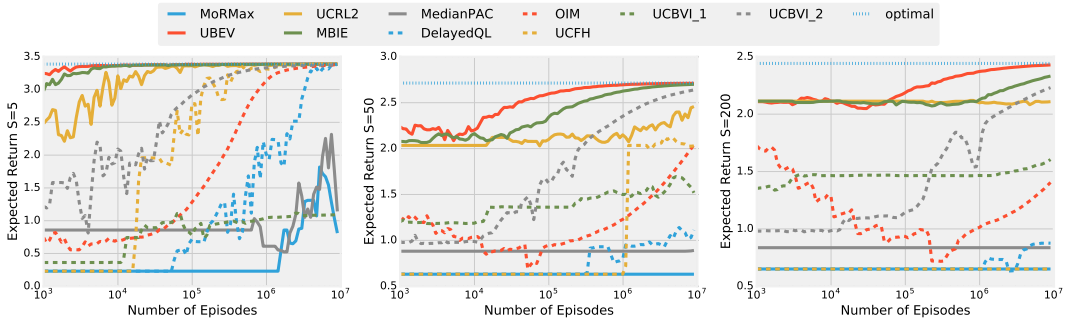

Figure 2: Empirical comparison of optimism-based algorithms with frequentist regret or PAC bounds on a randomly generated MDP with 3 actions, time horizon 10 and $S = 5, 50, 200$ states. All algorithms are run with parameters that satisfy their bound requirements. A detailed description of the experimental setup including a link to the source code can be found in Appendix B.

- Many algorithms update their policy less and less frequently (usually when the number of samples doubles), and only finitely often in total. Instead, we update the policy after every episode, which means that UBEV immediately leverages new observations.

- Confidence bounds in existing algorithms that keep improving the policy (e.g. Jaksch et al. [6], Azar et al. [16]) scale at a rate $\sqrt{\log(k)/n}$ where $k$ is the number of episodes played so far and $n$ is the number of times the specific $(s, a, t)$ has been observed. As the results of a brief empirical comparison in Figure 2 indicate, this leads to slow learning (compare UCBVI_1 and UBEV's performance which differ essentially only by their use of different rate bounds). Instead the width of UBEV's confidence bounds $\phi$ scales at rate $\sqrt{\ln\ln(\max\{e, n\})/n} \approx \sqrt{(\log\log n)/n}$ which is the best achievable rate and results in significantly faster learning.

## 4   Uniform PAC Analysis

We now discuss the Uniform-PAC analysis of UBEV which results in the following Uniform-PAC and regret guarantee.

**Theorem 4.** *Let $\pi_k$ be the policy of UBEV in the kth episode. Then with probability at least $1 - \delta$ for all $\varepsilon > 0$ jointly the number of episodes $k$ where the expected return from the start state is not $\varepsilon$-optimal (that is $\hat{\Delta}_k > \varepsilon$) is at most*

$$O\left(\frac{SAH^4}{\varepsilon^2}\min\left\{1+\varepsilon S^2 A, S\right\}\text{polylog}\left(A, S, H, \frac{1}{\varepsilon}, \frac{1}{\delta}\right)\right).$$

*Therefore, with probability at least $1 - \delta$ UBEV converges to optimal policies and for all episodes $T$ has regret*

$$R(T) = O\left(H^2(\sqrt{SAT} + S^3 A^2)\text{polylog}(S, A, H, T)\right).$$

Here $\text{polylog}(x\ldots)$ is a function that can be bounded by a polynomial of logarithm, that is, $\exists k, C : \text{polylog}(x\ldots) \leq \ln(x\ldots)^k + C$. In Appendix C we provide a lower bound on the sample complexity that shows that if $\varepsilon < 1/(S^2 A)$, the Uniform-PAC bound is tight up to log-factors and a factor of $H$. To our knowledge, UBEV is the first algorithm with both near-tight (up to $H$ factors) high probability regret and $(\varepsilon, \delta)$ PAC bounds as well as the first algorithm with any nontrivial uniform-PAC bound.

Using Theorem 3 the convergence and regret bound follows immediately from the uniform PAC bound. After a discussion of the different confidence bounds allowing us to prove uniform-PAC bounds, we will provide a short proof sketch of the uniform PAC bound.

## 4.1 Enabling Uniform PAC With Law-of-Iterated-Logarithm Confidence Bounds

To have a PAC bound for all $\varepsilon$ jointly, it is critical that UBEV continually make use of new experience. If UBEV stopped leveraging new observations after some fixed number, it would not be able to distinguish with high probability among which of the remaining possible MDPs do or do not have optimal policies that are sufficiently optimal in the other MDPs. The algorithm therefore could potentially follow a policy that is not at least $\varepsilon$-optimal for infinitely many episodes for a sufficiently small $\varepsilon$. To enable UBEV to incorporate all new observations, the confidence bounds in UBEV must hold for an infinite number of updates. We therefore require a proof that the total probability of all possible failure events (of the high confidence bounds not holding) is bounded by $\delta$, in order to obtain high probability guarantees. In contrast to prior $(\varepsilon, \delta)$-PAC proofs that only consider a finite number of failure events (which is enabled by requiring an RL algorithm to stop using additional data), we must bound the probability of an infinite set of possible failure events.

Some choices of confidence bounds will hold uniformly across all sample sizes but are not sufficiently tight for uniform PAC results. For example, the recent work by Azar et al. [16] uses confidence intervals that shrink at a rate of $\sqrt{\frac{\ln T}{n}}$, where $T$ is the number of episodes, and $n$ is the number of samples of a $(s, a)$ pair at a particular time step. This confidence interval will hold for all episodes, but these intervals do not shrink sufficiently quickly and can even increase. One simple approach for constructing confidence intervals that is sufficient for uniform PAC guarantees is to combine bounds for fixed number of samples with a union bound allocating failure probability $\delta/n^2$ to the failure case with $n$ samples. This results in confidence intervals that shrink at rate $\sqrt{1/n \ln n}$. Interestingly we know of no algorithms that do such in our setting.

We follow a similarly simple but much stronger approach of using law-of-iterated logarithm (LIL) bounds that shrink at the better rate of $\sqrt{1/n \ln \ln n}$. Such bounds have sparked recent interest in sequential decision making [18, 19, 20, 21, 22] but to the best of our knowledge we are the first to leverage them for RL. We prove several general LIL bounds in Appendix F and explain how we use these results in our analysis in Appendix E.2. These LIL bounds are both sufficient to ensure uniform PAC bounds, and much tighter (and therefore will lead to much better performance) than $\sqrt{1/n \ln T}$ bounds. Indeed, LIL have the tightest possible rate dependence on the number of samples $n$ for a bound that holds for all timesteps (though they are not tight with respect to constants).

## 4.2 Proof Sketch

We now provide a short overview of our uniform PAC bound in Theorem 4. It follows the typical scheme for optimism based algorithms: we show that in each episode UBEV follows a policy that is

optimal with respect to the MDP $\tilde{M}_k$ that yields highest return in a set of MDPs $\mathcal{M}_k$ given by the constraints in Eqs. (1)–(2) (Lemma D.1 in the appendix). We then define a failure event $F$ (more details see below) such that on the complement $F^C$, the true MDP is in $\mathcal{M}_k$ for all $k$.

Under the event that the true MDP is in the desired set, the $V_1^\pi \leq V_1^\star \leq \tilde{V}_1^{\pi_k}$, i.e., the value $\tilde{V}_1^{\pi_k}$ of $\pi_k$ in MDP $\tilde{M}_k$ is higher than the optimal value function of the true MDP $M$ (Lemma E.16). Therefore, the optimality gap is bounded by $\Delta_k \leq p_0^\top (\tilde{V}_1^{\pi_k} - V_1^{\pi_k})$. The right hand side this expression is then decomposed via a standard identity (Lemma E.15) as

$$\sum_{t=1}^H \sum_{(s,a)\in\mathcal{S}\times\mathcal{A}} w_{tk}(s,a)((\tilde{P}_k - P)(s,a,t))^\top \tilde{V}_{t+1}^{\pi_k} + \sum_{t=1}^H \sum_{(s,a)\in\mathcal{S}\times\mathcal{A}} w_{tk}(s,a)(\tilde{r}_k(s,a,t) - r(s,a,t)),$$

where $w_{tk}(s,a)$ is the probability that when following policy $\pi_k$ in the true MDP we encounter $s_t = s$ and $a_t = a$. The quantities $\tilde{P}_k, \tilde{r}_k$ are the model parameters of the optimistic MDP $\tilde{M}_k$ For the sake of conciseness, we ignore the second term above in the following which can be bounded by $\varepsilon/3$ in the same way as the first. We further decompose the first term as

$$\sum_{\substack{t\in[H]\\(s,a)\in L_{tk}^c}} w_{tk}(s,a)((\tilde{P}_k - P)(s,a,t))^\top \tilde{V}_{t+1}^{\pi_k} \tag{3}$$

$$+ \sum_{\substack{t\in[H]\\(s,a)\in L_{tk}}} w_{tk}(s,a)((\tilde{P}_k - \hat{P}_k)(s,a,t))^\top \tilde{V}_{t+1}^{\pi_k} + \sum_{\substack{t\in[H]\\(s,a)\in L_{tk}}} w_{tk}(s,a)((\hat{P}_k - P)(s,a,t))^\top \tilde{V}_{t+1}^{\pi_k} \tag{4}$$

where $L_{tk} = \left\{(s,a)\in\mathcal{S}\times\mathcal{A} : w_{tk}(s,a) \geq w_{\min} = \frac{\varepsilon}{3HS^2}\right\}$ is the set of state-action pairs with non-negligible visitation probability. The value of $w_{\min}$ is chosen so that (3) is bounded by $\varepsilon/3$. Since $\tilde{V}^{\pi_k}$ is the optimal solution of the optimization problem in Eq. (1), we can bound

$$|((\tilde{P}_k - \hat{P}_k)(s,a,t))^\top \tilde{V}_{t+1}^{\pi_k}| \leq \phi_k(s,a,t)(H-t) = O\left(\sqrt{\frac{H^2 \ln(\ln(n_{tk}(s,a))/\delta)}{n_{tk}(s,a)}}\right), \tag{5}$$

where $\phi_k(s,a,t)$ is the value of $\phi(s,a,t)$ and $n_{tk}(s,a)$ the value of $n(s,a,t)$ right before episode $k$. Further we decompose

$$|((\hat{P}_k - P)(s,a,t))^\top \tilde{V}_{t+1}^{\pi_k}| \leq \|(\hat{P}_k - P)(s,a,t)\|_1 \|\tilde{V}_{t+1}^{\pi_k}\|_\infty \leq O\left(\sqrt{\frac{SH^2 \ln\frac{\ln n_{tk}(s,a)}{\delta}}{n_{tk}(s,a)}}\right), \tag{6}$$

where the second inequality follows from a standard concentration bound used in the definition of the failure event $F$ (see below). Substituting this and (5) into (4) leads to

$$(4) \leq O\left(\sum_{t=1}^H \sum_{s,a\in L_{tk}} w_{tk}(s,a)\sqrt{\frac{SH^2 \ln(\ln(n_{tk}(s,a))/\delta)}{n_{tk}(s,a)}}\right). \tag{7}$$

On $F^C$ it also holds that $n_{tk}(s,a) \geq \frac{1}{2}\sum_{i<k} w_{ti}(s,a) - \ln\frac{9SAH}{\delta}$ and so on *nice episodes* where each $(s,a)\in L_{tk}$ with significant probability $w_{tk}(s,a)$ also had significant probability in the past, i.e., $\sum_{i<k} w_{ti}(s,a) \geq 4\ln\frac{9SA}{\delta}$, it holds that $n_{tk}(s,a) \geq \frac{1}{4}\sum_{i<k} w_{ti}(s,a)$. Substituting this into (7), we can use a careful pidgeon-hole argument laid out it Lemma E.3 in the appendix to show that this term is bounded by $\varepsilon/3$ on all but $O(AS^2H^4/\varepsilon^2 \operatorname{polylog}(A,S,H,1/\varepsilon,1/\delta))$ nice episodes. Again using a pidgeon-hole argument, one can show that all but at most $O(S^2AH^3/\varepsilon \ln(SAH/\delta))$ episodes are nice. Combining both bounds, we get that on $F^C$ the optimality gap $\Delta_k$ is at most $\varepsilon$ except for at most $O(AS^2H^4/\varepsilon^2 \operatorname{polylog}(A,S,H,1/\varepsilon,1/\delta))$ episodes.

We decompose the failure event into multiple components. In addition to the events $F_k^N$ that a $(s,a,t)$ triple has been observed few times compared to its visitation probabilities in the past, i.e., $n_{tk}(s,a) < \frac{1}{2}\sum_{i<k} w_{ti}(s,a) - \ln\frac{9SAH}{\delta}$ as well as a conditional version of this statement, the failure event $F$ contains events where empirical estimates of the immediate rewards, the expected optimal value of the successor states and the individual transition probabilites are far from their true

expectations. For the full definition of $F$ see Appendix E.2. $F$ also contains event $F^{L1}$ we used in Eq. (6) defined as

$$\left\{ \exists k, s, a, t \; : \; \|\hat{P}_k(s, a, t) - P(s, a, t)\|_1 \geq \sqrt{\tfrac{4}{n_{tk}(s,a)} \left( 2 \operatorname{llnp}(n_{tk}(s,a)) + \ln \tfrac{18 S A H (2^S - 2)}{\delta} \right)} \right\}.$$

It states that the L1-distance of the empirical transition probabilities to the true probabilities for any $(s, a, t)$ in any episode $k$ is too large and we show that $\mathbb{P}(F^{L1}) \leq 1 - \delta/9$ using a uniform version of the popular bound by Weissman et al. [23] which we prove in Appendix F. We show in similar manner that the other events in $F$ have small probability uniformly for all episodes $k$ so that $\mathbb{P}(F) \leq \delta$. Together this yields the uniform PAC bound in Thm. 4 using the second term in the $\min$.

With a more refined analysis that avoids the use of Hölder's inequality in (6) and a stronger notion of nice episodes called friendly episodes we obtain the bound with the first term in the $\min$. However, since a similar analysis has been recently released [16], we defer this discussion to the appendix.

### 4.3 Discussion of UBEV Bound

The (Uniform-)PAC bound for UBEV in Theorem 4 is never worse than $\tilde{O}(S^2 A H^4 / \varepsilon^2)$, which improves on the similar MBIE algorithm by a factor of $H^2$ (after adapting the discounted setting for which MBIE was analysed to our setting). For $\varepsilon < 1/(S^2 A)$ our bound has a linear dependence on the size of the state-space and depends on $H^4$, which is a tighter dependence on the horizon than MoRMax's $\tilde{O}(S A H^6 / \varepsilon^2)$, the best sample-complexity bound with linear dependency $S$ so far.

Comparing UBEV's regret bound to the ones of UCRL2 [6] and REGAL [24] requires care because (a) we measure the regret over entire episodes and (b) our transition dynamics are time-dependent within each episode, which effectively increases the state-space by a factor of $H$. Converting the bounds for UCRL2/REGAL to our setting yields a regret bound of order $S H^2 \sqrt{A H T}$. Here, the diameter is $H$, the state space increases by $H$ due to time-dependent transition dynamics and an additional $\sqrt{H}$ is gained by stating the regret in terms of episodes $T$ instead of time steps. Hence, UBEV's bounds are better by a factor of $\sqrt{SH}$. Our bound matches the recent regret bound for episodic RL by Azar et al. [16] in the $S$, $A$ and $T$ terms but not in $H$. Azar et al. [16] has regret bounds that are optimal in $H$ but their algorithm is not uniform PAC, due to the characteristics we outlined in Section 2.

## 5 Conclusion

The Uniform-PAC framework strengthens and unifies the PAC and high-probability regret performance criteria for reinforcement learning in episodic MDPs. The newly proposed algorithm is Uniform-PAC, which as a side-effect means it is the first algorithm that is both PAC and has sublinear (and nearly optimal) regret. Besides this, the use of law-of-the-iterated-logarithm confidence bounds in RL algorithms for MDPs provides a practical and theoretical boost at no cost in terms of computation or implementation complexity.

This work opens up several immediate research questions for future work. The definition of Uniform-PAC and the relations to other PAC and regret notions directly apply to multi-armed bandits and contextual bandits as special cases of episodic RL, but not to infinite horizon reinforcement learning. An extension to these non-episodic RL settings is highly desirable. Similarly, a version of the UBEV algorithm for infinite-horizon RL with linear state-space sample complexity would be of interest. More broadly, if theory is ever to say something useful about practical algorithms for large-scale reinforcement learning, then it will have to deal with the unrealizable function approximation setup (unlike the tabular function representation setting considered here), which is a major long-standing open challenge.

**Acknowledgements**. We appreciate the support of a NSF CAREER award and a gift from Yahoo.

## Footnotes

[2]The reward may be allowed to depend on the next-state with no further effort in the proofs. The boundedness assumption could be replaced by the assumption of subgaussian noise with known subgaussian parameter.

[3]The average per-step regret in [6] is superficially a PAC bound, but does not hold over infinitely many time-steps and exhibits the limitations of a conventional regret bound. The translation to average loss in [15] comes at additional costs due to the discounted infinite horizon setting.

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
