[Supplementary Material 1]

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

# Appendices of Unifying PAC and Regret: Uniform PAC Bounds for Episodic Reinforcement Learning

# A  Framework Relation Proofs

## A.1  Proof of Theorem 1

*Proof.* We will use two episodic MDPs, $M_1$ and $M_2$, which are essentially 2-armed bandits and hard to distinguish to prove this statement. Both MDPs have one state, horizon $H = 1$, and two actions $\mathcal{A} = \{1, 2\}$. For a fixed $\alpha > 0$, the rewards are Bernoulli$(1/2 + \alpha/2)$ distributed for actions 1 in both MDPs. Playing action 2 in $M_1$ gives Bernoulli$(1/2)$ rewards and action 2 in $M_2$ gives Bernoulli$(1/2 + \alpha)$ rewards.

Assume now that an algorithm in MDP $M_1$ with nonzero probability plays the suboptimal action only at most $N$ times in total, i.e., $\mathbb{P}_{M_1}(n_2 \leq N) \geq \beta$ where $n_2$ is the number of times action 2 is played and $\infty > N > 0, \beta > 0$. Then

$$\mathbb{P}_{M_1}(n_2 \leq N) = \mathbb{E}_{M_1}\left[\mathbb{I}\{n_2 \leq N\}\right] = \mathbb{E}_{M_2}\left[\frac{\mathbb{P}_{M_1}(Y_\infty)}{\mathbb{P}_{M_2}(Y_\infty)}\mathbb{I}\{n_2 \leq N\}\right]$$

where $Y_k = (A_1, R_1, A_2, R_2, \ldots A_k, R_k)$ denotes the entire sequence of observed rewards $R_i$ and action indices $A_i$ after $k$ episodes. Since $\mathbb{P}_{M_1}(A_k|Y_{k-1}) = \mathbb{P}_{M_2}(A_k|Y_{k-1})$ and $\mathbb{P}_{M_1}(R_k|A_k = 1, Y_{k-1}) = \mathbb{P}_{M_2}(R_k|A_k = 1, Y_{k-1})$ and

$$\frac{\mathbb{P}_{M_1}(R_k|A_k = 2, Y_{k-1})}{\mathbb{P}_{M_2}(R_k|A_k = 2, Y_{k-1})} \leq \max\left\{\frac{1/2}{1/2 + \alpha}, \frac{1/2}{1/2 - \alpha}\right\} = \frac{1}{1 - 2\alpha}$$

the likelihood ratio of $Y_\infty$ is upper bounded by $(1 + 2\alpha)^N$ if the second action has been chosen at most $N$ times. Hence

$$\mathbb{P}_{M_2}[n_2 \leq N] = \frac{(1 - 2\alpha)^N}{(1 - 2\alpha)^N}\mathbb{E}_{M_2}\left[\mathbb{I}\{n_2 \leq N\}\right] \geq (1 - 2\alpha)^N \mathbb{E}_{M_2}\left[\frac{\mathbb{P}_{M_1}(Y_\infty)}{\mathbb{P}_{M_2}(Y_\infty)}\mathbb{I}\{n_2 \leq N\}\right]$$

$$\geq (1 - 2\alpha)^N \beta > 0$$

Therefore, the regret for $M_2$ is for $T$ large enough $\mathbb{E}_{M_2}R(T) \geq (T - N)\beta(1 - 2\alpha)^N\alpha/2 = O(T)$. Hence, for the algorithm to ensure sublinear regret for $M_2$, it has to play the suboptimal action for $M_1$ infinitely often with probability 1. This however implies that the algorithm cannot satisfy any finite PAC bound for accuracy $\varepsilon < \alpha/2$. □

## A.2  Proof of Theorem 2

*Proof.* **PAC Bound to high-probability regret bound:** Consider a fixed $\delta > 0$ and PAC bound with $F_{\text{PAC}} = \Theta(1/\varepsilon^2)$. Then there is a $C > 0$ such that the following algorithm satisfies the PAC bound. The algorithm uses the worst possible policy with optimality gap $H$ in all episodes on some event $E$ and in the first $C/\varepsilon^2$ episodes on the complimentary event $E^C$. For the remaining episodes on $E^C$ it follows a policy with optimality gap $\varepsilon$. The probability of $E$ is $\delta$. The regret of the algorithm on $E$ is $R(T) = TH$ and on $E^C$ it is $R(T) = \min\{T, C/\varepsilon^2\}H + \min\{T - C/\varepsilon^2, 0\}\varepsilon$. For $T \geq C/\varepsilon^2$, on any event the regret of this algorithm is at least

$$R(T) = \frac{CH}{\varepsilon^2} + \left(T - \frac{C}{\varepsilon^2}\right)\varepsilon = T\varepsilon + \frac{C(H - \varepsilon)}{\varepsilon^2}. \tag{8}$$

The quantity

$$\frac{R(T)}{T^{2/3}} = \frac{C(H - \varepsilon)}{T^{2/3}\varepsilon^2} + \varepsilon T^{1/3}$$

takes its minimum at $T = \frac{C(H-\varepsilon)}{\varepsilon^3}$ with a positive value and hence $R(T) = \Omega(T^{2/3})$. Therefore a PAC bound with rate $1/\varepsilon^2$ implies at best a high-probability regret bound of order $O(T^{2/3})$ and is only tight at $T = \Theta(1/\varepsilon^3)$. Furthermore, by looking at Equation (8), we see that for any fixed $\varepsilon$, there is an algorithm that has uniform high-probability regret that is $\Omega(T)$.

**PAC Bound to uniform high-probability regret bound:** Consider a fixed $\delta > 0$ and $\varepsilon > 0$ and a PAC bound $F_{\text{PAC}}$ that evaluates to some value $N$ for parameter $\varepsilon$. The algorithm uses the worst possible policy with optimality gap $H$ in all episodes on some event $E$ and in the first $N$ episodes on

Figure 3: Relation of PAC-bound and Regret; The area of the shaded regions are a bound on the regret after $T$ episodes.

the complimentary event $E^C$. For the remaining episodes on $E^C$ it follows a policy with optimality gap $\varepsilon$. The probability of $E$ is $\delta$. The regret of the algorithm on $E$ is $R(T) = TH$ and on $E^C$ it is $R(T) = \min\{T, N\}H + \min\{T - N, 0\}\varepsilon$. For $T \geq N$, on any event the regret of this algorithm is at least

$$R(T) = NH + (T - N)\varepsilon = T\varepsilon + H(T - N) = \Omega(T).$$

**Uniform high-probability regret bound to PAC bound:** Consider an MDP such that at least one suboptimal policy exists with optimality gap $\varepsilon > 0$. Further let $L(T)$ be a nondecreasing function with $F_{\text{UHPR}}(T) \geq L(T)$ and $L(T) \to \infty$ as $T \to \infty$. Then the algorithm plays the optimal policy except for episodes $k$ where $\lfloor L(k-1)/\varepsilon \rfloor \neq \lfloor L(k)/\varepsilon \rfloor$. This algorithm satisfies the regret bound but makes infinitely many $\varepsilon/2$-mistakes with probability 1.

**Uniform high-probability regret bound to expected regret bound:** Consider an MDP such that at least one suboptimal policy exists with optimality gap $\varepsilon > 0$. Consider an algorithm that with probability $\delta$ always plays the suboptimal policy and with probability $1 - \delta$ always plays the optimal policy. This algorithm satisfies the uniform high-probability regret bound but suffers regret $\mathbb{E}R(T) = \delta\varepsilon T = \Omega(T)$. □

### A.3 Proof of Theorem 3

*Proof.* **Convergence to optimal policies:** The convergence to the set of optimal policies follows directly by using the definition of limits on the $\Delta_k$ sequence for each outcome in the high-probability event where the bound holds.

$(\varepsilon, \delta)$-**PAC:** Due to sub-additivity of probabilities, we have

$$\mathbb{P}\left(N_\varepsilon > F_{\text{PAC}}\left(\frac{1}{\varepsilon}, \log\frac{1}{\delta}\right)\right) \leq \mathbb{P}\left(\bigcup_{\varepsilon'}\left\{N_{\varepsilon'} > F_{\text{PAC}}\left(\frac{1}{\varepsilon'}, \log\frac{1}{\delta}\right)\right\}\right)$$

$$= \mathbb{P}\left(\exists\varepsilon' \,:\, N_{\varepsilon'} > F_{\text{PAC}}\left(\frac{1}{\varepsilon'}, \log\frac{1}{\delta}\right)\right) \leq \delta.$$

**High-Probability Regret Bound:** This part is proved separately in Theorem A.1 below. □

**Theorem A.1** (Uniform-PAC to Regret Conversion Theorem). *Assume on some event $E$ an algorithm follows for all $\varepsilon$ an $\varepsilon$-optimal policy $\pi_k$, i.e., $\Delta_k \leq \varepsilon$, on all but at most*

$$\frac{C_1}{\varepsilon}\left(\ln\frac{C_3}{\varepsilon}\right)^k + \frac{C_2}{\varepsilon^2}\left(\ln\frac{C_3}{\varepsilon}\right)^{2k}$$

*episodes where $C_1 \geq C_2 \geq 2$ and $C_3 \geq \max\{H, e\}$ and $C_1, C_2, C_3$ do not depend on $\varepsilon$. Then this algorithm has on this event a regret of*

$$R(T) \leq (\sqrt{C_2 T} + C_1)\,\text{polylog}(T, C_3, C_1) = O(\sqrt{C_2 T}\,\text{polylog}(T, C_3, C_1, H))$$

*for all number of episodes $T$.*

*Proof.* The mistake bound $g(\varepsilon) = \frac{C_1}{\varepsilon}\left(\ln\frac{C_3}{\varepsilon}\right)^k + \frac{C_2}{\varepsilon^2}\left(\ln\frac{C_3}{\varepsilon}\right)^{2k} \leq T$ is monotonically decreasing for $\varepsilon \in (0, H]$. For a given $T$ large enough, we can therefore find an $\varepsilon_{\min} \in (0, H]$ such that $g(\varepsilon) \leq T$ for all $\varepsilon \in (\varepsilon_{\min}, H]$. The regret $R(T)$ of the algorithm can then be bounded as follows

$$R(T) \leq T\varepsilon_{\min} + \int_{\varepsilon_{\min}}^{H} g(\varepsilon)d\varepsilon.$$

This bound assumes the worst case where first the algorithm makes the worst mistakes possible with regret $H$ and subsequently less and less severe mistakes controlled by the mistake bound. For a better intuition, see Figure 3.

We first find a suitable $\varepsilon_{\min}$. Define $y = \frac{1}{\varepsilon}\left(\ln\frac{C_3}{\varepsilon}\right)^k$ then since $g$ is monotonically decreasing, it is sufficient to find a $\varepsilon$ with $g(\varepsilon) \leq T$. That is equivalent to $C_1 y + C_2 y^2 \leq T$ for which

$$\frac{1}{\varepsilon}\left(\ln\frac{C_3}{\varepsilon}\right)^k = y \leq \frac{C_1}{2C_2} + \frac{\sqrt{C_1^2 + 4TC_2}}{2C_2} =: a$$

is sufficient. We set now

$$\varepsilon_{\min} = \frac{\ln(C_3 a)^k}{a} = \frac{2C_2}{C_1 + \sqrt{C_1^2 + 4TC_2}}\left(\ln\frac{(C_1 + \sqrt{C_1^2 + 4TC_2})C_3}{2C_2}\right)^k$$

which is a valid choice as

$$\frac{1}{\varepsilon_{\min}}\left(\ln\frac{C_3}{\varepsilon_{\min}}\right)^k = \frac{a}{\ln(C_3 a)^k}\left(\ln\frac{C_3 a}{\ln(C_3 a)^k}\right)^k = \frac{a}{\ln(C_3 a)^k}\left(\ln(C_3 a) - k\ln\ln(C_3 a)\right)^k$$
$$\leq \frac{a}{\ln(C_3 a)^k}\left(\ln(C_3 a)\right)^k = a.$$

We now first bound the regret further as

$$R(T) \leq T\varepsilon_{\min} + \int_{\varepsilon_{\min}}^{H} g(\varepsilon)d\varepsilon \leq T\varepsilon_{\min} + C_1\left(\ln\frac{C_3}{\varepsilon_{\min}}\right)^k\int_{\varepsilon_{\min}}^{H}\frac{1}{\varepsilon}d\varepsilon + C_2\left(\ln\frac{C_3}{\varepsilon_{\min}}\right)^{2k}\int_{\varepsilon_{\min}}^{H}\frac{1}{\varepsilon^2}d\varepsilon$$

$$= T\varepsilon_{\min} + C_1\left(\ln\frac{C_3}{\varepsilon_{\min}}\right)^k\ln\frac{H}{\varepsilon_{\min}} + C_2\left(\ln\frac{C_3}{\varepsilon_{\min}}\right)^{2k}\left[\frac{1}{\varepsilon_{\min}} - \frac{1}{H}\right]$$

and then use the choice of $\varepsilon_{\min}$ from above to look at each of the terms in this bound individually. In the following bounds we extensively use the fact $\ln(a+b) \leq \ln(a) + \ln(b) = \ln(ab)$ for all $a, b \geq 2$ and that $\sqrt{a+b} \leq \sqrt{a} + \sqrt{b}$ which holds for all $a, b \geq 0$.

$$T\varepsilon_{\min} = \frac{2TC_2}{C_1 + \sqrt{C_1^2 + 4TC_2}}\left(\ln\frac{C_3(C_1 + \sqrt{C_1^2 + 4TC_2})}{2C_2}\right)^k$$
$$\leq \frac{2TC_2}{\sqrt{4TC_2}}\left(\ln C_3 + \ln C_1 + \ln C_1 + \ln\frac{2\sqrt{TC_2}}{2C_2}\right)^k$$
$$\leq \sqrt{TC_2}\left(\ln(C_3 C_1^2\sqrt{T})\right)^k$$

Now for a $C \geq 0$ we first look at

$$\ln\frac{C}{\varepsilon_{\min}} = \ln C + \ln\frac{C_1 + \sqrt{C_1^2 + 4TC_2}}{2C_2} - k\ln\ln\frac{C_3(C_1 + \sqrt{C_1^2 + 4TC_2})}{2C_2}$$
$$\leq \ln C + \ln\frac{C_1 + \sqrt{C_1^2 + 4TC_2}}{2C_2}$$
$$\leq \ln C + \ln C_1 + \ln C_1 + \ln\frac{\sqrt{4TC_2}}{2C_2}$$
$$\leq \ln(CC_1^2\sqrt{T})$$

where the first inequality follows from the fact that $\frac{C_3(C_1+\sqrt{C_1^2+4TC_2})}{2C_2} \geq \frac{C_3 2C_1}{2C_2} \geq e$. Hence, we can bound

$$C_1 \left( \ln \frac{C_3}{\varepsilon_{\min}} \right)^k \ln \frac{H}{\varepsilon_{\min}} \leq C_1 \left( \ln(C_3 C_1^2 \sqrt{T}) \right)^k \ln(H C_1^2 \sqrt{T}).$$

Now since

$$\frac{1}{\varepsilon_{\min}} = \frac{C_1 + \sqrt{C_1^2 + 4TC_2}}{2C_2} \left( \ln \frac{C_3(C_1 + \sqrt{C_1^2 + 4TC_2})}{2C_2} \right)^{-k} \leq \frac{C_1}{C_2} + \sqrt{\frac{T}{C_2}}$$

we get

$$C_2 \left( \ln \frac{C_3}{\varepsilon_{\min}} \right)^{2k} \left[ \frac{1}{\varepsilon_{\min}} - \frac{1}{H} \right] \leq C_2 \left( \ln(C_3 C_1^2 \sqrt{T}) \right)^{2k} \left[ \frac{C_1}{C_2} + \sqrt{\frac{T}{C_2}} \right]$$

$$\leq \left( \ln(C_3 C_1^2 \sqrt{T}) \right)^{2k} \left[ C_1 + \sqrt{TC_2} \right].$$

As a result we can conclude that $R(T) \leq (\sqrt{C_2 T} + C_1) \operatorname{polylog}(T, C_3, C_1, H) = O(\sqrt{C_2 T} \operatorname{polylog}(T, C_3, C_1, H))$. $\qquad\square$

## B  Experimental Details

We generated the MDPs with $S = 5, 50, 200$ states, $A = 3$ actions and $H = 10$ timesteps as follows: The transition probabilities $P(s, a, t)$ were sampled independently from Dirichlet $\left( \frac{1}{10}, \ldots \frac{1}{10} \right)$ and the rewards were all deterministic with their value $r(s, a, t)$ set to 0 with probability $85\%$ and set uniformly at random in $[0, 1]$ otherwise. This construction results in MDPs that have concentrated but non-deterministic transition probabilities and sparse rewards.

Since some algorithms have been proposed assuming the rewards $r(s, a, t)$ are known and we aim for a fair comparison, we assumed for all algorithms that the immediate rewards $r(s, a, t)$ are known and adapted the algorithms accordingly. For example, in UBEV, the $\min \left\{ 1, \frac{l(s,a,t)}{\max\{1, n(s,a,t)\}} + \phi \right\}$ term was replaced by the true known rewards $r(s, a, t)$ and the $\delta$ parameter in $\phi$ was scaled by $9/7$ accordingly since the concentration result for immediate rewards is not necessary in this case. We used $\delta = \frac{1}{10}$ for all algorithms and $\varepsilon = \frac{1}{10}$ if they require to know $\varepsilon$ beforehand.

We adapted MoRMax, UCRL2, UCFH, MBIE, MedianPAC, Delayed Q-Learning and OIM to the episodic MDP setting with time-dependent transition dynamics by using allowing them to learn time-dependent dynamics and use finite-horizon planning. We did adapt the confidence intervals and but did not re-derive the constants for each algorithm. When in doubt we opted for smaller constants typically resulting better performance of the competitors. We further replaced the range of the value function $O(H)$ by the observed range of the optimistic next state values in the confidence bounds. We also reduced the number of episodes used in the delays by a factor of $\frac{1}{1000}$ for MoRMax and Delayed Q-Learning and by $10^{-6}$ for UCFH because they would otherwise not have performed a single policy update even for $S = 5$ within the 10 million episodes we considered. This scaling violates their theoretical guarantees but at least shows that the methods work in principle.

The performance reported in Figure 2 are the expected return of the current policy of each algorithm averaged over 1000 episodes. The figure shows a single run of the same randomly generated MDP but the results are representative. We reran this experiments with different random seeds and consistently obtained qualitatively similar results.

Source code for the experiments including concise but efficient implementations of the algorithms is available at https://github.com/chrodan/FiniteEpisodicRL.jl.

## C  PAC Lower Bound

**Theorem C.1.** *There exist positive constants $c$, $\delta_0 > 0$, $\varepsilon_0 > 0$ such that for every $\varepsilon \in (0, \varepsilon_0)$, $S \geq 4, A \geq 2$ and for every algorithm $A$ that and $n \leq \frac{cASH^3}{\varepsilon^2}$ there is a fixed-horizon episodic MDP*

$M_{hard}$ with time-dependent transition probabilities and $S$ states and $A$ actions so that returning an $\varepsilon$-optimal policy after $n$ episodes is at most $1 - \delta_0$. That implies that no algorithm can have a PAC guarantee better than $\Omega\left(\frac{ASH^3}{\varepsilon^2}\right)$ for sufficiently small $\varepsilon$.

Note that this lower bound on the sample complexity of any method in episodic MDPs with time-dependent dynamics applies to the arbitrary but fixed $\varepsilon$ PAC bound and therefore immediately to the stronger uniform-PAC bounds. This theorem can be proved in the same way as Theorem 5 by Jiang et al. [4], which itself is a standard construction involving a careful layering of difficult instances of the multi-armed bandit problem.[4] For simplicity, we omitted the dependency on the failure probability $\delta$, but using the techniques in the proof of Theorem 26 by Strehl et al. [5], a lower bound of order $\Omega\left(\frac{ASH^3}{\varepsilon^2}\log(SA/\delta)\right)$ can be obtained. The lower bound shows for small $\varepsilon$ the sample complexity of UBEV given in Theorem 4 is optimal except for a factor of $H$ and logarithmic terms.

## D Planning Problem of UBEV

**Lemma D.1** (Planning Problem). *The policy update in Lines 3–9 of Algorithm 1 finds an optimal solution to the optimization problem*

$$\max_{P',V',\pi',r'} \mathbb{E}_{s\sim p_0}[V_1'(s)]$$

$$\forall s \in \mathcal{S}, a \in \mathcal{A}, t \in [H]:$$

$$V_{H+1}' = 0, \qquad P'(s,a,t) \in \Delta_S, \qquad r'(s,a,t) \in [0,1]$$

$$V_t'(s) = r'(s,\pi'(s,t),t) + \mathbb{E}_{s'\sim P'(s,\pi'(s,t),t)}[V_{t+1}']$$

$$|(P'(s,a,t) - \hat{P}_k(s,a,t))^\top V_{t+1}'| \leq \phi(s,a,t)(H-t)$$

$$|r'(s,a,t) - \hat{r}_k(s,a,t)| \leq \phi(s,a,t)$$

*where* $\phi(s,a,t) = \sqrt{\frac{2\,\mathrm{llnp}(n(s,a,t)) + \ln(18SAH/\delta)}{n(s,a,t)}}$ *is a confidence bound and* $\hat{P}_k(s'|s,a,t) = m(s',s,a,t)/n(s,a,t)$ *are the empirical transition probabilities and* $\hat{r}_k(s,a,t) = l(s,a,t)/n(s,a,t)$ *the empirical average rewards.*

*Proof.* Since $\tilde{V}_{H+1}(\cdot)$ is initialized with 0 and never changed, we immediately get that it is an optimal value for $V_{H+1}'(\cdot)$ which is constrained to be 0. Consider now a single time step $t$ and assume $V_{t+1}'$ are fixed to the optimal values $\tilde{V}_{t+1}$. Plugging in the computation of $Q(a)$ into the computation of $\tilde{V}_t(s)$, we get

$$\tilde{V}_t(s) = \max_a Q(a) = \max_{a\in\mathcal{A}} \left[ \min\{1, \hat{r}(s,a,t) + \phi(s,a,t)\} \right.$$

$$\left. + \min\left\{\max\tilde{V}_{t+1}, \mathbb{I}\{n(s,a,t) > 0\}(\hat{P}(s,a,t)^\top \tilde{V}_{t+1}) + \phi(s,a,t)(H-t)\right\} \right]$$

using the convention that $\hat{r}(s,a,t) = 0$ if $n(s,a,t) = 0$. Assuming that $V_{t+1}' = \tilde{V}_{t+1}$, and that our goal for now is to maximize $\tilde{V}_t(s)$, this can be rewritten as

$$\max_{P'(s,a,t),r'(s,a,t)} \tilde{V}_t(s) = \max_{P'(s,a,t),r'(s,a,t),\pi'(s,t)} \left[ r'(s,\pi'(s,t),t) + P'(s,\pi'(s,t),t)^\top \tilde{V}_{t+1} \right]$$

$$\text{s.t.} \quad \forall a \in \mathcal{A}: r'(s,a,t) \in [0,1], \qquad P'(s,a,t) \in \Delta_S$$

$$|(P'(s,a,t) - \hat{P}_k(s,a,t))^\top V_{t+1}'| \leq \phi(s,a,t)(H-t)$$

$$|r'(s,a,t) - \hat{r}_k(s,a,t)| \leq \phi(s,a,t)$$

since in this problem either $P'(s,\pi'(s,t),t)^\top \tilde{V}_{t+1} = \hat{P}(s,\pi'(s,t),t)^\top \tilde{V}_{t+1} + \phi(s,a,t)(H-t)$ if that does not violate $P'(s,\pi'(s,t),t)^\top \tilde{V}_{t+1} \leq \max \tilde{V}_{t+1}$ and otherwise $P'(s',s,\pi'(s,t),t) = 1$

for one state $s'$ with $\tilde{V}_{t+1}(s') = \max \tilde{V}_{t+1}$. Similarly, either $r'(s, \pi'(s,t), t) = \hat{r}(s, \pi'(s,t), t) + \phi(s, \pi'(s,t), t)$ if that does not violate $r'(s, \pi'(s,t), t) \leq 1$ or $r'(s, \pi'(s,t), t) = 1$ otherwise. Using induction for $t = H, H-1 \ldots 1$, we see that UBEV computes an optimal solution to

$$\max_{P',V',\pi',r'} V_1'(\tilde{s})$$

$$\forall s \in \mathcal{S}, a \in \mathcal{A}, t \in [H]:$$
$$V_{H+1}' = 0, \qquad P'(s,a,t) \in \Delta_S, \qquad r'(s,a,t) \in [0,1]$$
$$V_t'(s) = r'(s, \pi'(s,t), t) + \mathbb{E}_{s' \sim P'(s,\pi'(s,t),t)}[V_{t+1}']$$
$$|(P'(s,a,t) - \hat{P}_k(s,a,t))^\top V_{t+1}'| \leq \phi(s,a,t)(H-t)$$
$$|r'(s,a,t) - \hat{r}_k(s,a,t)| \leq \phi(s,a,t)$$

for any fixed $\tilde{s}$. The intersection of all optimal solutions to this problem for all $\tilde{s} \in \mathcal{S}$ are also an optimal solution to

$$\max_{P',V',\pi',r'} p_0^\top V_1'$$

$$\forall s \in \mathcal{S}, a \in \mathcal{A}, t \in [H]:$$
$$V_{H+1}' = 0, \qquad P'(s,a,t) \in \Delta_S, \qquad r'(s,a,t) \in [0,1]$$
$$V_t'(s) = r'(s, \pi'(s,t), t) + \mathbb{E}_{s' \sim P'(s,\pi'(s,t),t)}[V_{t+1}']$$
$$|(P'(s,a,t) - \hat{P}_k(s,a,t))^\top V_{t+1}'| \leq \phi(s,a,t)(H-t)$$
$$|r'(s,a,t) - \hat{r}_k(s,a,t)| \leq \phi(s,a,t).$$

Hence, UBEV computes an optimal solution to this problem. $\qquad \square$

# E  Details of PAC Analysis

In the analysis, we denote the value of $n(\cdot, t)$ after the planning in iteration $k$ as $n_{tk}(\cdot)$. We further denote by $P(s'|s,a,t)$ the probability of sampling state $s'$ as $s_{t+1}$ when $s_t = s, a_t = a$. With slight abuse of notation, $P(s,a,t) \in [0,1]^S$ denotes the probability vector of $P(\cdot|s,a,t)$. We further use $\tilde{P}_k(s'|s,a,t)$ as conditional probability of $s_{t+1} = s'$ given $s_t = s, a_t = a$ but in the optimistic MDP $\tilde{M}$ computed in the optimistic planning steps in iteration $k$. We also use the following definitions:

$$w_{\min} = w'_{\min} = \frac{\varepsilon c_\varepsilon}{H^2 S}$$
$$c_\varepsilon = \frac{1}{3}$$
$$L_{tk} = \{(s,a) \in \mathcal{S} \times \mathcal{A} : w_{tk}(s,a) \geq w_{\min}\}$$
$$\text{llnp}(x) = \ln(\ln(\max\{x, e\}))$$
$$\text{rng}(x) = \max(x) - \min(x)$$
$$\delta' = \frac{\delta}{9}$$

In the following, we provide the formal proof for Theorem 4 and then present all necessary lemmas:

## E.1  Proof of Theorem 4

*Proof of Theorem 4.* Corollary E.5 ensures that the failure event has probability at most $\delta$. Outside the failure event Lemma E.2 ensures that all but at most $\frac{48A^2 S^3 H^4}{\varepsilon}$ polylog$(A, S, H, 1/\varepsilon, 1/\delta)$ episodes are friendly. Finally, Lemma E.8 shows that all friendly episodes except at most $\left(\frac{9216}{\varepsilon} + 417S\right) \frac{ASH^4}{\varepsilon}$ polylog$(A, S, H, 1/\varepsilon, 1/\delta)$ are $\varepsilon$-optimal. The second bound follows from replacing $AS^2$ by $1/\varepsilon$ in the second term. Furthermore, outside the failure event Lemma E.2 ensures that all but at most $\frac{6AS^2 H^3}{\varepsilon}$ polylog$(A, S, H, 1/\varepsilon, 1/\delta)$ episodes are nice. Finally, Lemma E.7 shows that all nice episodes except at most $(4+S) 576 \frac{ASH^4}{\varepsilon}$ polylog$(A, S, H, 1/\varepsilon, 1/\delta)$ are $\varepsilon$-optimal.

$\qquad \square$

## E.2 Failure Events and Their Probabilities

In this section, we define a failure event $F$ in which we cannot guarantee the performance of UBEV. We then show that this event $F$ only occurs with low probability. All our arguments are based on general uniform concentration of measure statements that we prove in Section F. In the following we argue how the apply in our setting and finally combine all concentration results to get $\mathbb{P}(F) \leq \delta$. The failure event is defined as

$$F = \bigcup_k \left[ F_k^N \cup F_k^{CN} \cup F_k^P \cup F_k^V \cup F_k^{L1} \cup F_k^R \right]$$

where

$$F_k^N = \left\{ \exists s, a, t : n_{tk}(s,a) < \frac{1}{2} \sum_{i<k} w_{ti}(s,a) - \ln \frac{SAH}{\delta'} \right\}$$

$$F_k^{CN} = \left\{ \exists s, a, s', a', u < t : n_{tk}(s,a) < \frac{1}{2} n_{uk}(s',a') \sum_{i<k} w_{ui}^t(s,a|s',a') - \ln \left( \frac{S^2 A^2 H^2}{\delta'} \right) \right\}$$

$$F_k^V = \left\{ \exists s, a, t : |(\hat{P}_k(s,a,t) - P(s,a,t))^\top V_{t+1}^\star| \geq \sqrt{\frac{\mathrm{rng}(V_{t+1}^\star)^2}{n_{tk}(s,a)} \left( 2\,\mathrm{llnp}(n_{tk}(s,a)) + \ln \frac{3SAH}{\delta'} \right)} \right\}$$

$$F_k^P = \left\{ \exists s, s', a, t : |\hat{P}_k(s'|s,a,t) - P(s'|s,a,t)| \geq \sqrt{\frac{2P(s'|s,a,t)}{n_{tk}(s,a)} \left( 2\,\mathrm{llnp}(n_{tk}(s,a)) + \ln \frac{3S^2 AH}{\delta'} \right)} \right.$$

$$\left. + \frac{1}{n_{tk}(s,a)} \left( 2\,\mathrm{llnp}(n_{tk}(s,a)) + \ln \frac{3S^2 AH}{\delta'} \right) \right\}$$

$$F_k^{L1} = \left\{ \exists s, a, t : \|\hat{P}_k(s,a,t) - P(s,a,t)\|_1 \geq \sqrt{\frac{4}{n_{tk}(s,a)} \left( 2\,\mathrm{llnp}(n_{tk}(s,a)) + \ln \frac{3SAH(2^S - 2)}{\delta'} \right)} \right\}$$

$$F_k^R = \left\{ \exists s, a, t : |\hat{r}_k(s,a,t) - r(s,a,t)| \geq \sqrt{\frac{1}{n_{tk}(s,a)} \left( 2\,\mathrm{llnp}(n_{tk}(s,a)) + \ln \frac{3SAH}{\delta'} \right)} \right\}.$$

We now bound the probability of each type of failure event individually:

**Corollary E.1.** *For any $\delta' > 0$, it holds that $\mathbb{P}\left(\bigcup_{k=1}^\infty F_k^V\right) \leq 2\delta'$ and $\mathbb{P}\left(\bigcup_{k=1}^\infty F_k^R\right) \leq 2\delta'$*

*Proof.* Consider a fix $s \in \mathcal{S}, a \in \mathcal{A}, t \in [H]$ and denote $\mathcal{F}_k$ the sigma-field induced by the first $k-1$ episodes and the $k$-th episode up to $s_t$ and $a_t$ but not $s_{t+1}$. Define $\tau_i$ to be the index of the episode where $(s,a)$ was observed at time $t$ the $i$th time. Note that $\tau_i$ are stopping times with respect to $\mathcal{F}_i$. Define now the filtration $\mathcal{G}_i = \mathcal{F}_{\tau_i} = \{A \in \mathcal{F}_\infty : A \cap \{\tau_i \leq t\} \in \mathcal{F}_t \ \forall t \geq 0\}$ and $X_k = (V_{t+1}^\star(s_k') - P(s,a,t)^\top V_{t+1}^\star)\mathbb{I}\{\tau_k < \infty\}$ where $s_i'$ is the value of $s_{t+1}$ in episode $\tau_i$ (or arbitrary, if $\tau_i = \infty$).

By the Markov property of the MDP, we have that $X_i$ is a martingale difference sequence with respect to the filtration $\mathcal{G}_i$. Further, since $\mathbb{E}[X_i|\mathcal{G}_{i-1}] = 0$ and $|X_i| \in [0, \mathrm{rng}(V_{t+1}^\star)]$, $X_i$ conditionally $\mathrm{rng}(V_{t+1}^\star)/2$-subgaussian due to Hoeffding's Lemma, i.e., satisfies $\mathbb{E}[\exp(\lambda X_i)|\mathcal{G}_{i-1}] \leq \exp(\lambda^2 \mathrm{rng}(V_{t+1}^\star)^2/2)$.

We can therefore apply Lemma F.1 and conclude that

$$\mathbb{P}\left( \exists k : |(\hat{P}_k(s,a,t) - P(s,a,t))^\top V_{t+1}^\star| \geq \sqrt{\frac{\mathrm{rng}(V_{t+1}^\star)^2}{n_{tk}(s,a)} \left( 2\,\mathrm{llnp}(n_{tk}(s,a)) + \ln \frac{3}{\delta'} \right)} \right) \leq 2\delta'.$$

Analogously

$$\mathbb{P}\left( \exists k : |\hat{r}_k(s,a,t) - r(s,a,t)| \geq \sqrt{\frac{1}{n_{tk}(s,a)} \left( 2\,\mathrm{llnp}(n_{tk}(s,a)) + \ln \frac{3}{\delta'} \right)} \right) \leq 2\delta'.$$

Applying the union bound over all $s \in \mathcal{S}, a \in \mathcal{A}$ and $t \in [H]$, we obtain the desired statement for $F^V$. In complete analogy using the same filtration, we can show the statement for $F^R$. $\qquad\square$

**Corollary E.2.** *For any $\delta' > 0$, it holds that $\mathbb{P}\left(\bigcup_{k=1}^{\infty} F_k^P\right) \leq 2\delta'$.*

*Proof.* Consider first a fix $s', s \in \mathcal{S}$, $t \in [H]$ and $a \in \mathcal{A}$. Let $K$ denote the number of times the triple $s, a, t$ was encountered in total during the run of the algorithm. Define the random sequence $X_i$ as follows. For $i \leq K$, let $X_i$ be the indicator of whether $s'$ was the next state when $s, a, t$ was encountered the $i$th time and for $i > K$, let $X_i \sim \mathrm{Bernoulli}(P(s'|s, a, t))$ be drawn i.i.d. By construction this is a sequence of i.i.d. Bernoulli random variables with mean $P(s'|s, a, t)$. Further the event

$$
\bigcup_k \left\{ \left| \hat{P}_k(s'|s, a, t) - P(s'|s, a, t) \right| \geq \sqrt{\frac{2P(s'|s, a, t)}{n_{tk}(s, a)} \left( 2\,\mathrm{llnp}(n(s, a, t)) + \ln \frac{3S^2 AH}{\delta'} \right)} \right.
$$
$$
\left. + \frac{1}{n_{tk}(s, a)} \left( 2\,\mathrm{llnp}(n_{tk}(s, a)) + \ln \frac{3S^2 AH}{\delta'} \right) \right\}
$$

is contained in the event

$$
\bigcup_i \left\{ |\hat{\mu}_i - \mu| \geq \sqrt{\frac{2\mu}{i} \left( 2\,\mathrm{llnp}(i) + \ln \frac{3}{\delta'} \right)} + \frac{1}{i} \left( 2\,\mathrm{llnp}(i) + \ln \frac{3S^2 AH}{\delta'} \right) \right\}
$$

whose probability can be bounded by $2\delta'/S^2/A/H$ using Lemma F.2. The statement now follows by applying the union bound. $\qquad\square$

**Corollary E.3.** *For any $\delta' > 0$, it holds that $\mathbb{P}\left(\bigcup_{k=1}^{\infty} F_k^{L1}\right) \leq \delta'$*

*Proof.* Using the same argument as in the proof of Corollary E.2 the statement follows from Lemma F.3. $\qquad\square$

**Corollary E.4.** *It holds that*

$$
\mathbb{P}\left(\bigcup_k F_k^N\right) \leq \delta' \quad \text{and} \quad \mathbb{P}\left(\bigcup_k F_k^{CN}\right) \leq \delta'.
$$

*Proof.* Consider a fix $s \in \mathcal{S}, a \in \mathcal{A}, t \in [H]$. We define $\mathcal{F}_k$ to be the sigma-field induced by the first $k - 1$ episodes and $X_k$ as the indicator whether $s, a, t$ was observed in episode $k$. The probability $w_{tk}(s, a)$ pf whether $X_k = 1$ is $F_k$ measurable and hence we can apply Lemma F.4 with $W = \ln \frac{SAH}{\delta'}$ and obtain that $\mathbb{P}\left(\bigcup_k F_k^N\right) \leq \delta'$ after applying the union bound.

For the second statement, consider again a fix $s, s' \in \mathcal{S}, a, a' \in \mathcal{A}, u, t \in [H]$ with $u < t$ and denote by $\mathcal{F}_k$ the sigma-field induced by the first $k - 1$ episodes and the $k$-th episode up to $s_u$ and $a_u$ but not $s_{u+1}$. Define $\tau_i$ to be the index of the episode where $(s', a')$ was observed at time $u$ the $i$th time. Note that $\tau_i$ are stopping times with respect to $\mathcal{F}_i$. Define now the filtration $\mathcal{G}_i = \mathcal{F}_{\tau_i} = \{A \in \mathcal{F}_\infty : A \cap \{\tau_i \leq k\} \in \mathcal{F}_k \,\forall k \geq 0\}$ and $X_i$ to be the indicator whether $s, a, t$ and $s', a', u$ was observed in episode $\tau_i$. If $\tau_i = \infty$, we set $X_i = 0$. Note that the probablity $w_{ui}^t(s, a|s', a')\mathbb{I}\{\tau_i < \infty\}$ of $X_i = 1$ is $\mathcal{G}_i$-measureable.

By the Markov property of the MDP, we have that $X_i$ is a martingale difference sequence with respect to the filtration $\mathcal{G}_i$. We can therefore apply Lemma F.4 with $W = \ln \frac{S^2 A^2 H^2}{\delta'}$ and using the union bound over all $s, a, s', a', u, t$, we get $\mathbb{P}\left(\bigcup_k F_k^{CN}\right) \leq \delta'$. $\qquad\square$

**Corollary E.5.** *The total failure probability of the algorithm is bounded by $\mathbb{P}(F) \leq 9\delta' = \delta$.*

*Proof.* Statement follows directly from Corollary E.1, Corollary E.2, Corollary E.3, Corollary E.4 and the union bound. $\qquad\square$

### E.3 Nice and Friendly Episodes

We now define the notion of *nice* and the stronger *friendly* episodes. In nice episodes, all states either have low probability of occuring or the sum of probability of occuring in the previous episodes is large enough so that outside the failure event we can guarantee that

$$n_{tk}(s,a) \geq \frac{1}{4}\sum_{i<k} w_{ti}(s,a).$$

This allows us to then bound the number of nice episodes by the number of times terms of the form

$$\sum_{t=1}^{H}\sum_{s,a\in L_{tk}} w_{tk}(s,a)\sqrt{\frac{\mathrm{llnp}(n_{tk}(s,a)) + D}{n_{tk}(s,a)}}$$

can exceed a chosen threshold (see Lemma E.3 below). In the next section, we will bound the optimality gap of an episode by terms of such form and use the results derived here to bound the number of nice episodes where the algorithm can follow a $\varepsilon$-suboptimal policy. Together with a bound on the number of non-nice episodes, we obtain the sample complexity of UBEV shown in Theorem 4.

Similarly, we use a more refined analysis of the optimality gap of friendly episodes together with Lemma E.4 below to obtain the tighter sample complexity linear-polylog in $S$.

**Definition 2** (Nice and Friendly Episodes). *An episode $k$ is* nice *if and only if for all $s \in \mathcal{S}$, $a \in \mathcal{A}$ and $t \in [H]$ the following two conditions hold:*

$$w_{tk}(s,a) \leq w_{\min} \quad \vee \quad \frac{1}{4}\sum_{i<k} w_{ti}(s,a) \geq \ln\frac{SAH}{\delta'}$$

*An episode $k$ is* friendly *if and only if it is nice and for all $s, s' \in \mathcal{S}$, $a, a' \in \mathcal{A}$ and $u, t \in [H]$ with $u < t$ the following two conditions hold:*

$$w_{uk}^{t}(s,a|s',a') \leq w'_{\min} \quad \vee \quad \frac{1}{4}\sum_{i<k} w_{ui}^{t}(s,a|s',a') \geq \ln\frac{S^2 A^2 H^2}{\delta'}.$$

*We denote the set of all nice episodes by $N \subseteq \mathbb{N}$ and the set of all friendly episodes by $K \subseteq N$.*

**Lemma E.1** (Properties of nice and friendly episodes). *If an episode $k$ is nice, i.e., $k \in N$, then on $F^c$ (outside the failure event) for all $s \in \mathcal{S}$, $a \in \mathcal{A}$ and $t \in [H]$ with $u < t$ the following statement holds:*

$$w_{tk}(s,a) \leq w_{\min} \quad \vee \quad n_{tk}(s,a) \geq \frac{1}{4}\sum_{i<k} w_{ti}(s,a).$$

*If an episode $k$ is friendly, i.e., $k \in K$, then on $F^c$ (outside the failure event) for all $s, s' \in \mathcal{S}$, $a, a' \in \mathcal{A}$ and $u, t \in [H]$ with $u < t$ the above statement holds as well as*

$$w_{uk}^{t}(s,a|s',a') \leq w'_{\min} \quad \vee \quad n_{tk}(s,a) \geq \frac{1}{4}n_{uk}(s',a')\sum_{i<k} w_{ui}^{t}(s,a|s',a').$$

*Proof.* Since we consider the event $F_k^{N^c}$, it holds for all $s, a, t$ triples with $w_{tk}(s,a) > w_{\min}$

$$n_{tk}(s,a) \geq \frac{1}{2}\sum_{i<k} w_{ti}(s,a) - \ln\frac{SAH}{\delta'} \geq \frac{1}{4}\sum_{i<k} w_{ti}(s,a)$$

for $k \in N$ Further, since we only consider the event $F_k^{CN^c}$, we have for all $s, s' \in \mathcal{S}$, $a, a' \in \mathcal{A}$, $u, t \in [H]$ with $u < t$ and $w_{uk}^{t}(s,a|s',a') > w_{\min}$

$$n_{tk}(s,a) \geq \frac{1}{2}n_{uk}(s',a')\sum_{i<k} w_{ui}^{t}(s,a|s',a') - \ln\frac{S^2 A^2 H^2}{\delta'}$$

for $k \in E$. If $n_{uk}(s', a') = 0$ then $n_{tk}(s, a) \geq 0 = \frac{1}{4} n_{uk}(s', a') \sum_{i<k} w_{ui}^t(s, a|s', a')$ holds trivially. Otherwise $n_{uk}(s', a') \geq 1$ and therefore

$$
\begin{aligned}
n_{tk}(s, a) &\geq \frac{1}{2} n_{uk}(s', a') \sum_{i<k} w_{ui}^t(s, a|s', a') - \ln \frac{S^2 A^2 H^2}{\delta'} \\
&\geq \frac{1}{2} n_{uk}(s', a') \sum_{i<k} w_{ui}^t(s, a|s', a') - \frac{1}{4} \sum_{i<k} w_{ui}^t(s, a|s', a') \\
&\geq \frac{1}{4} n_{uk}(s', a') \sum_{i<k} w_{ui}^t(s, a|s', a')
\end{aligned}
$$

$\square$

**Lemma E.2** (Number of non-nice and non-friendly episodes)**.** *On the good event $F^c$, the number of episodes that are not friendly is at most*

$$
48 \frac{S^3 A^2 H^4}{\varepsilon} \ln \frac{S^2 A^2 H^2}{\delta'}
$$

*and the number episodes that are not nice is at most*

$$
\frac{6 S^2 A H^3}{\varepsilon} \ln \frac{S A H}{\delta'}.
$$

*Proof.* If an episode $k$ is not nice, then there is $s, a, t$ with $w_{tk}(s, a) > w_{\min}$ and $\sum_{i<k} w_{ti}(s, a) < 4 \ln \frac{SAH}{\delta'}$. Since the sum on the left-hand side of this inequality increases by at least $w_{\min}$ when this happens and the right hand side stays constant, this situation can occur at most

$$
\frac{4 S A H}{w_{\min}} \ln \frac{S A H}{\delta'} = \frac{24 S^2 A H^3}{\varepsilon} \ln \frac{S A H}{\delta'}
$$

times in total. If an episode $k$ is not friendly, it is either not nice or there is $s, a, t$ and $s', a', u$ with $u < t$ and $w_{uk}^t(s', a'|s, a) > w'_{\min}$ and $\sum_{i<k} w_{ui}^t(s, a|s', a') < 4 \ln \frac{S^2 A^2 H^2}{\delta'}$. Since the sum on the left-hand side of this inequality increases by at least $w'_{\min}$ each time this happens while the right hand side stays constant, this can happen at most $\frac{4 S^2 A^2 H^2}{w'_{\min}} \ln \frac{S^2 A^2 H^2}{\delta'}$ times in total. Therefore, there can only be at most

$$
\begin{aligned}
&\frac{4 S A H}{w_{\min}} \ln \frac{S A H}{\delta'} + \frac{4 S^2 A^2 H^2}{w'_{\min}} \ln \frac{S^2 A^2 H^2}{\delta'} \\
= &\frac{4 S^2 A H^3}{c_\varepsilon \varepsilon} \ln \frac{S A H}{\delta'} + \frac{4 S^3 A^2 H^4}{c_\varepsilon \varepsilon} \ln \frac{S^2 A^2 H^2}{\delta'} \leq \frac{48 S^3 A^2 H^4}{\varepsilon^2} \ln \frac{S^2 A^2 H^2}{\delta'}
\end{aligned}
$$

non-friendly episodes. $\square$

**Lemma E.3** (Main Rate Lemma)**.** *Let $r \geq 1$ fix and $C > 0$ which can depend polynomially on the relevant quantities and $\varepsilon' > 0$ and let $D \geq 1$ which can depend poly-logarithmically on the relevant quantities. Then*

$$
\sum_t \sum_{s,a \in L_{tk}} w_{tk}(s, a) \left( \frac{C(\mathrm{llnp}(n_{tk}(s, a)) + D)}{n_{tk}(s, a)} \right)^{1/r} \leq \varepsilon'
$$

*on all but at most*

$$
\frac{8 C A S H^r}{\varepsilon'^r} \mathrm{polylog}(S, A, H, \delta^{-1}, \varepsilon'^{-1}).
$$

*nice episodes.*

*Proof.* Define

$$\Delta_k = \sum_t \sum_{s,a \in L_{tk}} w_{tk}(s,a) \left( \frac{C(\mathrm{llnp}(n_{tk}(s,a)) + D)}{n_{tk}(s,a)} \right)^{1/r}$$

$$= \sum_t \sum_{s,a \in L_{tk}} w_{tk}(s,a)^{1-\frac{1}{r}} \left( w_{tk}(s,a) \frac{C(\mathrm{llnp}(n_{tk}(s,a)) + D)}{n_{tk}(s,a)} \right)^{1/r}.$$

We first bound using Hölder's inequality

$$\Delta_k \le \left( \sum_t \sum_{s,a \in L_{tk}} \frac{CH^{r-1} w_{tk}(s,a)(\mathrm{llnp}(n_{tk}(s,a)) + D)}{n_{tk}(s,a)} \right)^{\frac{1}{r}}.$$

Using the property in Lemma E.1 of nice episodes as well as the fact that $w_{tk}(s,a) \le 1$ and $\sum_{i<k} w_{ti}(s,a) \ge 4 \ln \frac{SAH}{\delta'} \ge 4 \ln(2) \ge 2$, we bound

$$n_{tk}(s,a) \ge \frac{1}{4} \sum_{i<k} w_{ti}(s,a) \ge \frac{1}{8} \sum_{i \le k} w_{ti}(s,a).$$

The function $\frac{\mathrm{llnp}(x)+D}{x}$ is monotonically decreasing in $x \ge 0$ since $D \ge 1$ (see Lemma E.6). This allows us to bound

$$\Delta_k^r \le \sum_t \sum_{s,a \in L_{tk}} \frac{CH^{r-1} w_{tk}(s,a)(\mathrm{llnp}(n_{tk}(s,a)) + D)}{n_{tk}(s,a)}$$

$$\le 8CH^{r-1} \sum_t \sum_{s,a \in L_{tk}} \frac{w_{tk}(s,a) \left( \mathrm{llnp}\left( \frac{1}{8} \sum_{i \le k} w_{ti}(s,a) \right) + D \right)}{\sum_{i \le k} w_{ti}(s,a)}$$

$$\le 8CH^{r-1} \sum_t \sum_{s,a \in L_{tk}} \frac{w_{tk}(s,a) \left( \mathrm{llnp}\left( \sum_{i \le k} w_{ti}(s,a) \right) + D \right)}{\sum_{i \le k} w_{ti}(s,a)}.$$

Assume now $\Delta_k > \varepsilon'$. In this case the right-hand side of the inequality above is also larger than $\varepsilon'^r$ and there is at least one $(s,a,t)$ with $w_{tk}(s,a) > w_{\min}$ and

$$\frac{8CSAH^r \left( \mathrm{llnp}\left( \sum_{i \le k} w_{ti}(s,a) \right) + D \right)}{\sum_{i \le k} w_{ti}(s,a)} > \varepsilon'^r$$

$$\Leftrightarrow \quad \frac{\mathrm{llnp}\left( \sum_{i \le k} w_{ti}(s,a) \right) + D}{\sum_{i \le k} w_{ti}(s,a)} > \frac{\varepsilon'^r}{8CSAH^r}.$$

Let us denote $C' = \frac{8CASH^r}{\varepsilon'^r}$. Since $\frac{\mathrm{llnp}(x)+D}{x}$ is monotonically decreasing and $x = C'^2 + 3C'D$ satisfies $\frac{\mathrm{llnp}(x)+D}{x} \le \frac{\sqrt{x}+D}{x} \le \frac{1}{C'}$, we know that if $\sum_{i \le k} w_{ti}(s,a) \ge C'^2 + 3C'D$ then the above condition cannot be satisfied for $s,a,t$. Since each time the condition is satisfied, it holds that $w_{tk}(s,a) > w_{\min}$ and so $\sum_{i \le k} w_{ti}(s,a)$ increases by at least $w_{\min}$, it can happen at most

$$m \le \frac{ASH(C'^2 + 3C'D)}{w_{\min}}$$

times that $\Delta_k > \varepsilon'$. Define $K = \{k : \Delta_k > \varepsilon'\} \cap N$ and we know that $|K| \le m$. Now we consider the sum

$$\sum_{k \in K} \Delta_k^r \le \sum_{k \in K} 8CH^{r-1} \sum_t \sum_{s,a \in L_{tk}} \frac{w_{tk}(s,a) \left( \mathrm{llnp}\left( \sum_{i \le k} w_{ti}(s,a) \right) + D \right)}{\sum_{i \le k} w_{ti}(s,a)}$$

$$\le 8CH^{r-1} \left( \mathrm{llnp}\left( C'^2 + 3C'D \right) + D \right) \sum_t \sum_{s,a \in L_{tk}} \sum_{k \in K} \frac{w_{tk}(s,a)}{\sum_{i \le k} w_{ti}(s,a) \mathbb{I}\{w_{ti}(s,a) \ge w_{\min}\}}$$

For every $(s, a, t)$, we consider the sequence of $w_{ti}(s, a) \in [w_{\min}, 1]$ with $i \in I = \{i \in \mathbb{N} : w_{ti}(s, a) \geq w_{\min}\}$ and apply Lemma E.5. This yields that

$$\sum_{k \in K} \frac{w_{tk}(s, a)}{\sum_{i \leq k} w_{ti}(s, a) \mathbb{I}\{w_{ti}(s, a) \geq w_{\min}\}} \leq 1 + \ln(m/w_{\min}) = \ln\left(\frac{me}{w_{\min}}\right)$$

and hence

$$\sum_{k \in K} \Delta_k^r \leq 8CASH^r \ln\left(\frac{me}{w_{\min}}\right) \left(\text{llnp}\left(C'^2 + 3C'D\right) + D\right)$$

Since each element in $K$ has to contribute at least $\varepsilon'^r$ to this bound, we can conclude that

$$\sum_{k \in N} \mathbb{I}\{\Delta_k \geq \varepsilon'\} \leq \sum_{k \in K} \mathbb{I}\{\Delta_k \geq \varepsilon'\} \leq |K| \leq \frac{8CASH^r}{\varepsilon'^r} \ln\left(\frac{me}{w_{\min}}\right) \left(\text{llnp}\left(C'^2 + 3C'D\right) + D\right).$$

Since $\ln\left(\frac{me}{w_{\min}}\right) \left(\text{llnp}\left(C'^2 + 3C'D\right) + D\right)$ is $\text{polylog}(S, A, H, \delta^{-1}, \varepsilon'^{-1})$, the proof is complete. $\qquad \square$

**Lemma E.4** (Conditional Rate Lemma). *Let $r \geq 1$ fix and $C > 0$ which can depend polynomially on the relevant quantities and $\varepsilon' > 0$ and let $D \geq 1$ which can depend poly-logarithmically on the relevant quantities. Further $T \subset [H]$ is a subset of time-indices with $u < t$ for all $t \in T$. Then*

$$\sum_{t \in T} \sum_{s,a \in L_k^{ut}} w_{uk}^t(s, a|s', a') \left(\frac{C(\text{llnp}(n_{tk}(s, a)) + D)}{n_{tk}(s, a)}\right)^{1/r} \leq \varepsilon' \left(\frac{\text{llnp}(n_{uk}(s', a')) + D + 1}{n_{uk}(s', a')}\right)^{1/r}$$

*on all but at most*

$$\frac{8CAS|T|^r}{\varepsilon'^r} \text{polylog}(S, A, H, \delta^{-1}, \varepsilon'^{-1}).$$

*friendly episodes E.*

*Proof.* The proof follows mainly the structure of Lemma E.3. For the sake of completeness, we still present all steps here. Define

$$\Delta_k = \sum_{t \in T} \sum_{s,a \in L_k^{ut}} w_{uk}^t(s, a|s', a') \left(\frac{C(\text{llnp}(n_{tk}(s, a)) + D)}{n_{tk}(s, a)}\right)^{1/r}$$

$$= \sum_{t \in T} \sum_{s,a \in L_k^{ut}} w_{uk}^t(s, a|s', a')^{1-1/r} \left(w_{uk}^t(s, a|s', a') \frac{C(\text{llnp}(n_{tk}(s, a)) + D)}{n_{tk}(s, a)}\right)^{1/r}.$$

We first bound using Hölder's inequality

$$\Delta_k \leq \left(\sum_{t \geq u} \sum_{s,a \in L_k^{ut}} w_{uk}^t(s, a|s', a') \frac{C|T|^{r-1}(\text{llnp}(n_{tk}(s, a)) + D)}{n_{tk}(s, a)}\right)^{\frac{1}{r}}$$

Using the property in Lemma E.1 of friendly episodes as well as the fact that $w_{uk}^t(s, a|s', a') \leq 1$ and $\sum_{i < k} w_{ui}^t(s, a|s', a') \geq 4 \ln \frac{S^2 A^2 H^2}{\delta'} \geq 4 \ln(2) \geq 2$, we bound

$$n_{tk}(s, a) \geq \frac{1}{4} n_{uk}(s', a') \sum_{i < k} w_{ui}^t(s, a|s', a') \geq \frac{1}{8} n_{uk}(s', a') \sum_{i \leq k} w_{ui}^t(s, a|s', a').$$

The function $\frac{\text{llnp}(x)+D}{x}$ is monotonically decreasing in $x \geq 0$ since $D \geq 1$ (see Lemma E.6). This allows us to bound

$$\Delta_k^r \leq \sum_{t\in T}\sum_{s,a\in L_k^{ut}} w_{uk}^t(s,a|s',a')\frac{C|T|^{r-1}(\text{llnp}(n_{tk}(s,a))+D)}{n_{tk}(s,a)}$$

$$\leq 8C|T|^{r-1}\sum_{t\in T}\sum_{s,a\in L_k^{ut}} \frac{w_{uk}^t(s,a|s',a')(\text{llnp}\left(\frac{1}{8}n_{uk}(s',a')\sum_{i\leq k}w_{ui}^t(s,a|s',a')\right)+D)}{n_{uk}(s',a')\sum_{i\leq k}w_{ui}^t(s,a|s',a')}$$

$$\leq 8C|T|^{r-1}\sum_{t\in T}\sum_{s,a\in L_k^{ut}} \frac{w_{uk}^t(s,a|s',a')(\text{llnp}\left(\sum_{i\leq k}w_{ui}^t(s,a|s',a')\right)+\text{llnp}(n_{uk}(s',a'))+D+1)}{n_{uk}(s',a')\sum_{i\leq k}w_{ui}^t(s,a|s',a')},$$

where for the last line we used the first and last property in Lemma E.6. For notational convenience, we will use $D' = D + 1 + \text{llnp}(n_{uk}(s',a'))$. Assume now $\Delta_k > \varepsilon'\left(\frac{D'}{n_{uk}(s',a')}\right)^{1/r}$. In this case the right-hand side of the inequality above is also larger than $\varepsilon'^r\left(\frac{D'}{n_{uk}(s',a')}\right)$ and there is at least one $(s,a,t)$ with $w_{uk}^t(s,a|s',a') > w_{\min}$ and

$$\frac{8CSA|T|^r\left(\text{llnp}\left(\sum_{i\leq k}w_{ui}^t(s,a|s',a')\right)+D'\right)}{\sum_{i\leq k}w_{ui}^t(s,a|s',a')} > D'\varepsilon'^r$$

$$\Leftrightarrow \frac{\left(\text{llnp}\left(\sum_{i\leq k}w_{ui}^t(s,a|s',a')\right)+D'\right)}{\sum_{i\leq k}w_{ui}^t(s,a|s',a')} > \frac{D'\varepsilon'^r}{8CSA|T|^r}.$$

Let us denote $C' = \frac{8CAS|T|^r}{\varepsilon'^r}$. Since $\frac{\text{llnp}(x)+D'}{x}$ is monotonically decreasing and $x = C'^2 + 3C'$ satisfies $\frac{\text{llnp}(x)+D'}{x} \leq \frac{\sqrt{x}+D'}{x} \leq D'\frac{\sqrt{x}+1}{x} \leq \frac{D'}{C'}$, we know that if $\sum_{i\leq k}w_{ui}^t(s,a|s',a') \geq C'^2+3C'$ then the above condition cannot be satisfied for $s,a,t$. Since each time the condition is satisfied, it holds that $w_{uk}^t(s,a|s',a') > w_{\min}$ and so $\sum_{i\leq k}w_{ui}^t(s,a|s',a')$ increases by at least $w_{\min}$, it can happen at most

$$m \leq \frac{AS|T|(C'^2+3C')}{w_{\min}}$$

times that $\Delta_k > \varepsilon'\left(\frac{D'}{n_{uk}(s',a')}\right)^{1/r}$. Define $K = \left\{k : \Delta_k > \varepsilon'\left(\frac{D'}{n_{uk}(s',a')}\right)^{1/r}\right\} \cap E$ and we know that $|K| \leq m$. Now we consider the sum

$$\sum_{k\in K}\Delta_k^r \leq \sum_{k\in K}8C|T|^{r-1}\sum_{t\in T}\sum_{s,a\in L_k^{ut}} \frac{w_{uk}^t(s,a|s',a')(\text{llnp}\left(\sum_{i\leq k}w_{ui}^t(s,a|s',a')\right)+D')}{n_{uk}(s',a')\sum_{i\leq k}w_{ui}^t(s,a|s',a')}$$

$$\leq \frac{8C|T|^{r-1}(\text{llnp}\left(C'^2+3C'\right)+D')}{n_{uk}(s',a')}\sum_{t\in T}\sum_{s,a\in L_k^{ut}}\sum_{k\in K} \frac{w_{uk}^t(s,a|s',a')}{\sum_{i\leq k}w_{ui}^t(s,a|s',a')}$$

$$\leq \frac{8C|T|^{r-1}D'(\text{llnp}\left(C'^2+3C'\right)+1)}{n_{uk}(s',a')}\sum_{t\in T}\sum_{s,a\in L_k^{ut}}\sum_{k\in K} \frac{w_{uk}^t(s,a|s',a')}{\sum_{i\leq k}w_{ui}^t(s,a|s',a')\mathbb{I}\{w_{ui}^t(s,a|s',a') \geq w_{\min}\}}$$

For every $(s,a,t)$, we consider the sequence of $w_{ui}^t(s,a|s',a') \in [w_{\min},1]$ with $i \in I = \{i \in \mathbb{N} : w_{ui}^t(s,a|s',a') \geq w_{\min}\}$ and apply Lemma E.5. This yields that

$$\sum_{k\in K} \frac{w_{uk}^t(s,a|s',a')}{\sum_{i\leq k}w_{ui}^t(s,a|s',a')\mathbb{I}\{w_{ui}^t(s,a|s',a') \geq w_{\min}\}} \leq \ln\left(\frac{me}{w_{\min}}\right)$$

and hence

$$\sum_{k \in K} \Delta_k^r \le \frac{8CAS|T|^r D' \left(\text{llnp}\left(C'^2 + 3C'\right) + 1\right)}{n_{uk}(s', a')} \ln\left(\frac{me}{w_{\min}}\right)$$

Since each element in $K$ has to contribute at least $\frac{D' \varepsilon'^r}{n_{uk}(s', a')}$ to this bound, we can conclude that

$$\sum_{k \in E} \mathbb{I}\{\Delta_k \ge \varepsilon'\} = \sum_{k \in K} \mathbb{I}\{\Delta_k \ge \varepsilon'\}$$

$$\le |K| \le \frac{8CAS|T|^r}{\varepsilon'^r} \ln\left(\frac{me}{w_{\min}}\right) \left(\text{llnp}\left(C'^2 + 3C'\right) + 1\right).$$

Since $\ln\left(\frac{me}{w_{\min}}\right)\left(\text{llnp}\left(C'^2 + 3C'\right) + 1\right)$ is $\text{polylog}(S, A, H, \delta^{-1}, \varepsilon'^{-1})$, the proof is complete. $\quad\square$

**Lemma E.5.** *Let $a_i$ be a sequence taking values in $[a_{\min}, 1]$ with $a_{\min} > 0$ and $m > 0$, then*

$$\sum_{k=1}^{m} \frac{a_k}{\sum_{i=1}^{k} a_i} \le \ln\left(\frac{me}{a_{\min}}\right).$$

*Proof.* Let $f$ be a step-function taking value $a_i$ on $[i-1, i)$ for all $i$. We have $F(t) := \int_0^t f(x)dx = \sum_{i=1}^{t} a_i$. By the fundamental theorem of Calculus, we can bound

$$\sum_{k=1}^{m} \frac{a_k}{\sum_{i=1}^{k} a_i} = \frac{a_1}{a_1} + \int_1^m \frac{f(x)}{F(x) - F(0)} dx = 1 + \ln F(m) - \ln F(1)$$

$$\le 1 + \ln(m) - \ln a_{\min} = \ln\left(\frac{me}{a_{\min}}\right),$$

where the inequality follows from $a_1 \ge a_{\min}$ and $\sum_{i=1}^{m} a_i \le m$. $\quad\square$

**Lemma E.6** (Properties of llnp). *The following properties hold:*

1. *llnp is continuous and nondecreasing.*

2. *$f(x) = \frac{\text{llnp}(nx) + D}{x}$ with $n \ge 0$ and $D \ge 1$ is monotonically decreasing on $\mathbb{R}_+$.*

3. *$\text{llnp}(xy) \le \text{llnp}(x) + \text{llnp}(y) + 1$ for all $x, y \ge 0$.*

*Proof.*   1. For $x \le e$ we have $\text{llnp}(x) = 0$ and for $x \ge e$ we have $\text{llnp}(x) = \ln(\ln(x))$ which is continuous and monotonically increasing and $\lim_{x \searrow e} \ln(\ln(x)) = 0$.

2. The function llnp is continuous as well as $1/x$ on $\mathbb{R}_+$ and therefore so it $f$. Further, $f$ is differentiable except at $x = e/n$. For $x \in [0, e/n)$, we have $f(x) = D/x$ with derivative $-D/x^2 < 0$. Hence $f$ is monotonically decreasing on $x \in [0, e/n)$. For $x > e/n$, we have $f(x) = \frac{\ln(\ln(nx)) + D}{x}$ with derivative

$$-\frac{D + \ln(\ln(nx))}{x^2} + \frac{1}{x^2 \ln(nx)} = \frac{1 - \ln(nx)(D + \ln(\ln(nx)))}{x^2 \ln(nx)}.$$

The denominator is always positive in this range so $f$ is monotonically decreasing if and only if $\ln(nx)(D - \ln(\ln(nx))) \ge 1$. Using $D \ge 1$, we have $\ln(nx)(D + \ln(\ln(nx))) \ge 1(1 + 0) = 1$.

3. First note that for $xy \le e^e$ we have $\text{llnp}(xy) \le 1 \le \text{llnp}(x) + \text{llnp}(y) + 1$ and therfore the statement holds for $x, y \le e$.

   Then consider the case that $x, y \ge e$ and $\text{llnp}(x) + \text{llnp}(y) + 1 - \text{llnp}(xy) = \ln\ln x + \ln\ln y + 1 - \ln(\ln(x) + \ln(y)) = -\ln(a + b) + 1 + \ln(a) + \ln(b)$ where $a = \ln x \ge 1$ and $b = \ln y \ge 1$. The function $g(a, b) = -\ln(a + b) + 1 + \ln(a) + \ln(b)$ is continuous and

differentiable with $\frac{\partial g}{\partial a} = \frac{b}{a(a+b)} > 0$ and $\frac{\partial g}{\partial b} = \frac{a}{b(a+b)} > 0$. Therefore, $g$ attains its minimum on $[1, \infty) \times [1, \infty)$ at $a = 1, b = 1$. Since $g(1,1) = 1 - \ln(2) \geq 0$, the statement also holds for $x, y \geq e$.

Finally consider the case where $x \leq e \leq y$. Then $\mathrm{llnp}(xy) \leq \mathrm{llnp}(ey) = \ln(1 + \ln y) \leq \ln \ln y + 1 \leq \mathrm{llnp}(x) + \mathrm{llnp}(y) + 1$. Due to symmetry this also holds for $y \leq e \leq x$.

$\square$

## E.4 Decomposition of Optimality Gap

In this section we decompose the optimality gap and then bound each term individually. Finally, both rate lemmas presented in the previous section are used to determine a bound on the number of nice / friendly episodes where the optimality gap can be larger than $\varepsilon$. The decomposition in the following lemma is a the simpler version bounding the number of $\varepsilon$-suboptimal nice episodes and eventually lead to the first bound in Theorem 4.

**Lemma E.7** (Optimality Gap Bound On Nice Episodes). *On the good event $F^c$ it holds that $V_1^\star(s_0) - V_1^{\pi_k}(s_0) \leq \varepsilon$ on all nice episodes $k \in N$ except at most*

$$\frac{144(4 + 3H^2 + 4SH^2)ASH^2}{\varepsilon^2} \, \mathrm{polylog}(A, S, H, 1/\varepsilon, 1/\delta)$$

*episodes.*

*Proof.* Using optimism of the algorithm shown in Lemma E.16, we can bound

$$V_1^\star(s_0) - V_1^{\pi_k}(s_0)$$

$$\leq |\tilde{V}_1^{\pi_k}(s_0) - V_1^{\pi_k}(s_0)|$$

$$\leq \sum_{t=1}^{H} \sum_{s,a} w_{tk}(s,a) |(\tilde{P}_k(s,a,t) - P(s,a,t))^\top \tilde{V}_{t+1}^{\pi_k}| + \sum_{t=1}^{H} \sum_{s,a} w_{tk}(s,a) |\tilde{r}_k(s,a,t) - r(s,a,t)|$$

$$\leq \sum_{t=1}^{H} \sum_{s,a \in L_{tk}} w_{tk}(s,a) |(\tilde{P}_k(s,a,t) - P(s,a,t))^\top \tilde{V}_{t+1}^{\pi_k}| + \sum_{t=1}^{H} \sum_{s,a \in L_{tk}} w_{tk}(s,a) |\tilde{r}_k(s,a,t) - r(s,a,t)|$$

$$+ \sum_{t=1}^{H} \sum_{s,a \notin L_{tk}} w_{tk}(s,a) |(\tilde{P}_k(s,a,t) - P(s,a,t))^\top \tilde{V}_{t+1}^{\pi_k}| + \sum_{t=1}^{H} \sum_{s,a \notin L_{tk}} w_{tk}(s,a) |\tilde{r}_k(s,a,t) - r(s,a,t)|$$

$$\leq \sum_{t=1}^{H} \sum_{s,a \notin L_{tk}} H w_{\min} + \sum_{t=1}^{H} \sum_{s,a \in L_{tk}} w_{tk}(s,a) \left[ |(\tilde{P}_k(s,a,t) - \hat{P}_k(s,a,t))^\top \tilde{V}_{t+1}^{\pi_k}| \right.$$

$$\left. + |(\hat{P}_k(s,a,t) - P(s,a,t))^\top \tilde{V}_{t+1}^{\pi_k}| + |\tilde{r}_k(s,a,t) - r(s,a,t)| \right] \tag{9}$$

The first term is bounded by $c_\varepsilon \varepsilon = \frac{\varepsilon}{3}$. We now can use Lemma E.9, Lemma E.10 to bound the other terms by

$$\sum_{t=1}^{H} \sum_{s,a \in L_{tk}} w_{tk}(s,a) \sqrt{\frac{8(H + H\sqrt{S} + 2)^2}{n_{tk}(s,a)} \left( \mathrm{llnp}(n_{tk}(s,a)) + \frac{1}{2} \ln \frac{6SAH'}{\delta} \right)}.$$

We can then apply Lemma E.3 with $r = 2$, $C = 8(H + H\sqrt{S} + 2)^2$, $D = \frac{1}{2} \ln \frac{6SAH}{\delta'}$ ($\geq 1$ for any nontrivial setting) and $\varepsilon' = 2\varepsilon/3$ to bound this term by $\frac{2\varepsilon}{3}$ on all nice episodes except at most

$$\frac{64(H + \sqrt{S}H + 2)^2 ASH^2 3^2}{4\varepsilon^2} \, \mathrm{polylog}(A, S, H, 1/\varepsilon, 1/\delta)$$

$$\leq \frac{144(4 + 3H^2 + 4SH^2)ASH^2}{\varepsilon^2} \, \mathrm{polylog}(A, S, H, 1/\varepsilon, 1/\delta)$$

Hence $V_1^\star(s_0) - V_1^{\pi_k}(s_0) \leq \varepsilon$ holds on all nice episodes except those. $\square$

The lemma below is a refined version of the bound above and uses the stronger concept of friendly episodes to eventually lead to the second bound in Theorem 4.

**Lemma E.8** (Optimality Gap Bound On Friendly Episodes). *On the good event $F^c$ it holds that $p_0^\top (V_1^\star - V_1^{\pi_k}) \le \varepsilon$ on all friendly episodes $E$ except at most*

$$\left(\frac{9216}{\varepsilon} + 417S\right) \frac{ASH^4}{\varepsilon} \operatorname{polylog}(S, A, H, 1/\varepsilon, \delta)$$

*episodes if $\delta' \le \frac{3AS^2H}{e^2}$.*

*Proof.* We can further decompose the optimality gap bound in Equation (9) in the proof of Lemma E.7 as

$$\sum_{t=1}^{H} \sum_{s,a \notin L_{tk}} (H+1)w_{\min} + \sum_{t=1}^{H} \sum_{s,a \in L_{tk}} w_{tk}(s,a) \Bigg[ |(\tilde{P}_k(s,a,t) - \hat{P}_k(s,a,t))^\top \tilde{V}_{t+1}^{\pi_k}| + |\tilde{r}_k(s,a,t) - r(s,a,t)|$$

$$+ |(\hat{P}_k(s,a,t) - P(s,a,t))^\top V_{t+1}^\star| + |(\hat{P}_k(s,a,t) - P(s,a,t))^\top (V_{t+1}^\star - \tilde{V}_{t+1}^{\pi_k})| \Bigg].$$

$$\le c_\varepsilon \varepsilon + \sum_{t=1}^{H} \sum_{s,a \in L_{tk}} w_{tk}(s,a) \Bigg[ |(\tilde{P}_k(s,a,t) - \hat{P}_k(s,a,t))^\top \tilde{V}_{t+1}^{\pi_k}| + |\tilde{r}_k(s,a,t) - r(s,a,t)|$$

$$+ \qquad |(\hat{P}_k(s,a,t) - P(s,a,t))^\top V_{t+1}^\star| \Bigg]$$

$$+ \sum_{t=1}^{H} \sum_{s,a \in L_{tk}} w_{tk}(s,a) |(\hat{P}_k(s,a,t) - P(s,a,t))^\top (V_{t+1}^\star - \tilde{V}_{t+1}^{\pi_k})|.$$

The second term can be bounded using Lemmas E.11, E.10 and E.9 by

$$\sum_{t=1}^{H} \sum_{s,a \in L_{tk}} w_{tk}(s,a) \sqrt{\frac{32(H+1)^2}{n_{tk}(s,a)} \left( \operatorname{llnp}(n_{tk}(s,a)) + \frac{1}{2} \ln \frac{6SAH}{\delta'} \right)}.$$

which we bound by $\varepsilon/3$ using Lemma E.3 with $r = 2$, $C = 32(H+1)^2$, $D = \frac{1}{2} \ln \frac{6SAH}{\delta'}$ and $\varepsilon' = \varepsilon/3$ on all friendly episodes except at most

$$\frac{8CASH^2}{\varepsilon'^2} \operatorname{polylog}(S, A, H, 1/\varepsilon, 1/\delta) \le \frac{9216ASH^4}{\operatorname{polylog}}(S, A, H, 1/\varepsilon, 1/\delta).$$

Finally, we apply Lemma E.12 bound to bound the last term in Equation E.4 by $\varepsilon/3$ on all friendly epsiodes but at most

$$\frac{417AS^2H^4}{\varepsilon} \operatorname{polylog}(S, A, H, 1/\delta, 1/\varepsilon).$$

It hence follows that $p_0^\top (V_1^\star - V_1^{\pi_k}) \le \varepsilon$ on all friendly episodes but at most

$$\left(\frac{9216ASH^4}{\varepsilon^2} + \frac{417AS^2H^4}{\varepsilon}\right) \operatorname{polylog}(S, A, H, 1/\delta, 1/\varepsilon).$$

$\square$

**Lemma E.9** (Algorithm Learns Fast Enough). *It holds for all $s \in \mathcal{S}, a \in \mathcal{A}$ and $t \in [H]$*

$$|(\hat{P}_k(s,a,t) - \tilde{P}_k(s,a,t))^\top \tilde{V}_{t+1}| \le \sqrt{\frac{2H^2}{n_{tk}(s,a)} \left( \operatorname{llnp}(n_{tk}(s,a)) + \frac{1}{2} \ln \frac{3SAH}{\delta'} \right)}.$$

*Proof.* Using the definition of the constraint in the planning step of the algorithm shown in Lemma D.1 we can bound

$$|(\hat{P}_k(s,a,t) - \tilde{P}_k(s,a,t))^\top \tilde{V}_{t+1}| \leq \sqrt{\frac{H^2}{n_{tk}(s,a)} \left( 2\operatorname{llnp}(n_{tk}(s,a)) + \ln \frac{3SAH}{\delta'} \right)}.$$

$$\leq \sqrt{\frac{2H^2}{n_{tk}(s,a)} \left( \operatorname{llnp}(n_{tk}(s,a)) + \frac{1}{2} \ln \frac{3SAH}{\delta'} \right)}.$$

$\square$

**Lemma E.10** (Basic Decompsition Bound). *On the good event $F^c$ it holds for all $s \in \mathcal{S}, a \in \mathcal{A}$ and $t \in [H]$*

$$|(\hat{P}_k(s,a,t) - P(s,a,t))^\top \tilde{V}_{t+1}| \leq \sqrt{\frac{8H^2S}{n_{tk}(s,a)} \left( \operatorname{llnp}(n_{tk}(s,a)) + \frac{1}{2} \ln \frac{6SAH}{\delta'} \right)}$$

$$|\tilde{r}_k(s,a,t) - r(s,a,t)| \leq \sqrt{\frac{4}{n_{tk}(s,a)} \left( \operatorname{llnp}(n_{tk}(s,a)) + \frac{1}{2} \ln \frac{3SAH}{\delta'} \right)}.$$

*Proof.* On the good event $(F_k^{L1})c$ we have using Hölder's inequality

$$|(\hat{P}_k(s,a,t) - P(s,a,t))^\top \tilde{V}_{t+1}| \leq \|\hat{P}_k(s,a,t) - P(s,a,t))\|_1 \|\tilde{V}_{t+1}\|_\infty$$

$$\leq H \sqrt{\frac{4}{n_{tk}(s,a)} \left( 2\operatorname{llnp}(n_{tk}(s,a)) + \ln \frac{3SAH(2^S - 2)}{\delta'} \right)}$$

$$\leq \sqrt{\frac{8H^2S}{n_{tk}(s,a)} \left( \operatorname{llnp}(n_{tk}(s,a)) + \frac{1}{2} \ln \frac{6SAH}{\delta'} \right)}.$$

Further, on $(F_k^R)^c$ we have

$$|\tilde{r}_k(s,a,t) - r(s,a,t)| \leq |\tilde{r}_k(s,a,t) - r(s,a,t)| + |\tilde{r}_k(s,a,t) - \hat{r}(s,a,t)|$$

$$\leq 2 \sqrt{\frac{1}{n_{tk}(s,a)} \left( 2\operatorname{llnp}(n_{tk}(s,a)) + \ln \frac{3SAH}{\delta'} \right)}$$

$\square$

**Lemma E.11** (Fixed V Term Confidence Bound). *On the good event $F^c$ it holds for all $s \in \mathcal{S}, a \in \mathcal{A}$ and $t \in [H]$*

$$|(\hat{P}_k(s,a,t) - P(s,a,t))^\top V_{t+1}^\star| \leq \sqrt{\frac{2H^2}{n_{tk}(s,a)} \left( \operatorname{llnp} n_{tk}(s,a) + \frac{1}{2} \ln \frac{3SAH}{\delta'} \right)}$$

*Proof.* Since we consider the event $(F_k^V)^c$, we can bound

$$|(\hat{P}_k(s,a,t) - P(s,a,t))^\top V_{t+1}^\star| \leq \sqrt{\frac{2H^2}{n_{tk}(s,a)} \left( \operatorname{llnp} n_{tk}(s,a) + \frac{1}{2} \ln \frac{3SAH}{\delta'} \right)}$$

$\square$

**Lemma E.12** (Lower Order Term). *Assume $\delta' \leq \frac{3AS^2H}{e^2}$. On the good event $F^c$ on all friendly episodes $k \in E$ except at most $\frac{417AS^2H^4}{\varepsilon} \operatorname{polylog}(S, A, H, 1/\delta, 1/\varepsilon)$. it holds that*

$$\sum_{t=1}^H \sum_{s,a \in L_{tk}} w_{tk}(s,a)|(\hat{P}_k(s,a,t) - P(s,a,t))^\top (\tilde{V}_{t+1}^{\pi_k} - V_{t+1}^\star)| \leq \frac{\varepsilon}{3}.$$

*Proof.*

$$\sum_{t=1}^{H} \sum_{s,a\in L_{tk}} w_{tk}(s,a)|(\hat{P}_k(s,a,t) - P(s,a,t))^{\top}(\tilde{V}_{t+1}^{\pi_k} - V_{t+1}^{\star})|$$

$$\leq \sum_{t=1}^{H} \sum_{s,a\in L_{tk}} w_{tk}(s,a) \sum_{s'} \sqrt{\frac{2P(s'|s,a,t)}{n_{tk}(s,a)}\left(2\,\mathrm{llnp}(n_{tk}(s,a)) + \ln\frac{3S^2AH}{\delta'}\right)}|\tilde{V}_{t+1}^{\pi_k}(s') - V_{t+1}^{\star}(s')|$$

$$+ \sum_{t=1}^{H} \sum_{s,a\in L_{tk}} w_{tk}(s,a) \sum_{s'} \frac{1}{n_{tk}(s,a)}\left(2\,\mathrm{llnp}(n_{tk}(s,a)) + \ln\frac{3S^2AH}{\delta'}\right)|\tilde{V}_{t+1}^{\pi_k}(s') - V_{t+1}^{\star}(s')|$$

$$\leq \sum_{t=1}^{H} \sum_{s,a\in L_{tk}} w_{tk}(s,a) \sum_{s'} \sqrt{\frac{2P(s'|s,a,t)}{n_{tk}(s,a)}\left(2\,\mathrm{llnp}(n_{tk}(s,a)) + \ln\frac{3S^2AH}{\delta'}\right)\left(\tilde{V}_{t+1}^{\pi_k}(s') - V_{t+1}^{\star}(s')\right)^2}$$

$$+ \sum_{t=1}^{H} \sum_{s,a\in L_{tk}} \frac{w_{tk}(s,a)HS}{n_{tk}(s,a)}\left(2\,\mathrm{llnp}(n_{tk}(s,a)) + \ln\frac{3S^2AH}{\delta'}\right)$$

$$\leq \sum_{t=1}^{H} \sum_{s,a\in L_{tk}} w_{tk}(s,a)\sqrt{\frac{2S}{n_{tk}(s,a)}\left(2\,\mathrm{llnp}(n_{tk}(s,a)) + \ln\frac{3S^2AH}{\delta'}\right)P(s,a,t)^{\top}\left(\tilde{V}_{t+1}^{\pi_k} - V_{t+1}^{\star}\right)^2}$$

$$+ \sum_{t=1}^{H} \sum_{s,a\in L_{tk}} \frac{w_{tk}(s,a)HS}{n_{tk}(s,a)}\left(2\,\mathrm{llnp}(n_{tk}(s,a)) + \ln\frac{3S^2AH}{\delta'}\right)$$

The first inequality follows since we only consider outcomes in the event $(F_k^P)^c$, the second from the fact that value function are in the range $[0, H]$ and the third is an application of the Cauchy-Schwarz inequality. Using of optimism of the algorithm (Lemma E.16), we now bound $P(s,a,t)^{\top}\left(\tilde{V}_{t+1}^{\pi_k} - V_{t+1}^{\star}\right)^2 \leq P(s,a,t)^{\top}\left(\tilde{V}_{t+1}^{\pi_k} - V_{t+1}^{\pi_k}\right)^2$ which we bound by $c_{\varepsilon}\varepsilon + \left(c_{\varepsilon}\varepsilon + \sqrt{\frac{C'^2}{n_{tk}(s,a)S}}\left(\mathrm{llnp}(n_{tk}(s,a) + \frac{1}{2}\ln\frac{3AS^2H\varepsilon^4}{\delta'})\right)\right)^2 \leq c_{\varepsilon}\varepsilon + (c_{\varepsilon}\varepsilon + \frac{C'}{\sqrt{S}}\sqrt{J(s,a,t)})^2$ using Lemma E.13. To keep the notation concise, we use here the shorthand $J(s,a,t) = \frac{1}{n_{tk}(s,a)}\left(\mathrm{llnp}(n_{tk}(s,a)) + \frac{1}{2}\ln\frac{3e^4S^2AH}{\delta'}\right)$. This bound holds on all friendly episodes except at most $\left(32ASH^2 + 48AS^2H^3 + AS^2H^4 + 16AS^2\right)\mathrm{polylog}(S, A, H, 1/\delta, 1/\varepsilon)$. Plugging this into the bound from above, we get the upper bound

$$\sum_{t=1}^{H} \sum_{s,a\in L_{tk}} w_{tk}(s,a)\sqrt{4SJ(s,a,t)\left(c_{\varepsilon}\varepsilon + (c_{\varepsilon}\varepsilon + C'\sqrt{J(s,a,t)/S})^2\right)} + \sum_{t=1}^{H} \sum_{s,a\in L_{tk}} 2w_{tk}(s,a)HSJ(s,a,t)$$

$$\leq \sum_{t=1}^{H} \sum_{s,a\in L_{tk}} w_{tk}(s,a)\sqrt{4SJ(s,a,t)c_{\varepsilon}\varepsilon} + \sum_{t=1}^{H} \sum_{s,a\in L_{tk}} w_{tk}(s,a)\sqrt{4SJ(s,a,t)(c_{\varepsilon}\varepsilon + C'\sqrt{J(s,a,t)/S})^2}$$

$$+ \sum_{t=1}^{H} \sum_{s,a\in L_{tk}} 2w_{tk}(s,a)HSJ(s,a,t)$$

$$= \sum_{t=1}^{H} \sum_{s,a\in L_{tk}} w_{tk}(s,a)\sqrt{4(c_{\varepsilon}\varepsilon + c_{\varepsilon}^2\varepsilon^2)SJ(s,a,t)} + \sum_{t=1}^{H} \sum_{s,a\in L_{tk}} 2w_{tk}(s,a)J(s,a,t)(C' + SH),$$

where we used $\sqrt{a+b} \leq \sqrt{a} + \sqrt{b}$. We now bound the first term using Lemma E.3 with $r = 2, \varepsilon' = \varepsilon/6, D = \frac{1}{2}\ln\frac{3e^4 S^2 AH}{\delta'}, C = 4(c_\varepsilon \varepsilon + c_\varepsilon^2 \varepsilon^2)S$ on all but $\frac{8CASH^2}{\varepsilon'^2}$ polylog$(\dots) = \frac{192 c_\varepsilon(1+c_\varepsilon \varepsilon)AS^2 H^2}{\varepsilon}$ polylog$(\dots)$ friendly episodes by $\varepsilon/6$.

Applying Lemma E.3 with $r = 1, \varepsilon' = \varepsilon/6, D = \frac{1}{2}\ln\frac{3e^4 S^2 AH}{\delta'}$ and $C = 2(C' + HS)$, we can bound the second term by $\varepsilon/6$ on all but $\frac{8CASH}{\varepsilon'}$ polylog$(\dots) = \frac{96AS(C'+HS)H^2}{\varepsilon}$ polylog$(\dots)$ friendly episodes. Hence, it holds

$$\sum_{t=1}^{H} \sum_{s,a \in L_{tk}} w_{tk}(s,a)|(\hat{P}_k(s,a,t) - P(s,a,t))^\top (\tilde{V}_{t+1}^{\pi_k} - V_{t+1}^\star)| \leq \frac{\varepsilon}{3}$$

on all friendly episodes except at most

$$\left( \frac{96AS(C'+HS)H^2}{\varepsilon} + \frac{192c_\varepsilon(1+c_\varepsilon\varepsilon)AS^2 H^2}{\varepsilon} \right.$$

$$\left. + 32ASH^2 + 48AS^2 H^3 + AS^2 H^4 + 16AS^2 \right) \text{polylog}(S,A,H,1/\delta,1/\varepsilon)$$

episodes. Since $C' = \text{polylog}(S,A,H,1/\delta,1/\varepsilon)$, this simplifies to

$$\left( \frac{96AS}{\varepsilon} + \frac{96AS^2 H^3}{\varepsilon} + \frac{64AS^2 H^2}{\varepsilon} + 64AS^2 H^2 \right.$$

$$\left. + 32ASH^2 + 48AS^2 H^3 + AS^2 H^4 + 16AS^2 \right) \text{polylog}(S,A,H,1/\delta,1/\varepsilon)$$

$$\leq ((64 + 32 + 48 + 1 + 16)AS^2 H^4 + \frac{96 + 96 + 64}{\varepsilon}AS^2 H^3) \text{polylog}(S,A,H,1/\delta,1/\varepsilon)$$

failure episodes in $E$. We can finally bound the failure episodes by

$$\frac{417AS^2 H^4}{\varepsilon} \text{polylog}(S,A,H,1/\delta,1/\varepsilon).$$

$\square$

**Lemma E.13.** *On the good event $F^c$ for any $s \in \mathcal{S}$, $a \in \mathcal{A}$ and $t \in [H]$ with $\delta' \leq \frac{3AS^2 H}{e^2}$ it holds*

$$P(s,a,t)^\top (\tilde{V}_{t+1}^{\pi_k} - V_{t+1}^{\pi_k})^2 \leq c_\varepsilon \varepsilon + \left( c_\varepsilon \varepsilon + \sqrt{\frac{1}{n_{tk}(s,a)S} \left( \text{llnp}(n_{tk}(s,a) + \frac{1}{2}\ln\frac{3AS^2 H\varepsilon^4}{\delta'} \right)} \right)^2$$

*where $C' = 1 + \sqrt{\frac{1}{2}\ln\frac{3e^2 S^2 AH}{\delta'}}$ on all friendly episodes except for at most*

$$\left( 32ASH^2 + 48AS^2 H^3 + AS^2 H^4 + 16AS^2 \right) \text{polylog}(S,A,H,1/\delta,1/\varepsilon)$$

*episodes.*

*Proof.* Define $L' = \{s' : w_{tk}^{t+1}(s',a'|s,a) > w'_{\min}\}$ and $J(s') = \frac{\text{llnp}\, n_{t+1k}(s',a') + \frac{1}{2}\ln\frac{3e^2 S^2 AH}{\delta'}}{n_{t+1k}(s',a')}$ where $a' = \pi_k(s',t+1)$ and $C' = 1 + \sqrt{\frac{1}{2}\ln\frac{3e^2 S^2 AH}{\delta'}}$. Using Lemma E.14, we bound

$$P(s,a,t)^\top (\tilde{V}_{t+1}^{\pi_k} - V_{t+1}^{\pi_k})^2 = \sum_{s'} P(s'|s,a,t)(\tilde{V}_{t+1}^{\pi_k}(s') - V_{t+1}^{\pi_k}(s'))^2$$

$$\leq Sw_{\min}H^2 + \sum_{s' \in L'} P(s'|s,a,t)\left( c_\varepsilon \varepsilon + C'\sqrt{J(s')} \right)^2$$

$$\leq c_\varepsilon \varepsilon + C'^2 \sum_{s' \in L'} P(s'|s,a,t)J(s') + c_\varepsilon^2 \varepsilon^2 + 2c_\varepsilon \varepsilon C' \sum_{s' \in L'} P(s'|s,a,t)\sqrt{J(s')}$$

on all friendly episodes except at most $\left(32 + 48SH + SH^2\right) ASH^2 \operatorname{polylog}(S, A, H, 1/\delta, 1/\varepsilon)$. Define now $L'' = \{(s', a') \ : \ s' \in L', a' = \pi_k(s', t+1)\}$. We apply Lemma E.4 with $|T| = \{t+1\}, C = 1, D = \frac{1}{2} \ln \frac{3e^2 S^2 AH}{\delta'} \geq 1, r = 1$ and $\varepsilon' = 1/S$ to

$$
\sum_{s' \in L'} P(s'|s, a, t) J(s') = \sum_{s', a' \in L''} \frac{w_{tk}^{t+1}(s', a'|s, a)}{n_{t+1k}(s', a')} \left(\operatorname{llnp} n_{t+1k}(s', a') + \frac{1}{2} \ln \frac{3e^2 S^2 AH}{\delta'}\right)
$$

$$
\leq \frac{1}{n_{tk}(s, a)S} \left(\operatorname{llnp}(n_{tk}(s, a)) + \frac{1}{2} \ln \frac{3S^2 AHe^4}{\delta'}\right)
$$

on all but at most $8AS^2 \operatorname{polylog}(A, S, H, 1/\delta, 1/\varepsilon)$ friendly episodes. Similarly, we bound

$$
\sum_{s' \in L'} P(s'|s, a, t) \sqrt{J(s')}
$$

$$
= \sum_{s', a' \in L''} w_{tk}^{t+1}(s', a'|s, a) \sqrt{\frac{1}{n_{t+1k}(s', a')} \left(\operatorname{llnp} n_{t+1k}(s', a') + \frac{1}{2} \ln \frac{3e^2 S^2 AH}{\delta'}\right)}
$$

$$
\leq \sqrt{\frac{1}{n_{tk}(s, a)S} \left(\operatorname{llnp}(n_{tk}(s, a)) + \frac{1}{2} \ln \frac{3S^2 AHe^4}{\delta'}\right)}
$$

on all but at most $8AS^2 \operatorname{polylog}(A, S, H, 1/\delta, 1/\varepsilon)$ friendly episodes. Hence on all friendly episodes except those failure episodes, we get

$$
P(s, a, t)^\top (\tilde{V}_{t+1}^{\pi_k} - V_{t+1}^{\pi_k})^2 \leq c_\varepsilon \varepsilon + \left(c_\varepsilon \varepsilon + \sqrt{\frac{C'^2}{n_{tk}(s, a)S} \left(\operatorname{llnp}(n_{tk}(s, a) + \frac{1}{2} \ln \frac{3AS^2 H \varepsilon^4}{\delta'}\right)}\right)^2.
$$

$\square$

**Lemma E.14.** *Consider a fix $s' \in \mathcal{S}$ and $t \in [H]$, $\delta' \leq \frac{3AS^2 H}{e^2}$ and the good event $F^c$. On all but at most*

$$
\left(32 + 48SH + SH^2\right) ASH^2 \operatorname{polylog}(S, A, H, 1/\delta, 1/\varepsilon)
$$

*friendly episodes $E$ it holds that*

$$
V_t^{\pi_k}(s') - \tilde{V}_t^{\pi_k}(s') \leq c_\varepsilon \varepsilon + \left(1 + \sqrt{12 \ln \frac{3e^2 S^2 AH}{\delta'}}\right) \sqrt{\frac{1}{n_{tk}(s', a')} \left(\operatorname{llnp} n_{tk}(s', a') + \frac{1}{2} \ln \frac{3e^2 S^2 AH}{\delta'}\right)},
$$

*where $a' = \pi_k(s', t)$.*

*Proof.* For any $t, s'$ and $a' = \pi_k(s', t)$ we use Lemma E.15 to write the value difference as

$$
\tilde{V}_t^{\pi_k}(s') - V_t^{\pi_k}(s') = \sum_{u=t}^{H} \sum_{s, a} w_{tk}^u(s, a|s', a')(\tilde{P}_k(s, a, u) - P(s, a, u))^\top \tilde{V}_{u+1}
$$

$$
+ \sum_{u=t}^{H} \sum_{s, a} w_{tk}^u(s, a|s', a')(\tilde{r}_k(s, a, u) - r(s, a, u))
$$

Let $L_k^{ut} = \{s, a \in \mathcal{S} \times \mathcal{A} \ : \ w_{tk}^u(s, a|s', a') \geq w_{\min}\}$ be the set of state-action pairs for which the conditional probability of observing is sufficiently large. Then we can bound the low-probability differences as

$$
\sum_{u=t}^{H} \sum_{s, a \in (L_k^{ut})^c} w_{tk}^u(s, a|s', a')[(\tilde{r}_k(s, a, u) - r(s, a, u)) + (P(s, a, u) - \tilde{P}_k(s, a, u))^\top \tilde{V}_{u+1}]
$$

$$
\leq \sum_{u=t}^{H} \sum_{s, a \in (L_k^{ut})^c} w_{\min} H \leq w_{\min} H^2 S = c_\varepsilon \varepsilon.
$$

For the other terms with significant conditional probability, we can leverage the fact that we only consider events in $(F_k^R)^c$ and $(F_k^P)^c$ to bound

$$\sum_{u=t}^{H} \sum_{s,a \in L_k^{ut}} w_{tk}^u(s,a|s',a')(\tilde{r}_k(s,a,u) - r(s,a,u))$$

$$\leq \sum_{u=t}^{H} \sum_{s,a \in L_k^{ut}} w_{tk}^u(s,a|s',a')\sqrt{\frac{32}{n_{tk}(s,a)}\left(\text{llnp}(n_{tk}(s,a)) + \frac{1}{2}\ln\frac{3SAH}{\delta'}\right)}$$

and

$$\sum_{u=t}^{H} \sum_{s,a \in L_k^{ut}} w_{tk}^u(s,a|s',a')(P(s,a,u) - \tilde{P}_k(s,a,u))^\top \tilde{V}_{u+1}$$

$$\leq \sum_{u=t}^{H} \sum_{s,a \in L_k^{ut}} w_{tk}^u(s,a|s',a') \sum_{s''} \tilde{V}_{u+1}(s'')\sqrt{\frac{2P(s''|s,a,u)}{n_{uk}(s,a)}\left(2\,\text{llnp}(n_{uk}(s,a)) + \ln\frac{3S^2AH}{\delta'}\right)}$$

$$+ \sum_{u=t}^{H} \sum_{s,a \in L_k^{ut}} w_{tk}^u(s,a|s',a') \sum_{s''} \frac{\tilde{V}_{u+1}(s'')}{n_{uk}(s,a)}\left(2\,\text{llnp}(n_{uk}(s,a)) + \ln\frac{3S^2AH}{\delta'}\right)$$

$$\leq \sum_{u=t}^{H} \sum_{s,a \in L_k^{ut}} w_{tk}^u(s,a|s',a')\sqrt{\frac{2SH^2}{n_{uk}(s,a)}\left(2\,\text{llnp}(n_{uk}(s,a)) + \ln\frac{3S^2AH}{\delta'}\right)}$$

$$+ \sum_{u=t}^{H} \sum_{s,a \in L_k^{ut}} w_{tk}^u(s,a|s',a')\frac{SH}{n_{uk}(s,a)}\left(2\,\text{llnp}(n_{uk}(s,a)) + \ln\frac{3S^2AH}{\delta'}\right)$$

where we use Cauchy Schwarz for the last inequality. Combining these individual bounds, we can upper-bound the value difference as

$$\tilde{V}_t^{\pi_k}(s') - V_t^{\pi_k}(s')$$

$$\leq c_\varepsilon \varepsilon + \sum_{u=t}^{H} \sum_{s,a \in L_k^{ut}} w_{tk}^u(s,a|s',a')\sqrt{\frac{(4\sqrt{2}+2\sqrt{S}H)^2}{n_{uk}(s,a)}\left(\text{llnp}(n_{uk}(s,a)) + \frac{1}{2}\ln\frac{3S^2AH}{\delta'}\right)}$$

$$+ \sum_{u=t}^{H} \sum_{s,a \in L_k^{ut}} w_{tk}^u(s,a|s',a')\frac{2SH}{n_{uk}(s,a)}\left(\text{llnp}(n_{uk}(s,a)) + \frac{1}{2}\ln\frac{3S^2AH}{\delta'}\right) \tag{10}$$

We now apply Lemma E.4 with $r=2, D = \frac{1}{2}\ln\frac{3S^2AH}{\delta'}, C = (4\sqrt{2}+2\sqrt{S}H)^2, T = \{t+1, t+2, \dots H\}$ and $\varepsilon' = 1$ and get that the second term above is bounded by

$$\sqrt{\frac{1}{n_{tk}(s',a')}\left(\text{llnp}\, n_{tk}(s',a') + \frac{1}{2}\ln\frac{3e^2S^2AH}{\delta'}\right)}$$

on all friendly episodes but at most

$$\frac{8CASH^2}{\varepsilon'^2}\,\text{polylog}(S,A,H,1/\delta,1/\varepsilon) = (32 + 16\sqrt{2S}H + SH^2)ASH^2\,\text{polylog}(S,A,H,1/\delta,1/\varepsilon)$$

episodes. We apply Lemma E.4 again to the final term in Equation (10) above with $r=1, D = \frac{1}{2}\ln\frac{3S^2AH}{\delta'} \geq 1, T = \{t+1, t+2, \dots H\}, C = 2SH$ and $\varepsilon' = 1$. Then the final term is bounded by $\frac{1}{n_{tk}(s',a')}\left(\text{llnp}\, n_{tk}(s',a') + \frac{1}{2}\ln\frac{3e^2S^2AH}{\delta'}\right)$. on all friendly episodes but

$$\frac{8CASH^2}{\varepsilon'^2}\,\text{polylog}(S,A,H,1/\delta,1/\varepsilon) = 16AS^2H^3\,\text{polylog}(S,A,H,1/\delta,1/\varepsilon)$$

many. Combining these bounds, we arrive at

$$V_t^{\pi_k}(s') - \tilde{V}_t^{\pi_k}(s')$$

$$\leq c_\varepsilon \varepsilon + \sqrt{\frac{1}{n_{tk}(s',a')}\left(\text{llnp}\, n_{tk}(s',a') + \frac{1}{2}\ln\frac{3e^2 S^2 AH}{\delta'}\right)}$$

$$+ \frac{1}{n_{tk}(s',a')}\left(\text{llnp}\, n_{tk}(s',a') + \frac{1}{2}\ln\frac{3e^2 S^2 AH}{\delta'}\right)$$

$$\leq c_\varepsilon \varepsilon + \left(1 + \sqrt{\frac{1}{2}\ln\frac{3e^2 S^2 AH}{\delta'}}\right)\sqrt{\frac{1}{n_{tk}(s',a')}\left(\text{llnp}\, n_{tk}(s',a') + \frac{1}{2}\ln\frac{3e^2 S^2 AH}{\delta'}\right)},$$

where we bounded $\sqrt{\frac{1}{n_{tk}(s',a')}\left(\text{llnp}\, n_{tk}(s',a') + \frac{1}{2}\ln\frac{3e^2 S^2 AH}{\delta'}\right)}$ by $\frac{1}{2}\ln\frac{3e^2 S^2 AH}{\delta'}$ since it is decreasing in $n_{tk}(s',a')$ and we therefore can simply use $n_{tk}(s',a') = 1$ (entire bound holds trivially for $n_{tk}(s',a') = 0$). $\qquad\square$

### E.5 Useful Lemmas

**Lemma E.15** (Value Difference Lemma). *For any two MDPs $M'$ and $M''$ with rewards $r'$ and $r''$ and transition probabilities $P'$ and $P''$, the difference in values with respect to the same policy $\pi$ can be written as*

$$V_i'(s) - V_i''(s) = \mathbb{E}''\left[\sum_{t=i}^{H}(r'(s_t,a_t,t) - r''(s_t,a_t,t))\Big| s_i = s\right] + \mathbb{E}''\left[\sum_{t=i}^{H}(P'(s_t,a_t,t) - P''(s_t,a_t,t))^\top V_{t+1}'\Big| s_i = s\right]$$

*where $V_{H+1}' = V_{H+1}'' = \vec{0}$ and the expectation $\mathbb{E}'$ is taken w.r.t to $P'$ and $\pi$ and $\mathbb{E}''$ w.r.t. $P''$ and $\pi$.*

*Proof.* For $i = H + 1$ the statement is trivially true. We assume now it holds for $i + 1$ and show it holds also for $i$. Using only this induction hypothesis and basic algebra, we can write

$$V_i'(s) - V_i''(s)$$
$$=\mathbb{E}_\pi[r'(s_i,a_i,i) + V_{i+1}'^\top P'(s_i,a_i,i) - r''(s_i,a_i,i) - V_{i+1}''^\top P''(s_i,a_i,i)|s_i = s]$$
$$=\mathbb{E}_\pi[r'(s_i,a_i,i) - r''(s_i,a_i,i)|s_i = s] + \mathbb{E}_\pi\left[\sum_{s'\in\mathcal{S}} V_{i+1}'(s')(P'(s'|s_i,a_i,i) - P''(s'|s_i,a_i,i))\Big| s_i = s\right]$$

$$+ \mathbb{E}_\pi\left[\sum_{s'\in\mathcal{S}} P''(s'|s_i,a_i,i)(V_{i+1}'(s') - V_{i+1}''(s'))\Big| s_i = s\right]$$
$$=\mathbb{E}_\pi[r'(s_i,a_i,i) - r''(s_i,a_i,i)|s_i = s] + \mathbb{E}_\pi\left[\sum_{s'\in\mathcal{S}} V_{i+1}'(s')(P'(s'|s_i,a_i,i) - P''(s'|s_i,a_i,i))\Big| s_i = s\right]$$

$$+ \mathbb{E}''\left[V_{i+1}'(s_{i+1}) - V_{i+1}''(s_{i+1}))\Big| s_i = s\right]$$
$$=\mathbb{E}_\pi[r'(s_i,a_i,i) - r''(s_i,a_i,i)|s_i = s] + \mathbb{E}_\pi\left[\sum_{s'\in\mathcal{S}} V_{i+1}'(s')(P'(s'|s_i,a_i,i) - P''(s'|s_i,a_i,i))\Big| s_i = s\right]$$

$$+ \mathbb{E}''\left[\mathbb{E}''\left[\sum_{t=i+1}^{H}(r'(s_t,a_t,t) - r''(s_t,a_t,t))\Big| s_{i+1}\right] + \mathbb{E}''\left[\sum_{t=i+1}^{H}(P'(s_t,a_t,t) - P''(s_t,a_t,t))^\top V_{t+1}'\Big| s_{i+1}\right]\Big| s_i = s\right]$$

$$=\mathbb{E}''\left[\sum_{t=i}^{H}(r'(s_t,a_t,t)-r''(s_t,a_t,t))\Big|s_i=s\right]+\mathbb{E}''\left[\sum_{t=i}^{H}(P'(s_t,a_t,t)-P''(s_t,a_t,t))^\top V'_{t+1}\Big|s_i=s\right]$$

where the last equality follows from law of total expectation □

**Lemma E.16** (Algorithm ensures optimism). *On the good event $F^c$ it holds that for all episodes $k$, $t \in [H]$, $s \in \mathcal{S}$ that*

$$V_t^{\pi_k}(s) \leq V_t^\star(s) \leq \tilde{V}_t^{\pi_k}(s).$$

*Proof.* The first inequality follows simply from the definition of the optimal value function $V^\star$.

Since all outcome we consider are in the event $(F_k^V)^c$, we know that the true transition probabilities $P$, the optimal policy $\pi^\star$ and optimal policy $V^\star$ are a feasible solution for the optimistic planning problem in Lemma D.1 that UBEV solves. It therefore follows immediately that $p_0^\top \tilde{V}_1^{\pi_k} \geq p_0^\top V_1^\star$. □

## F General Concentration Bounds

**Lemma F.1.** *Let $X_1, X_2, \ldots$ be a martingale difference sequence adapted to filtration $\{\mathcal{F}_t\}_{t=1}^\infty$ with $X_t$ conditionally $\sigma^2$-subgaussian so that $\mathbb{E}[\exp(\lambda(X_t-\mu))|\mathcal{F}_{t-1}] \leq \exp(\lambda^2\sigma^2/2)$ almost surely for all $\lambda \in \mathbb{R}$. Then with $\hat{\mu}_t = \frac{1}{t}\sum_{i=1}^t X_i$ we have for all $\delta \in (0,1]$*

$$\mathbb{P}\left(\exists t : |\hat{\mu}_t - \mu| \geq \sqrt{\frac{4\sigma^2}{t}\left(2\operatorname{llnp}(t)+\ln\frac{3}{\delta}\right)}\right) \leq 2\delta.$$

*Proof.* Let $S_t = \sum_{s=1}^t (X_s - \mu)$. Then

$$\mathbb{P}\left(\exists t : \hat{\mu}_t - \mu \geq \sqrt{\frac{4\sigma^2}{t}\left(2\operatorname{llnp}(t)+\ln\frac{3}{\delta}\right)}\right)$$

$$\leq \mathbb{P}\left(\exists t : S_t \geq \sqrt{4\sigma^2 t\left(2\operatorname{llnp}(t)+\ln\frac{3}{\delta}\right)}\right)$$

$$\leq \sum_{k=0}^\infty \mathbb{P}\left(\exists t \in [2^k, 2^{k+1}] : S_t \geq \sqrt{4\sigma^2 t\left(2\operatorname{llnp}(t)+\ln\frac{3}{\delta}\right)}\right)$$

$$\leq \sum_{k=0}^\infty \mathbb{P}\left(\exists t \leq 2^{k+1} : S_t \geq \sqrt{2\sigma^2 2^{k+1}\left(2\operatorname{llnp}(2^k)+\ln\frac{3}{\delta}\right)}\right)$$

We now consider $M_t = \exp(\lambda S_t)$ for $\lambda > 0$ which is a nonnegative sub-martingale and use the short-hand $f = \sqrt{2\sigma^2 2^{k+1}\left(2\operatorname{llnp}(2^k)+\ln\frac{3}{\delta}\right)}$. Then by Doob's maximal inequality for nonnegative submartingales

$$\mathbb{P}\left(\exists t \leq 2^{k+1} : S_t \geq f\right) = \mathbb{P}\left(\max_{t \leq 2^{k+1}} M_t \geq \exp(\lambda f)\right) \leq \frac{\mathbb{E}[M_{2^{k+1}}]}{\exp(\lambda f)} \leq \exp\left(2^{k+1}\frac{\lambda^2\sigma^2}{2}-\lambda f\right).$$

Choosing the optimal $\lambda = \frac{f}{\sigma^2 2^{k+1}}$ we obtain the bound

$$\mathbb{P}\left(\exists t \leq 2^{k+1} : S_t \geq f\right) \leq \exp\left(-\frac{f^2}{2^{k+2}\sigma^2}\right) = \exp\left(-2\operatorname{llnp}(2^k)-\ln\frac{3}{\delta}\right) = \frac{\delta}{3}\exp\left(-2\operatorname{llnp}(2^k)\right)$$

$$\tag{11}$$

$$= \frac{\delta}{3}\exp\left(-\max\{0, 2\ln\max\{0, \ln 2^k\}\}\right) = \frac{\delta}{3}\min\left\{1, (k\ln 2)^{-2}\right\}$$

$$\leq \frac{\delta}{3}\min\left\{1, \frac{1}{k^2\ln 2}\right\}.$$

Plugging this back in the bound from above, we get

$$\mathbb{P}\left(\exists t : \hat{\mu}_t - \mu \geq \sqrt{\frac{4\sigma^2}{t}\left(2\,\mathrm{llnp}(t) + \ln\frac{3}{\delta}\right)}\right) \leq \frac{\delta}{3}\sum_{k=0}^{\infty}\min\left\{1, \frac{1}{k^2\ln(2)}\right\}$$

$$=\delta\,\frac{1}{3}\left(\frac{\pi^2}{6\ln 2} + 2 - 1/\ln(2)\right) \leq \delta. \quad (12)$$

For the other side, the argument follows completely analogously with

$$\mathbb{P}\left(\exists t \leq 2^{k+1} : S_t \leq -f\right) = \mathbb{P}\left(\exists t \leq 2^{k+1} : -S_t \geq f\right)$$

$$= \mathbb{P}\left(\max_{t \leq 2^{k+1}} \exp(-\lambda S_t) \geq \exp(\lambda f)\right)$$

$$\leq \frac{\mathbb{E}[\exp(-\lambda S_{2^{k+1}})]}{\exp(\lambda f)} \leq \exp\left(2^{k+1}\frac{\lambda^2\sigma^2}{2} - \lambda f\right).$$

$\square$

**Lemma F.2.** *Let $X_1, X_2, \ldots$ be a sequence of Bernoulli random variables with bias $\mu \in [0, 1]$. Then for all $\delta \in (0, 1]$*

$$\mathbb{P}\left(\exists t : |\hat{\mu}_t - \mu| \geq \sqrt{\frac{2\mu}{t}\left(2\,\mathrm{llnp}(t) + \ln\frac{3}{\delta}\right)} + \frac{1}{t}\left(2\,\mathrm{llnp}(t) + \ln\frac{3}{\delta}\right)\right) \leq 2\delta$$

*Proof.*

$$\mathbb{P}\left(\exists t : \hat{\mu}_t - \mu \geq \sqrt{\frac{2\mu}{t}\left(2\,\mathrm{llnp}(t) + \ln\frac{3}{\delta}\right)} + \frac{1}{t}\left(2\,\mathrm{llnp}(t) + \ln\frac{3}{\delta}\right)\right)$$

$$= \mathbb{P}\left(\exists t : S_t \geq \sqrt{2\mu t\left(2\,\mathrm{llnp}(t) + \ln\frac{3}{\delta}\right)} + 2\,\mathrm{llnp}(t) + \ln\frac{3}{\delta}\right)$$

$$\leq \sum_{k=0}^{\infty}\mathbb{P}\left(\exists t \leq 2^{k+1} : S_t \geq \sqrt{2\mu 2^k\left(2\,\mathrm{llnp}(2^k) + \ln\frac{3}{\delta}\right)} + 2\,\mathrm{llnp}(2^k) + \ln\frac{3}{\delta}\right)$$

Let $g = 2\,\mathrm{llnp}(2^k) + \ln\frac{3}{\delta}$ and $f = \sqrt{2^{k+1}\mu g} + g$. Further define $S_t = \sum_{i=1}^{t} X_i - t\mu$ and $M_t = \exp(\lambda S_t)$ which is by construction a nonnegative submartingale. Applying Doob's maximal inequality for nonnegative submartingales, we bound

$$\mathbb{P}\left(\exists t \leq 2^{k+1} : S_t \geq f\right) = \mathbb{P}\left(\max_{i \leq 2^{k+1}} M_i \geq \exp(\lambda f)\right) \leq \frac{\mathbb{E}[M_{2^{k+1}}]}{\exp(\lambda f)} = \exp\left(\ln\mathbb{E}[M_{2^{k+1}}] - \lambda f\right).$$

Since this holds for all $\lambda \in \mathbb{R}$, we can bound

$$\mathbb{P}\left(\exists t \leq 2^{k+1} : S_t \geq f\right) \leq \exp\left(-\sup_{\lambda \in \mathbb{R}}(\lambda f - \ln\mathbb{E}[M_{2^{k+1}}])\right)$$

and using Corollary 2.11 by Boucheron et al. [25] (see also note below proof of Corollary 2.11) bound that by

$$\exp\left(-\frac{f^2}{2(2^{k+1}\mu + f/3)}\right)$$

We now argue that this quantity can be upper-bounded by $\exp(-g)$. This is equivalent to

$$-\frac{f^2}{2(2^{k+1}\mu + f/3)} \leq -g$$

$$f^2 \geq 2g(2^{k+1}\mu + f/3) = \frac{2}{3}gf + \frac{2^{k+2}}{3}\mu g$$

$$g^2 + 2\sqrt{2^{k+1}\mu g}g + 2^{k+1}\mu g \geq \frac{2}{3}g^2 + \frac{2}{3}\sqrt{2^{k+1}\mu g}g + \frac{2^{k+2}}{3}\mu g$$

$$\frac{1}{3}g^2 + \frac{4}{3}\sqrt{2^{k+1}\mu g}g + \frac{1}{3}2^{k+1}\mu g \geq 0.$$

Each line is an equivalent inequality since $g, f \geq 0$ and each term on the left in the final inequality is nonnegative. Hence, we get $\mathbb{P}\left(\exists t \leq 2^{k+1} : S_t \geq f\right) \leq \exp(-g)$. Following now the arguments from the proof of Lemma F.1 in Equations (11)–(12), we obtain that

$$\mathbb{P}\left(\exists t : \hat{\mu}_t - \mu \geq \sqrt{\frac{2\mu}{t}\left(2\,\mathrm{llnp}(t) + \ln\frac{3}{\delta}\right)} + \frac{1}{t}\left(2\,\mathrm{llnp}(t) + \ln\frac{3}{\delta}\right)\right) \leq \delta.$$

For the other direction, we proceed analogously to above and arrive at

$$\mathbb{P}\left(\exists t \leq 2^{k+1} : -S_t \geq f\right) \leq \exp\left(-\sup_{\lambda \in \mathbb{R}}\left(-\lambda f - \ln \mathbb{E}[M_{2^{k+1}}]\right)\right)$$

which we bound similarly to above by

$$\exp\left(-\frac{f^2}{2(2^{k+1}\mu - f/3)}\right) \leq \exp\left(-\frac{f^2}{2(2^{k+1}\mu + f/3)}\right) \leq \exp(-g).$$

$\square$

**Lemma F.3** (Uniform L1-Deviation Bound for Empirical Distribution). *Let $X_1, X_2, \ldots$ be a sequence of i.i.d. categorical variables on $[U]$ with distribution $P$. Then for all $\delta \in (0, 1]$*

$$\mathbb{P}\left(\exists t : \|\hat{P}_t - P\|_1 \geq \sqrt{\frac{4}{t}\left(2\,\mathrm{llnp}(t) + \ln\frac{3(2^U - 2)}{\delta}\right)}\right) \leq \delta$$

*where $\hat{P}_t$ is the empirical distribution based on samples $X_1 \ldots X_t$.*

*Proof.* We use the identity $\|Q - P\|_1 = 2\max_{B \subseteq \mathcal{B}} Q(B) - P(B)$ which holds for all distributions $P, Q$ defined on the finite set $\mathcal{B}$ to bound

$$\mathbb{P}\left(\exists t : \|\hat{P}_t - P\|_1 \geq \sqrt{\frac{4}{t}\left(2\,\mathrm{llnp}(t) + \ln\frac{3(2^U - 2)}{\delta}\right)}\right)$$

$$= \mathbb{P}\left(\max_{t, B \subseteq [U]} \hat{P}_t(B) - P(B) \geq \frac{1}{2}\sqrt{\frac{4}{t}\left(2\,\mathrm{llnp}(t) + \ln\frac{3(2^U - 2)}{\delta}\right)}\right)$$

$$\leq \sum_{B \subseteq [U]} \mathbb{P}\left(\max_t \hat{P}_t(B) - P(B) \geq \sqrt{\frac{1}{t}\left(2\,\mathrm{llnp}(t) + \ln\frac{3(2^U - 2)}{\delta}\right)}\right).$$

Define now $S_t = \sum_{i=1}^{t} \mathbb{I}\{X_1 \in B\} - tP(B)$ which is a martingale sequence. Then the last line above is equivalent to

$$\sum_{B \subseteq [U]} \mathbb{P}\left(\max_t S_t \geq \sqrt{t\left(2\,\mathrm{llnp}(t) + \ln\frac{3(2^U - 2)}{\delta}\right)}\right)$$

$$\leq \sum_{B \subseteq [U]} \mathbb{P}\left(\max_{k \in \mathbb{N}, t \in [2^k, 2^{k+1}]} S_t \geq \sqrt{t\left(2\,\mathrm{llnp}(t) + \ln\frac{3(2^U - 2)}{\delta}\right)}\right)$$

$$\leq \sum_{B \subseteq [U]} \sum_{k=0}^{\infty} \mathbb{P}\left(\max_{t \in [2^k, 2^{k+1}]} S_t \geq \sqrt{t\left(2\,\mathrm{llnp}(t) + \ln\frac{3(2^U - 2)}{\delta}\right)}\right)$$

$$\leq \sum_{B \subseteq [U]} \sum_{k=0}^{\infty} \mathbb{P}\left(\max_{t \leq 2^{k+1}} S_t \geq \sqrt{2^k\left(2\,\mathrm{llnp}(2^k) + \ln\frac{3(2^U - 2)}{\delta}\right)}\right)$$

$$= \sum_{B \subseteq [U]} \sum_{k=0}^{\infty} \mathbb{P}\left(\max_{t \leq 2^{k+1}} \exp(\lambda S_t) \geq \exp(\lambda f)\right)$$

$$= \sum_{B \subseteq [U], B \neq \emptyset, B \neq [U]} \sum_{k=0}^{\infty} \mathbb{P}\left(\max_{t \leq 2^{k+1}} \exp(\lambda S_t) \geq \exp(\lambda f)\right)$$

where $f = \sqrt{2^k\left(2\,\mathrm{llnp}(2^k) + \ln\frac{3(2^U - 2)}{\delta}\right)}$ and $\lambda \in \mathbb{R}$ and the last equality follows from the fact that for $B = \emptyset$ and $B = [U]$ the difference between the distributions has to be 0. Since $\mathbb{I}\{X_1 \in B\} - tP(B)$ is a centered Bernoulli variable it is $1/2$-subgaussian and so $S_t$ satisfies $\mathbb{E}[\exp(\lambda S_t)] \leq \exp(\lambda^2 t/8)]$. Since $S_t$ is a martingale, $\exp(\lambda S_t)$ is a nonnegative sub-martingale and we can apply the maximal inequality to bound

$$\mathbb{P}\left(\max_{t \leq 2^{k+1}} \exp(\lambda S_t) \geq \exp(\lambda f)\right) \leq \exp\left(\frac{1}{8}\lambda^2 2^{k+1} - \lambda f\right).$$

Choosing $\lambda = \frac{4f}{2^{k+1}}$, we get $\mathbb{P}\left(\max_{t \leq 2^{k+1}} \exp(\lambda S_t) \geq \exp(\lambda f)\right) \leq \exp\left(-\frac{f^2}{2^k}\right)$. Hence, using the same steps as in the proof of Lemma F.1, we get $\mathbb{P}\left(\max_{t \leq 2^{k+1}} \exp(\lambda S_t) \geq \exp(\lambda f)\right) \leq \frac{\delta}{3(2^{[U]}-2)} \min\left\{1, \frac{1}{k^2 \ln 2}\right\}$ and then

$$\mathbb{P}\left(\exists t : \|\hat{P}_t - P\|_1 \geq \sqrt{\frac{4}{t}\left(2\,\mathrm{llnp}(t) + \ln\frac{3(2^U - 2)}{\delta}\right)}\right)$$

$$\leq \sum_{B \subseteq [U], B \neq \emptyset, B \neq [U]} \frac{\delta}{3(2^{[U]} - 2)} \sum_{k=0}^{\infty} \min\left\{1, \frac{1}{k^2 \ln 2}\right\} \leq \sum_{B \subseteq [U], B \neq \emptyset, B \neq [U]} \frac{\delta}{2^{[U]} - 2} = \delta.$$

$\square$

**Lemma F.4.** *Let $\mathcal{F}_i$ for $i = 1 \ldots$ be a filtration and $X_1, \ldots X_n$ be a sequence of Bernoulli random variables with $\mathbb{P}(X_i = 1 | \mathcal{F}_{i-1}) = P_i$ with $P_i$ being $\mathcal{F}_{i-1}$-measurable and $X_i$ being $\mathcal{F}_i$ measurable. It holds that*

$$\mathbb{P}\left(\exists n : \sum_{t=1}^{n} X_t < \sum_{t=1}^{n} P_t/2 - W\right) \leq e^{-W}$$

*Proof.* $P_t - X_t$ is a Martingale difference sequence with respect to the filtration $\mathcal{F}_t$. Since $X_t$ is nonnegative and has finite second moment, we have for any $\lambda > 0$ that $\mathbb{E}\left[e^{-\lambda(X_t - P_t)} | \mathcal{F}_{t-1}\right] \leq e^{\lambda^2 P_t/2}$ (Exercise 2.9, Boucheron et al. [25]). Hence, we have

$$\mathbb{E}\left[e^{\lambda(P_t - X_t) - \lambda^2 P_t/2} | \mathcal{F}_{t-1}\right] \leq 1$$

and by setting $\lambda = 1$, we see that

$$M_n = e^{\sum_{t=1}^n (-X_t + P_t/2)}$$

is a supermartingale. It hence holds by Markov's inequality

$$\mathbb{P}\left(\sum_{t=1}^n (-X_t + P_t/2) \geq W\right) = \mathbb{P}\left(M_n \geq e^W\right) \leq e^{-W}\mathbb{E}[M_n] \leq e^{-W}$$

wich gives us the derised result

$$\mathbb{P}\left(\sum_{t=1}^n X_t \leq \sum_{t=1}^n P_t/2 - W\right) \leq e^{-W}$$

for a fixed $n$. We define now the stopping time $\tau = \min\{t \in \mathbb{N} : M_t > e^W\}$ and the sequence $\tau_n = \min\{t \in \mathbb{N} : M_t > e^W \vee t \geq n\}$. Applying the convergence theorem for nonnegative supermartingales (Theorem 5.2.9 in Durrett [26]), we get that $\lim_{t\to\infty} M_t$ is well-defined almost surely. Therefore, $M_\tau$ is well-defined even when $\tau = \infty$. By the optional stopping theorem for nonnegative supermartingales (Theorem 5.7.6 by Durrett [26]), we have $\mathbb{E}[M_{\tau_n}] \leq \mathbb{E}[M_0] \leq 1$ for all $n$ and applying Fatou's lemma, we obtain $\mathbb{E}[M_\tau] = \mathbb{E}[\lim_{n\to\infty} M_{\tau_n}] \leq \liminf_{n\to\infty} \mathbb{E}[M_{\tau_n}] \leq 1$. Using Markov's inequality, we can finally bound

$$\mathbb{P}\left(\exists n : \sum_{t=1}^n X_t < \frac{1}{2}\sum_{t=1}^n P_t - W\right) \leq \mathbb{P}(\tau < \infty) \leq \mathbb{P}(M_\tau > e^W) \leq e^{-W}\mathbb{E}[M_\tau] \leq e^{-W}.$$

$\square$

[Supplementary Material 2]

# Appendices of Unifying PAC and Regret: Uniform PAC Bounds for Episodic Reinforcement Learning

# A    Framework Relation Proofs

## A.1    Proof of Theorem 1

*Proof.* We will use two episodic MDPs, $M_1$ and $M_2$, which are essentially 2-armed bandits and hard to distinguish to prove this statement. Both MDPs have one state, horizon $H = 1$, and two actions $\mathcal{A} = \{1, 2\}$. For a fixed $\alpha > 0$, the rewards are Bernoulli$(1/2 + \alpha/2)$ distributed for actions 1 in both MDPs. Playing action 2 in $M_1$ gives Bernoulli$(1/2)$ rewards and action 2 in $M_2$ gives Bernoulli$(1/2 + \alpha)$ rewards.

Assume now that an algorithm in MDP $M_1$ with nonzero probability plays the suboptimal action only at most $N$ times in total, i.e., $\mathbb{P}_{M_1}(n_2 \leq N) \geq \beta$ where $n_2$ is the number of times action 2 is played and $\infty > N > 0, \beta > 0$. Then

$$\mathbb{P}_{M_1}(n_2 \leq N) = \mathbb{E}_{M_1}\left[\mathbb{I}\{n_2 \leq N\}\right] = \mathbb{E}_{M_2}\left[\frac{\mathbb{P}_{M_1}(Y_\infty)}{\mathbb{P}_{M_2}(Y_\infty)}\mathbb{I}\{n_2 \leq N\}\right]$$

where $Y_k = (A_1, R_1, A_2, R_2, \ldots A_k, R_k)$ denotes the entire sequence of observed rewards $R_i$ and action indices $A_i$ after $k$ episodes. Since $\mathbb{P}_{M_1}(A_k|Y_{k-1}) = \mathbb{P}_{M_2}(A_k|Y_{k-1})$ and $\mathbb{P}_{M_1}(R_k|A_k = 1, Y_{k-1}) = \mathbb{P}_{M_2}(R_k|A_k = 1, Y_{k-1})$ and

$$\frac{\mathbb{P}_{M_1}(R_k|A_k = 2, Y_{k-1})}{\mathbb{P}_{M_2}(R_k|A_k = 2, Y_{k-1})} \leq \max\left\{\frac{1/2}{1/2 + \alpha}, \frac{1/2}{1/2 - \alpha}\right\} = \frac{1}{1 - 2\alpha}$$

the likelihood ratio of $Y_\infty$ is upper bounded by $(1 + 2\alpha)^N$ if the second action has been chosen at most $N$ times. Hence

$$\mathbb{P}_{M_2}[n_2 \leq N] = \frac{(1 - 2\alpha)^N}{(1 - 2\alpha)^N}\mathbb{E}_{M_2}\left[\mathbb{I}\{n_2 \leq N\}\right] \geq (1 - 2\alpha)^N\mathbb{E}_{M_2}\left[\frac{\mathbb{P}_{M_1}(Y_\infty)}{\mathbb{P}_{M_2}(Y_\infty)}\mathbb{I}\{n_2 \leq N\}\right]$$

$$\geq (1 - 2\alpha)^N\beta > 0$$

Therefore, the regret for $M_2$ is for $T$ large enough $\mathbb{E}_{M_2}R(T) \geq (T - N)\beta(1 - 2\alpha)^N\alpha/2 = O(T)$. Hence, for the algorithm to ensure sublinear regret for $M_2$, it has to play the suboptimal action for $M_1$ infinitely often with probability 1. This however implies that the algorithm cannot satisfy any finite PAC bound for accuracy $\varepsilon < \alpha/2$.                                                                       $\square$

## A.2    Proof of Theorem 2

*Proof.* **PAC Bound to high-probability regret bound:** Consider a fixed $\delta > 0$ and PAC bound with $F_{\text{PAC}} = \Theta(1/\varepsilon^2)$. Then there is a $C > 0$ such that the following algorithm satisfies the PAC bound. The algorithm uses the worst possible policy with optimality gap $H$ in all episodes on some event $E$ and in the first $C/\varepsilon^2$ episodes on the complimentary event $E^C$. For the remaining episodes on $E^C$ it follows a policy with optimality gap $\varepsilon$. The probability of $E$ is $\delta$. The regret of the algorithm on $E$ is $R(T) = TH$ and on $E^C$ it is $R(T) = \min\{T, C/\varepsilon^2\}H + \min\{T - C/\varepsilon^2, 0\}\varepsilon$. For $T \geq C/\varepsilon^2$, on any event the regret of this algorithm is at least

$$R(T) = \frac{CH}{\varepsilon^2} + \left(T - \frac{C}{\varepsilon^2}\right)\varepsilon = T\varepsilon + \frac{C(H - \varepsilon)}{\varepsilon^2}. \tag{8}$$

The quantity

$$\frac{R(T)}{T^{2/3}} = \frac{C(H - \varepsilon)}{T^{2/3}\varepsilon^2} + \varepsilon T^{1/3}$$

takes its minimum at $T = \frac{C(H-\varepsilon)}{\varepsilon^3}$ with a positive value and hence $R(T) = \Omega(T^{2/3})$. Therefore a PAC bound with rate $1/\varepsilon^2$ implies at best a high-probability regret bound of order $O(T^{2/3})$ and is only tight at $T = \Theta(1/\varepsilon^3)$. Furthermore, by looking at Equation (8), we see that for any fixed $\varepsilon$, there is an algorithm that has uniform high-probability regret that is $\Omega(T)$.

**PAC Bound to uniform high-probability regret bound:** Consider a fixed $\delta > 0$ and $\varepsilon > 0$ and a PAC bound $F_{\text{PAC}}$ that evaluates to some value $N$ for parameter $\varepsilon$. The algorithm uses the worst possible policy with optimality gap $H$ in all episodes on some event $E$ and in the first $N$ episodes on

Figure 3: Relation of PAC-bound and Regret; The area of the shaded regions are a bound on the regret after $T$ episodes.

the complimentary event $E^C$. For the remaining episodes on $E^C$ it follows a policy with optimality gap $\varepsilon$. The probability of $E$ is $\delta$. The regret of the algorithm on $E$ is $R(T) = TH$ and on $E^C$ it is $R(T) = \min\{T, N\}H + \min\{T - N, 0\}\varepsilon$. For $T \geq N$, on any event the regret of this algorithm is at least

$$R(T) = NH + (T - N)\varepsilon = T\varepsilon + H(T - N) = \Omega(T).$$

**Uniform high-probability regret bound to PAC bound:** Consider an MDP such that at least one suboptimal policy exists with optimality gap $\varepsilon > 0$. Further let $L(T)$ be a nondecreasing function with $F_{\text{UHPR}}(T) \geq L(T)$ and $L(T) \to \infty$ as $T \to \infty$. Then the algorithm plays the optimal policy except for episodes $k$ where $\lfloor L(k-1)/\varepsilon \rfloor \neq \lfloor L(k)/\varepsilon \rfloor$. This algorithm satisfies the regret bound but makes infinitely many $\varepsilon/2$-mistakes with probability 1.

**Uniform high-probability regret bound to expected regret bound:** Consider an MDP such that at least one suboptimal policy exists with optimality gap $\varepsilon > 0$. Consider an algorithm that with probability $\delta$ always plays the suboptimal policy and with probability $1 - \delta$ always plays the optimal policy. This algorithm satisfies the uniform high-probability regret bound but suffers regret $\mathbb{E}R(T) = \delta\varepsilon T = \Omega(T)$. $\qquad\square$

### A.3   Proof of Theorem 3

*Proof.* **Convergence to optimal policies:** The convergence to the set of optimal policies follows directly by using the definition of limits on the $\Delta_k$ sequence for each outcome in the high-probability event where the bound holds.

$(\varepsilon, \delta)$**-PAC:** Due to sub-additivity of probabilities, we have

$$\mathbb{P}\left(N_\varepsilon > F_{\text{PAC}}\left(\frac{1}{\varepsilon}, \log\frac{1}{\delta}\right)\right) \leq \mathbb{P}\left(\bigcup_{\varepsilon'}\left\{N_{\varepsilon'} > F_{\text{PAC}}\left(\frac{1}{\varepsilon'}, \log\frac{1}{\delta}\right)\right\}\right)$$

$$=\mathbb{P}\left(\exists \varepsilon' \; : \; N_{\varepsilon'} > F_{\text{PAC}}\left(\frac{1}{\varepsilon'}, \log\frac{1}{\delta}\right)\right) \leq \delta.$$

**High-Probability Regret Bound:** This part is proved separately in Theorem A.1 below. $\qquad\square$

**Theorem A.1** (Uniform-PAC to Regret Conversion Theorem). *Assume on some event $E$ an algorithm follows for all $\varepsilon$ an $\varepsilon$-optimal policy $\pi_k$, i.e., $\Delta_k \leq \varepsilon$, on all but at most*

$$\frac{C_1}{\varepsilon}\left(\ln\frac{C_3}{\varepsilon}\right)^k + \frac{C_2}{\varepsilon^2}\left(\ln\frac{C_3}{\varepsilon}\right)^{2k}$$

*episodes where $C_1 \geq C_2 \geq 2$ and $C_3 \geq \max\{H, e\}$ and $C_1, C_2, C_3$ do not depend on $\varepsilon$. Then this algorithm has on this event a regret of*

$$R(T) \leq (\sqrt{C_2 T} + C_1)\,\text{polylog}(T, C_3, C_1) = O(\sqrt{C_2 T}\,\text{polylog}(T, C_3, C_1, H))$$

*for all number of episodes $T$.*

*Proof.* The mistake bound $g(\varepsilon) = \frac{C_1}{\varepsilon}\left(\ln\frac{C_3}{\varepsilon}\right)^k + \frac{C_2}{\varepsilon^2}\left(\ln\frac{C_3}{\varepsilon}\right)^{2k} \leq T$ is monotonically decreasing for $\varepsilon \in (0, H]$. For a given $T$ large enough, we can therefore find an $\varepsilon_{\min} \in (0, H]$ such that $g(\varepsilon) \leq T$ for all $\varepsilon \in (\varepsilon_{\min}, H]$. The regret $R(T)$ of the algorithm can then be bounded as follows

$$R(T) \leq T\varepsilon_{\min} + \int_{\varepsilon_{\min}}^{H} g(\varepsilon)d\varepsilon.$$

This bound assumes the worst case where first the algorithm makes the worst mistakes possible with regret $H$ and subsequently less and less severe mistakes controlled by the mistake bound. For a better intuition, see Figure 3.

We first find a suitable $\varepsilon_{\min}$. Define $y = \frac{1}{\varepsilon}\left(\ln\frac{C_3}{\varepsilon}\right)^k$ then since $g$ is monotonically decreasing, it is sufficient to find a $\varepsilon$ with $g(\varepsilon) \leq T$. That is equivalent to $C_1 y + C_2 y^2 \leq T$ for which

$$\frac{1}{\varepsilon}\left(\ln\frac{C_3}{\varepsilon}\right)^k = y \leq \frac{C_1}{2C_2} + \frac{\sqrt{C_1^2 + 4TC_2}}{2C_2} =: a$$

is sufficient. We set now

$$\varepsilon_{\min} = \frac{\ln(C_3 a)^k}{a} = \frac{2C_2}{C_1 + \sqrt{C_1^2 + 4TC_2}}\left(\ln\frac{(C_1 + \sqrt{C_1^2 + 4TC_2})C_3}{2C_2}\right)^k$$

which is a valid choice as

$$\frac{1}{\varepsilon_{\min}}\left(\ln\frac{C_3}{\varepsilon_{\min}}\right)^k = \frac{a}{\ln(C_3 a)^k}\left(\ln\frac{C_3 a}{\ln(C_3 a)^k}\right)^k = \frac{a}{\ln(C_3 a)^k}\left(\ln(C_3 a) - k\ln\ln(C_3 a)\right)^k$$

$$\leq \frac{a}{\ln(C_3 a)^k}\left(\ln(C_3 a)\right)^k = a.$$

We now first bound the regret further as

$$R(T) \leq T\varepsilon_{\min} + \int_{\varepsilon_{\min}}^{H} g(\varepsilon)d\varepsilon \leq T\varepsilon_{\min} + C_1\left(\ln\frac{C_3}{\varepsilon_{\min}}\right)^k\int_{\varepsilon_{\min}}^{H}\frac{1}{\varepsilon}d\varepsilon + C_2\left(\ln\frac{C_3}{\varepsilon_{\min}}\right)^{2k}\int_{\varepsilon_{\min}}^{H}\frac{1}{\varepsilon^2}d\varepsilon$$

$$= T\varepsilon_{\min} + C_1\left(\ln\frac{C_3}{\varepsilon_{\min}}\right)^k\ln\frac{H}{\varepsilon_{\min}} + C_2\left(\ln\frac{C_3}{\varepsilon_{\min}}\right)^{2k}\left[\frac{1}{\varepsilon_{\min}} - \frac{1}{H}\right]$$

and then use the choice of $\varepsilon_{\min}$ from above to look at each of the terms in this bound individually. In the following bounds we extensively use the fact $\ln(a + b) \leq \ln(a) + \ln(b) = \ln(ab)$ for all $a, b \geq 2$ and that $\sqrt{a + b} \leq \sqrt{a} + \sqrt{b}$ which holds for all $a, b \geq 0$.

$$T\varepsilon_{\min} = \frac{2TC_2}{C_1 + \sqrt{C_1^2 + 4TC_2}}\left(\ln\frac{C_3(C_1 + \sqrt{C_1^2 + 4TC_2})}{2C_2}\right)^k$$

$$\leq \frac{2TC_2}{\sqrt{4TC_2}}\left(\ln C_3 + \ln C_1 + \ln C_1 + \ln\frac{2\sqrt{TC_2}}{2C_2}\right)^k$$

$$\leq \sqrt{TC_2}\left(\ln(C_3 C_1^2 \sqrt{T})\right)^k$$

Now for a $C \geq 0$ we first look at

$$\ln\frac{C}{\varepsilon_{\min}} = \ln C + \ln\frac{C_1 + \sqrt{C_1^2 + 4TC_2}}{2C_2} - k\ln\ln\frac{C_3(C_1 + \sqrt{C_1^2 + 4TC_2})}{2C_2}$$

$$\leq \ln C + \ln\frac{C_1 + \sqrt{C_1^2 + 4TC_2}}{2C_2}$$

$$\leq \ln C + \ln C_1 + \ln C_1 + \ln\frac{\sqrt{4TC_2}}{2C_2}$$

$$\leq \ln(CC_1^2\sqrt{T})$$

where the first inequality follows from the fact that $\frac{C_3(C_1+\sqrt{C_1^2+4TC_2})}{2C_2} \geq \frac{C_3 2C_1}{2C_2} \geq e$. Hence, we can bound

$$C_1 \left( \ln \frac{C_3}{\varepsilon_{\min}} \right)^k \ln \frac{H}{\varepsilon_{\min}} \leq C_1 \left( \ln(C_3 C_1^2 \sqrt{T}) \right)^k \ln(HC_1^2\sqrt{T}).$$

Now since

$$\frac{1}{\varepsilon_{\min}} = \frac{C_1 + \sqrt{C_1^2 + 4TC_2}}{2C_2} \left( \ln \frac{C_3(C_1 + \sqrt{C_1^2 + 4TC_2})}{2C_2} \right)^{-k} \leq \frac{C_1}{C_2} + \sqrt{\frac{T}{C_2}}$$

we get

$$C_2 \left( \ln \frac{C_3}{\varepsilon_{\min}} \right)^{2k} \left[ \frac{1}{\varepsilon_{\min}} - \frac{1}{H} \right] \leq C_2 \left( \ln(C_3 C_1^2 \sqrt{T}) \right)^{2k} \left[ \frac{C_1}{C_2} + \sqrt{\frac{T}{C_2}} \right]$$

$$\leq \left( \ln(C_3 C_1^2 \sqrt{T}) \right)^{2k} \left[ C_1 + \sqrt{TC_2} \right].$$

As a result we can conclude that $R(T) \leq (\sqrt{C_2 T} + C_1) \operatorname{polylog}(T, C_3, C_1, H) = O(\sqrt{C_2 T} \operatorname{polylog}(T, C_3, C_1, H))$. □

## B  Experimental Details

We generated the MDPs with $S = 5, 50, 200$ states, $A = 3$ actions and $H = 10$ timesteps as follows: The transition probabilities $P(s, a, t)$ were sampled independently from Dirichlet $\left( \frac{1}{10}, \ldots \frac{1}{10} \right)$ and the rewards were all deterministic with their value $r(s, a, t)$ set to 0 with probability $85\%$ and set uniformly at random in $[0, 1]$ otherwise. This construction results in MDPs that have concentrated but non-deterministic transition probabilities and sparse rewards.

Since some algorithms have been proposed assuming the rewards $r(s, a, t)$ are known and we aim for a fair comparison, we assumed for all algorithms that the immediate rewards $r(s, a, t)$ are known and adapted the algorithms accordingly. For example, in UBEV, the $\min \left\{ 1, \frac{l(s,a,t)}{\max\{1, n(s,a,t)\}} + \phi \right\}$ term was replaced by the true known rewards $r(s, a, t)$ and the $\delta$ parameter in $\phi$ was scaled by $9/7$ accordingly since the concentration result for immediate rewards is not necessary in this case. We used $\delta = \frac{1}{10}$ for all algorithms and $\varepsilon = \frac{1}{10}$ if they require to know $\varepsilon$ beforehand.

We adapted MoRMax, UCRL2, UCFH, MBIE, MedianPAC, Delayed Q-Learning and OIM to the episodic MDP setting with time-dependent transition dynamics by using allowing them to learn time-dependent dynamics and use finite-horizon planning. We did adapt the confidence intervals and but did not re-derive the constants for each algorithm. When in doubt we opted for smaller constants typically resulting better performance of the competitors. We further replaced the range of the value function $O(H)$ by the observed range of the optimistic next state values in the confidence bounds. We also reduced the number of episodes used in the delays by a factor of $\frac{1}{1000}$ for MoRMax and Delayed Q-Learning and by $10^{-6}$ for UCFH because they would otherwise not have performed a single policy update even for $S = 5$ within the 10 million episodes we considered. This scaling violates their theoretical guarantees but at least shows that the methods work in principle.

The performance reported in Figure 2 are the expected return of the current policy of each algorithm averaged over 1000 episodes. The figure shows a single run of the same randomly generated MDP but the results are representative. We reran this experiments with different random seeds and consistently obtained qualitatively similar results.

Source code for the experiments including concise but efficient implementations of the algorithms is available at <https://github.com/chrodan/FiniteEpisodicRL.jl>.

## C  PAC Lower Bound

**Theorem C.1.** *There exist positive constants* $c$, $\delta_0 > 0$, $\varepsilon_0 > 0$ *such that for every* $\varepsilon \in (0, \varepsilon_0)$, $S \geq 4, A \geq 2$ *and for every algorithm A that and* $n \leq \frac{cASH^3}{\varepsilon^2}$ *there is a fixed-horizon episodic MDP*

$M_{hard}$ with time-dependent transition probabilities and $S$ states and $A$ actions so that returning an $\varepsilon$-optimal policy after $n$ episodes is at most $1 - \delta_0$. That implies that no algorithm can have a PAC guarantee better than $\Omega\left(\frac{ASH^3}{\varepsilon^2}\right)$ for sufficiently small $\varepsilon$.

Note that this lower bound on the sample complexity of any method in episodic MDPs with time-dependent dynamics applies to the arbitrary but fixed $\varepsilon$ PAC bound and therefore immediately to the stronger uniform-PAC bounds. This theorem can be proved in the same way as Theorem 5 by Jiang et al. [4], which itself is a standard construction involving a careful layering of difficult instances of the multi-armed bandit problem.[4] For simplicity, we omitted the dependency on the failure probability $\delta$, but using the techniques in the proof of Theorem 26 by Strehl et al. [5], a lower bound of order $\Omega\left(\frac{ASH^3}{\varepsilon^2}\log(SA/\delta)\right)$ can be obtained. The lower bound shows for small $\varepsilon$ the sample complexity of UBEV given in Theorem 4 is optimal except for a factor of $H$ and logarithmic terms.

# D Planning Problem of UBEV

**Lemma D.1** (Planning Problem). *The policy update in Lines 3–9 of Algorithm 1 finds an optimal solution to the optimization problem*

$$\max_{P',V',\pi',r'} \mathbb{E}_{s\sim p_0}[V_1'(s)]$$

$$\forall s \in \mathcal{S}, a \in \mathcal{A}, t \in [H]:$$

$$V_{H+1}' = 0, \qquad P'(s,a,t) \in \Delta_S, \qquad r'(s,a,t) \in [0,1]$$

$$V_t'(s) = r'(s,\pi'(s,t),t) + \mathbb{E}_{s'\sim P'(s,\pi'(s,t),t)}[V_{t+1}']$$

$$|(P'(s,a,t) - \hat{P}_k(s,a,t))^\top V_{t+1}'| \le \phi(s,a,t)(H-t)$$

$$|r'(s,a,t) - \hat{r}_k(s,a,t)| \le \phi(s,a,t)$$

*where $\phi(s,a,t) = \sqrt{\frac{2\,\mathrm{llnp}(n(s,a,t)) + \ln(18SAH/\delta)}{n(s,a,t)}}$ is a confidence bound and $\hat{P}_k(s'|s,a,t) = m(s',s,a,t)/n(s,a,t)$ are the empirical transition probabilities and $\hat{r}_k(s,a,t) = l(s,a,t)/n(s,a,t)$ the empirical average rewards.*

*Proof.* Since $\tilde{V}_{H+1}(\cdot)$ is initialized with 0 and never changed, we immediately get that it is an optimal value for $V_{H+1}'(\cdot)$ which is constrained to be 0. Consider now a single time step $t$ and assume $V_{t+1}'$ are fixed to the optimal values $\tilde{V}_{t+1}$. Plugging in the computation of $Q(a)$ into the computation of $\tilde{V}_t(s)$, we get

$$\tilde{V}_t(s) = \max_a Q(a) = \max_{a\in\mathcal{A}}\Bigg[\min\left\{1, \hat{r}(s,a,t) + \phi(s,a,t)\right\}$$

$$+ \min\left\{\max \tilde{V}_{t+1}, \mathbb{I}\{n(s,a,t) > 0\}(\hat{P}(s,a,t)^\top \tilde{V}_{t+1}) + \phi(s,a,t)(H-t)\right\}\Bigg]$$

using the convention that $\hat{r}(s,a,t) = 0$ if $n(s,a,t) = 0$. Assuming that $V_{t+1}' = \tilde{V}_{t+1}$, and that our goal for now is to maximize $\tilde{V}_t(s)$, this can be rewritten as

$$\max_{P'(s,a,t),r'(s,a,t)} \tilde{V}_t(s) = \max_{P'(s,a,t),r'(s,a,t),\pi'(s,t)}\left[r'(s,\pi'(s,t),t) + P'(s,\pi'(s,t),t)^\top \tilde{V}_{t+1}\right]$$

$$\text{s.t.} \qquad \forall a \in \mathcal{A}: r'(s,a,t) \in [0,1], \qquad P'(s,a,t) \in \Delta_S$$

$$|(P'(s,a,t) - \hat{P}_k(s,a,t))^\top V_{t+1}'| \le \phi(s,a,t)(H-t)$$

$$|r'(s,a,t) - \hat{r}_k(s,a,t)| \le \phi(s,a,t)$$

since in this problem either $P'(s,\pi'(s,t),t)^\top \tilde{V}_{t+1} = \hat{P}(s,\pi'(s,t),t)^\top \tilde{V}_{t+1} + \phi(s,a,t)(H-t)$ if that does not violate $P'(s,\pi'(s,t),t)^\top \tilde{V}_{t+1} \le \max \tilde{V}_{t+1}$ and otherwise $P'(s',s,\pi'(s,t),t) = 1$

for one state $s'$ with $\tilde{V}_{t+1}(s') = \max \tilde{V}_{t+1}$. Similarly, either $r'(s, \pi'(s,t), t) = \hat{r}(s, \pi'(s,t), t) + \phi(s, \pi'(s,t), t)$ if that does not violate $r'(s, \pi'(s,t), t) \leq 1$ or $r'(s, \pi'(s,t), t) = 1$ otherwise. Using induction for $t = H, H-1 \ldots 1$, we see that UBEV computes an optimal solution to

$$
\max_{P', V', \pi', r'} V_1'(\tilde{s})
$$

$$
\forall s \in \mathcal{S}, a \in \mathcal{A}, t \in [H] :
$$
$$
V_{H+1}' = 0, \qquad P'(s,a,t) \in \Delta_S, \qquad r'(s,a,t) \in [0,1]
$$
$$
V_t'(s) = r'(s, \pi'(s,t), t) + \mathbb{E}_{s' \sim P'(s, \pi'(s,t), t)}[V_{t+1}']
$$
$$
|(P'(s,a,t) - \hat{P}_k(s,a,t))^\top V_{t+1}'| \leq \phi(s,a,t)(H-t)
$$
$$
|r'(s,a,t) - \hat{r}_k(s,a,t)| \leq \phi(s,a,t)
$$

for any fixed $\tilde{s}$. The intersection of all optimal solutions to this problem for all $\tilde{s} \in \mathcal{S}$ are also an optimal solution to

$$
\max_{P', V', \pi', r'} p_0^\top V_1'
$$

$$
\forall s \in \mathcal{S}, a \in \mathcal{A}, t \in [H] :
$$
$$
V_{H+1}' = 0, \qquad P'(s,a,t) \in \Delta_S, \qquad r'(s,a,t) \in [0,1]
$$
$$
V_t'(s) = r'(s, \pi'(s,t), t) + \mathbb{E}_{s' \sim P'(s, \pi'(s,t), t)}[V_{t+1}']
$$
$$
|(P'(s,a,t) - \hat{P}_k(s,a,t))^\top V_{t+1}'| \leq \phi(s,a,t)(H-t)
$$
$$
|r'(s,a,t) - \hat{r}_k(s,a,t)| \leq \phi(s,a,t).
$$

Hence, UBEV computes an optimal solution to this problem. $\qquad\square$

## E   Details of PAC Analysis

In the analysis, we denote the value of $n(\cdot, t)$ after the planning in iteration $k$ as $n_{tk}(\cdot)$. We further denote by $P(s'|s,a,t)$ the probability of sampling state $s'$ as $s_{t+1}$ when $s_t = s, a_t = a$. With slight abuse of notation, $P(s,a,t) \in [0,1]^S$ denotes the probability vector of $P(\cdot|s,a,t)$. We further use $\tilde{P}_k(s'|s,a,t)$ as conditional probability of $s_{t+1} = s'$ given $s_t = s, a_t = a$ but in the optimistic MDP $\tilde{M}$ computed in the optimistic planning steps in iteration $k$. We also use the following definitions:

$$
w_{\min} = w_{\min}' = \frac{\varepsilon c_\varepsilon}{H^2 S}
$$
$$
c_\varepsilon = \frac{1}{3}
$$
$$
L_{tk} = \{(s,a) \in \mathcal{S} \times \mathcal{A} : w_{tk}(s,a) \geq w_{\min}\}
$$
$$
\mathrm{llnp}(x) = \ln(\ln(\max\{x, e\}))
$$
$$
\mathrm{rng}(x) = \max(x) - \min(x)
$$
$$
\delta' = \frac{\delta}{9}
$$

In the following, we provide the formal proof for Theorem 4 and then present all necessary lemmas:

### E.1   Proof of Theorem 4

*Proof of Theorem 4.* Corollary E.5 ensures that the failure event has probability at most $\delta$. Outside the failure event Lemma E.2 ensures that all but at most $\frac{48A^2 S^3 H^4}{\varepsilon}$ polylog$(A, S, H, 1/\varepsilon, 1/\delta)$ episodes are friendly. Finally, Lemma E.8 shows that all friendly episodes except at most $\left(\frac{9216}{\varepsilon} + 417S\right) \frac{ASH^4}{\varepsilon}$ polylog$(A, S, H, 1/\varepsilon, 1/\delta)$ are $\varepsilon$-optimal. The second bound follows from replacing $AS^2$ by $1/\varepsilon$ in the second term. Furthermore, outside the failure event Lemma E.2 ensures that all but at most $\frac{6AS^2 H^3}{\varepsilon}$ polylog$(A, S, H, 1/\varepsilon, 1/\delta)$ episodes are nice. Finally, Lemma E.7 shows that all nice episodes except at most $(4 + S) 576 \frac{ASH^4}{\varepsilon}$ polylog$(A, S, H, 1/\varepsilon, 1/\delta)$ are $\varepsilon$-optimal.

$\qquad\square$

## E.2 Failure Events and Their Probabilities

In this section, we define a failure event $F$ in which we cannot guarantee the performance of UBEV. We then show that this event $F$ only occurs with low probability. All our arguments are based on general uniform concentration of measure statements that we prove in Section F. In the following we argue how the apply in our setting and finally combine all concentration results to get $\mathbb{P}(F) \leq \delta$. The failure event is defined as

$$F = \bigcup_k \left[ F_k^N \cup F_k^{CN} \cup F_k^P \cup F_k^V \cup F_k^{L1} \cup F_k^R \right]$$

where

$$F_k^N = \left\{ \exists s, a, t : n_{tk}(s, a) < \frac{1}{2} \sum_{i<k} w_{ti}(s, a) - \ln \frac{SAH}{\delta'} \right\}$$

$$F_k^{CN} = \left\{ \exists s, a, s', a', u < t : n_{tk}(s, a) < \frac{1}{2} n_{uk}(s', a') \sum_{i<k} w_{ui}^t(s, a | s', a') - \ln \left( \frac{S^2 A^2 H^2}{\delta'} \right) \right\}$$

$$F_k^V = \left\{ \exists s, a, t : |(\hat{P}_k(s, a, t) - P(s, a, t))^\top V_{t+1}^\star| \geq \sqrt{\frac{\mathrm{rng}(V_{t+1}^\star)^2}{n_{tk}(s, a)} \left( 2 \, \mathrm{llnp}(n_{tk}(s, a)) + \ln \frac{3SAH}{\delta'} \right)} \right\}$$

$$F_k^P = \left\{ \exists s, s', a, t : |\hat{P}_k(s'|s, a, t) - P(s'|s, a, t)| \geq \sqrt{\frac{2P(s'|s, a, t)}{n_{tk}(s, a)} \left( 2 \, \mathrm{llnp}(n_{tk}(s, a)) + \ln \frac{3S^2 AH}{\delta'} \right)} \right.$$

$$\left. + \frac{1}{n_{tk}(s, a)} \left( 2 \, \mathrm{llnp}(n_{tk}(s, a)) + \ln \frac{3S^2 AH}{\delta'} \right) \right\}$$

$$F_k^{L1} = \left\{ \exists s, a, t : \|\hat{P}_k(s, a, t) - P(s, a, t)\|_1 \geq \sqrt{\frac{4}{n_{tk}(s, a)} \left( 2 \, \mathrm{llnp}(n_{tk}(s, a)) + \ln \frac{3SAH(2^S - 2)}{\delta'} \right)} \right\}$$

$$F_k^R = \left\{ \exists s, a, t : |\hat{r}_k(s, a, t) - r(s, a, t)| \geq \sqrt{\frac{1}{n_{tk}(s, a)} \left( 2 \, \mathrm{llnp}(n_{tk}(s, a)) + \ln \frac{3SAH}{\delta'} \right)} \right\}.$$

We now bound the probability of each type of failure event individually:

**Corollary E.1.** *For any $\delta' > 0$, it holds that* $\mathbb{P}\left(\bigcup_{k=1}^\infty F_k^V\right) \leq 2\delta'$ *and* $\mathbb{P}\left(\bigcup_{k=1}^\infty F_k^R\right) \leq 2\delta'$

*Proof.* Consider a fix $s \in \mathcal{S}, a \in \mathcal{A}, t \in [H]$ and denote $\mathcal{F}_k$ the sigma-field induced by the first $k-1$ episodes and the $k$-th episode up to $s_t$ and $a_t$ but not $s_{t+1}$. Define $\tau_i$ to be the index of the episode where $(s, a)$ was observed at time $t$ the $i$th time. Note that $\tau_i$ are stopping times with respect to $\mathcal{F}_i$. Define now the filtration $\mathcal{G}_i = \mathcal{F}_{\tau_i} = \{ A \in \mathcal{F}_\infty : A \cap \{\tau_i \leq t\} \in \mathcal{F}_t \, \forall t \geq 0 \}$ and $X_k = (V_{t+1}^\star(s_k') - P(s, a, t)^\top V_{t+1}^\star) \mathbb{I}\{\tau_k < \infty\}$ where $s_i'$ is the value of $s_{t+1}$ in episode $\tau_i$ (or arbitrary, if $\tau_i = \infty$).

By the Markov property of the MDP, we have that $X_i$ is a martingale difference sequence with respect to the filtration $\mathcal{G}_i$. Further, since $\mathbb{E}[X_i | \mathcal{G}_{i-1}] = 0$ and $|X_i| \in [0, \mathrm{rng}(V_{t+1}^\star)]$, $X_i$ conditionally $\mathrm{rng}(V_{t+1}^\star)/2$-subgaussian due to Hoeffding's Lemma, i.e., satisfies $\mathbb{E}[\exp(\lambda X_i) | \mathcal{G}_{i-1}] \leq \exp(\lambda^2 \, \mathrm{rng}(V_{t+1}^\star)^2/2)$.

We can therefore apply Lemma F.1 and conclude that

$$\mathbb{P}\left( \exists k : |(\hat{P}_k(s, a, t) - P(s, a, t))^\top V_{t+1}^\star| \geq \sqrt{\frac{\mathrm{rng}(V_{t+1}^\star)^2}{n_{tk}(s, a)} \left( 2 \, \mathrm{llnp}(n_{tk}(s, a)) + \ln \frac{3}{\delta'} \right)} \right) \leq 2\delta'.$$

Analogously

$$\mathbb{P}\left( \exists k : |\hat{r}_k(s, a, t) - r(s, a, t)| \geq \sqrt{\frac{1}{n_{tk}(s, a)} \left( 2 \, \mathrm{llnp}(n_{tk}(s, a)) + \ln \frac{3}{\delta'} \right)} \right) \leq 2\delta'.$$

Applying the union bound over all $s \in \mathcal{S}, a \in \mathcal{A}$ and $t \in [H]$, we obtain the desired statement for $F^V$. In complete analogy using the same filtration, we can show the statement for $F^R$. $\qquad \square$

**Corollary E.2.** *For any $\delta' > 0$, it holds that $\mathbb{P}\left(\bigcup_{k=1}^{\infty} F_k^P\right) \leq 2\delta'$.*

*Proof.* Consider first a fix $s', s \in \mathcal{S}, t \in [H]$ and $a \in \mathcal{A}$. Let $K$ denote the number of times the triple $s, a, t$ was encountered in total during the run of the algorithm. Define the random sequence $X_i$ as follows. For $i \leq K$, let $X_i$ be the indicator of whether $s'$ was the next state when $s, a, t$ was encountered the $i$th time and for $i > K$, let $X_i \sim \text{Bernoulli}(P(s'|s, a, t))$ be drawn i.i.d. By construction this is a sequence of i.i.d. Bernoulli random variables with mean $P(s'|s, a, t)$. Further the event

$$\bigcup_k \left\{ \left| \hat{P}_k(s'|s, a, t) - P(s'|s, a, t) \right| \geq \sqrt{\frac{2P(s'|s, a, t)}{n_{tk}(s, a)} \left( 2 \operatorname{llnp}(n(s, a, t)) + \ln \frac{3S^2 AH}{\delta'} \right)} \right.$$
$$\left. + \frac{1}{n_{tk}(s, a)} \left( 2 \operatorname{llnp}(n_{tk}(s, a)) + \ln \frac{3S^2 AH}{\delta'} \right) \right\}$$

is contained in the event

$$\bigcup_i \left\{ |\hat{\mu}_i - \mu| \geq \sqrt{\frac{2\mu}{i} \left( 2 \operatorname{llnp}(i) + \ln \frac{3}{\delta'} \right)} + \frac{1}{i} \left( 2 \operatorname{llnp}(i) + \ln \frac{3S^2 AH}{\delta'} \right) \right\}$$

whose probability can be bounded by $2\delta'/S^2/A/H$ using Lemma F.2. The statement now follows by applying the union bound. $\qquad \square$

**Corollary E.3.** *For any $\delta' > 0$, it holds that $\mathbb{P}\left(\bigcup_{k=1}^{\infty} F_k^{L1}\right) \leq \delta'$*

*Proof.* Using the same argument as in the proof of Corollary E.2 the statement follows from Lemma F.3. $\qquad \square$

**Corollary E.4.** *It holds that*

$$\mathbb{P}\left(\bigcup_k F_k^N\right) \leq \delta' \quad and \quad \mathbb{P}\left(\bigcup_k F_k^{CN}\right) \leq \delta'.$$

*Proof.* Consider a fix $s \in \mathcal{S}, a \in \mathcal{A}, t \in [H]$. We define $\mathcal{F}_k$ to be the sigma-field induced by the first $k - 1$ episodes and $X_k$ as the indicator whether $s, a, t$ was observed in episode $k$. The probability $w_{tk}(s, a)$ pf whether $X_k = 1$ is $F_k$ measurable and hence we can apply Lemma F.4 with $W = \ln \frac{SAH}{\delta'}$ and obtain that $\mathbb{P}\left(\bigcup_k F_k^N\right) \leq \delta'$ after applying the union bound.

For the second statement, consider again a fix $s, s' \in \mathcal{S}, a, a' \in \mathcal{A}, u, t \in [H]$ with $u < t$ and denote by $\mathcal{F}_k$ the sigma-field induced by the first $k - 1$ episodes and the $k$-th episode up to $s_u$ and $a_u$ but not $s_{u+1}$. Define $\tau_i$ to be the index of the episode where $(s', a')$ was observed at time $u$ the $i$th time. Note that $\tau_i$ are stopping times with respect to $\mathcal{F}_i$. Define now the filtration $\mathcal{G}_i = \mathcal{F}_{\tau_i} = \{A \in \mathcal{F}_\infty : A \cap \{\tau_i \leq k\} \in \mathcal{F}_k \, \forall k \geq 0\}$ and $X_i$ to be the indicator whether $s, a, t$ and $s', a', u$ was observed in episode $\tau_i$. If $\tau_i = \infty$, we set $X_i = 0$. Note that the probablity $w_{ui}^t(s, a|s', a')\mathbb{I}\{\tau_i < \infty\}$ of $X_i = 1$ is $\mathcal{G}_i$-measureable.

By the Markov property of the MDP, we have that $X_i$ is a martingale difference sequence with respect to the filtration $\mathcal{G}_i$. We can therefore apply Lemma F.4 with $W = \ln \frac{S^2 A^2 H^2}{\delta'}$ and using the union bound over all $s, a, s', a', u, t$, we get $\mathbb{P}\left(\bigcup_k F_k^{CN}\right) \leq \delta'$. $\qquad \square$

**Corollary E.5.** *The total failure probability of the algorithm is bounded by $\mathbb{P}(F) \leq 9\delta' = \delta$.*

*Proof.* Statement follows directly from Corollary E.1, Corollary E.2, Corollary E.3, Corollary E.4 and the union bound. $\qquad \square$

### E.3 Nice and Friendly Episodes

We now define the notion of *nice* and the stronger *friendly* episodes. In nice episodes, all states either have low probability of occuring or the sum of probability of occuring in the previous episodes is large enough so that outside the failure event we can guarantee that

$$n_{tk}(s,a) \geq \frac{1}{4} \sum_{i<k} w_{ti}(s,a).$$

This allows us to then bound the number of nice episodes by the number of times terms of the form

$$\sum_{t=1}^{H} \sum_{s,a \in L_{tk}} w_{tk}(s,a) \sqrt{\frac{\mathrm{llnp}(n_{tk}(s,a)) + D}{n_{tk}(s,a)}}$$

can exceed a chosen threshold (see Lemma E.3 below). In the next section, we will bound the optimality gap of an episode by terms of such form and use the results derived here to bound the number of nice episodes where the algorithm can follow a $\varepsilon$-suboptimal policy. Together with a bound on the number of non-nice episodes, we obtain the sample complexity of UBEV shown in Theorem 4.

Similarly, we use a more refined analysis of the optimality gap of friendly episodes together with Lemma E.4 below to obtain the tighter sample complexity linear-polylog in $S$.

**Definition 2** (Nice and Friendly Episodes). *An episode $k$ is* nice *if and only if for all $s \in \mathcal{S}$, $a \in \mathcal{A}$ and $t \in [H]$ the following two conditions hold:*

$$w_{tk}(s,a) \leq w_{\min} \quad \lor \quad \frac{1}{4} \sum_{i<k} w_{ti}(s,a) \geq \ln \frac{SAH}{\delta'}$$

*An episode $k$ is* friendly *if and only if it is nice and for all $s, s' \in \mathcal{S}$, $a, a' \in \mathcal{A}$ and $u, t \in [H]$ with $u < t$ the following two conditions hold:*

$$w_{uk}^t(s,a|s',a') \leq w'_{\min} \quad \lor \quad \frac{1}{4} \sum_{i<k} w_{ui}^t(s,a|s',a') \geq \ln \frac{S^2 A^2 H^2}{\delta'}.$$

*We denote the set of all nice episodes by $N \subseteq \mathbb{N}$ and the set of all friendly episodes by $K \subseteq N$.*

**Lemma E.1** (Properties of nice and friendly episodes). *If an episode $k$ is nice, i.e., $k \in N$, then on $F^c$ (outside the failure event) for all $s \in \mathcal{S}$, $a \in \mathcal{A}$ and $t \in [H]$ with $u < t$ the following statement holds:*

$$w_{tk}(s,a) \leq w_{\min} \quad \lor \quad n_{tk}(s,a) \geq \frac{1}{4} \sum_{i<k} w_{ti}(s,a).$$

*If an episode $k$ is friendly, i.e., $k \in K$, then on $F^c$ (outside the failure event) for all $s, s' \in \mathcal{S}$, $a, a' \in \mathcal{A}$ and $u, t \in [H]$ with $u < t$ the above statement holds as well as*

$$w_{uk}^t(s,a|s',a') \leq w'_{\min} \quad \lor \quad n_{tk}(s,a) \geq \frac{1}{4} n_{uk}(s',a') \sum_{i<k} w_{ui}^t(s,a|s',a').$$

*Proof.* Since we consider the event $F_k^{N^c}$, it holds for all $s, a, t$ triples with $w_{tk}(s,a) > w_{\min}$

$$n_{tk}(s,a) \geq \frac{1}{2} \sum_{i<k} w_{ti}(s,a) - \ln \frac{SAH}{\delta'} \geq \frac{1}{4} \sum_{i<k} w_{ti}(s,a)$$

for $k \in N$ Further, since we only consider the event $F_k^{CN^c}$, we have for all $s, s' \in \mathcal{S}$, $a, a' \in \mathcal{A}$, $u, t \in [H]$ with $u < t$ and $w_{uk}^t(s,a|s',a') > w_{\min}$

$$n_{tk}(s,a) \geq \frac{1}{2} n_{uk}(s',a') \sum_{i<k} w_{ui}^t(s,a|s',a') - \ln \frac{S^2 A^2 H^2}{\delta'}$$

for $k \in E$. If $n_{uk}(s', a') = 0$ then $n_{tk}(s, a) \geq 0 = \frac{1}{4} n_{uk}(s', a') \sum_{i<k} w_{ui}^t(s, a|s', a')$ holds trivially. Otherwise $n_{uk}(s', a') \geq 1$ and therefore

$$
\begin{aligned}
n_{tk}(s, a) &\geq \frac{1}{2} n_{uk}(s', a') \sum_{i<k} w_{ui}^t(s, a|s', a') - \ln \frac{S^2 A^2 H^2}{\delta'} \\
&\geq \frac{1}{2} n_{uk}(s', a') \sum_{i<k} w_{ui}^t(s, a|s', a') - \frac{1}{4} \sum_{i<k} w_{ui}^t(s, a|s', a') \\
&\geq \frac{1}{4} n_{uk}(s', a') \sum_{i<k} w_{ui}^t(s, a|s', a')
\end{aligned}
$$

$\square$

**Lemma E.2** (Number of non-nice and non-friendly episodes)**.** *On the good event $F^c$, the number of episodes that are not friendly is at most*

$$
48 \frac{S^3 A^2 H^4}{\varepsilon} \ln \frac{S^2 A^2 H^2}{\delta'}
$$

*and the number episodes that are not nice is at most*

$$
\frac{6 S^2 A H^3}{\varepsilon} \ln \frac{S A H}{\delta'}.
$$

*Proof.* If an episode $k$ is not nice, then there is $s, a, t$ with $w_{tk}(s, a) > w_{\min}$ and $\sum_{i<k} w_{ti}(s, a) < 4 \ln \frac{S A H}{\delta'}$. Since the sum on the left-hand side of this inequality increases by at least $w_{\min}$ when this happens and the right hand side stays constant, this situation can occur at most

$$
\frac{4 S A H}{w_{\min}} \ln \frac{S A H}{\delta'} = \frac{24 S^2 A H^3}{\varepsilon} \ln \frac{S A H}{\delta'}
$$

times in total. If an episode $k$ is not friendly, it is either not nice or there is $s, a, t$ and $s', a', u$ with $u < t$ and $w_{uk}^t(s', a'|s, a) > w'_{\min}$ and $\sum_{i<k} w_{ui}^t(s, a|s', a') < 4 \ln \frac{S^2 A^2 H^2}{\delta'}$. Since the sum on the left-hand side of this inequality increases by at least $w'_{\min}$ each time this happens while the right hand side stays constant, this can happen at most $\frac{4 S^2 A^2 H^2}{w'_{\min}} \ln \frac{S^2 A^2 H^2}{\delta'}$ times in total. Therefore, there can only be at most

$$
\begin{aligned}
&\frac{4 S A H}{w_{\min}} \ln \frac{S A H}{\delta'} + \frac{4 S^2 A^2 H^2}{w'_{\min}} \ln \frac{S^2 A^2 H^2}{\delta'} \\
&= \frac{4 S^2 A H^3}{c_\varepsilon \varepsilon} \ln \frac{S A H}{\delta'} + \frac{4 S^3 A^2 H^4}{c_\varepsilon \varepsilon} \ln \frac{S^2 A^2 H^2}{\delta'} \leq \frac{48 S^3 A^2 H^4}{\varepsilon^2} \ln \frac{S^2 A^2 H^2}{\delta'}
\end{aligned}
$$

non-friendly episodes. $\square$

**Lemma E.3** (Main Rate Lemma)**.** *Let $r \geq 1$ fix and $C > 0$ which can depend polynomially on the relevant quantities and $\varepsilon' > 0$ and let $D \geq 1$ which can depend poly-logarithmically on the relevant quantities. Then*

$$
\sum_t \sum_{s, a \in L_{tk}} w_{tk}(s, a) \left( \frac{C(\mathrm{llnp}(n_{tk}(s, a)) + D)}{n_{tk}(s, a)} \right)^{1/r} \leq \varepsilon'
$$

*on all but at most*

$$
\frac{8 C A S H^r}{\varepsilon'^r} \mathrm{polylog}(S, A, H, \delta^{-1}, \varepsilon'^{-1}).
$$

*nice episodes.*

*Proof.* Define

$$\Delta_k = \sum_t \sum_{s,a \in L_{tk}} w_{tk}(s,a) \left( \frac{C(\text{llnp}(n_{tk}(s,a)) + D)}{n_{tk}(s,a)} \right)^{1/r}$$

$$= \sum_t \sum_{s,a \in L_{tk}} w_{tk}(s,a)^{1-\frac{1}{r}} \left( w_{tk}(s,a) \frac{C(\text{llnp}(n_{tk}(s,a)) + D)}{n_{tk}(s,a)} \right)^{1/r}.$$

We first bound using Hölder's inequality

$$\Delta_k \leq \left( \sum_t \sum_{s,a \in L_{tk}} \frac{CH^{r-1} w_{tk}(s,a)(\text{llnp}(n_{tk}(s,a)) + D)}{n_{tk}(s,a)} \right)^{\frac{1}{r}}.$$

Using the property in Lemma E.1 of nice episodes as well as the fact that $w_{tk}(s,a) \leq 1$ and $\sum_{i<k} w_{ti}(s,a) \geq 4 \ln \frac{SAH}{\delta'} \geq 4\ln(2) \geq 2$, we bound

$$n_{tk}(s,a) \geq \frac{1}{4} \sum_{i<k} w_{ti}(s,a) \geq \frac{1}{8} \sum_{i \leq k} w_{ti}(s,a).$$

The function $\frac{\text{llnp}(x)+D}{x}$ is monotonically decreasing in $x \geq 0$ since $D \geq 1$ (see Lemma E.6). This allows us to bound

$$\Delta_k^r \leq \sum_t \sum_{s,a \in L_{tk}} \frac{CH^{r-1} w_{tk}(s,a)(\text{llnp}(n_{tk}(s,a)) + D)}{n_{tk}(s,a)}$$

$$\leq 8CH^{r-1} \sum_t \sum_{s,a \in L_{tk}} \frac{w_{tk}(s,a) \left( \text{llnp}\left( \frac{1}{8} \sum_{i \leq k} w_{ti}(s,a) \right) + D \right)}{\sum_{i \leq k} w_{ti}(s,a)}$$

$$\leq 8CH^{r-1} \sum_t \sum_{s,a \in L_{tk}} \frac{w_{tk}(s,a) \left( \text{llnp}\left( \sum_{i \leq k} w_{ti}(s,a) \right) + D \right)}{\sum_{i \leq k} w_{ti}(s,a)}.$$

Assume now $\Delta_k > \varepsilon'$. In this case the right-hand side of the inequality above is also larger than $\varepsilon'^r$ and there is at least one $(s,a,t)$ with $w_{tk}(s,a) > w_{\min}$ and

$$\frac{8CSAH^r \left( \text{llnp}\left( \sum_{i \leq k} w_{ti}(s,a) \right) + D \right)}{\sum_{i \leq k} w_{ti}(s,a)} > \varepsilon'^r$$

$$\Leftrightarrow \frac{\text{llnp}\left( \sum_{i \leq k} w_{ti}(s,a) \right) + D}{\sum_{i \leq k} w_{ti}(s,a)} > \frac{\varepsilon'^r}{8CSAH^r}.$$

Let us denote $C' = \frac{8CASH^r}{\varepsilon'^r}$. Since $\frac{\text{llnp}(x)+D}{x}$ is monotonically decreasing and $x = C'^2 + 3C'D$ satisfies $\frac{\text{llnp}(x)+D}{x} \leq \frac{\sqrt{x}+D}{x} \leq \frac{1}{C'}$, we know that if $\sum_{i \leq k} w_{ti}(s,a) \geq C'^2 + 3C'D$ then the above condition cannot be satisfied for $s,a,t$. Since each time the condition is satisfied, it holds that $w_{tk}(s,a) > w_{\min}$ and so $\sum_{i \leq k} w_{ti}(s,a)$ increases by at least $w_{\min}$, it can happen at most

$$m \leq \frac{ASH(C'^2 + 3C'D)}{w_{\min}}$$

times that $\Delta_k > \varepsilon'$. Define $K = \{k : \Delta_k > \varepsilon'\} \cap N$ and we know that $|K| \leq m$. Now we consider the sum

$$\sum_{k \in K} \Delta_k^r \leq \sum_{k \in K} 8CH^{r-1} \sum_t \sum_{s,a \in L_{tk}} \frac{w_{tk}(s,a) \left( \text{llnp}\left( \sum_{i \leq k} w_{ti}(s,a) \right) + D \right)}{\sum_{i \leq k} w_{ti}(s,a)}$$

$$\leq 8CH^{r-1} \left( \text{llnp}\left( C'^2 + 3C'D \right) + D \right) \sum_t \sum_{s,a \in L_{tk}} \sum_{k \in K} \frac{w_{tk}(s,a)}{\sum_{i \leq k} w_{ti}(s,a) \mathbb{I}\{w_{ti}(s,a) \geq w_{\min}\}}$$

For every $(s, a, t)$, we consider the sequence of $w_{ti}(s, a) \in [w_{\min}, 1]$ with $i \in I = \{i \in \mathbb{N} : w_{ti}(s, a) \geq w_{\min}\}$ and apply Lemma E.5. This yields that

$$\sum_{k \in K} \frac{w_{tk}(s, a)}{\sum_{i \leq k} w_{ti}(s, a) \mathbb{I}\{w_{ti}(s, a) \geq w_{\min}\}} \leq 1 + \ln(m/w_{\min}) = \ln\left(\frac{me}{w_{\min}}\right)$$

and hence

$$\sum_{k \in K} \Delta_k^r \leq 8CASH^r \ln\left(\frac{me}{w_{\min}}\right)\left(\text{llnp}\left(C'^2 + 3C'D\right) + D\right)$$

Since each element in $K$ has to contribute at least $\varepsilon'^r$ to this bound, we can conclude that

$$\sum_{k \in N} \mathbb{I}\{\Delta_k \geq \varepsilon'\} \leq \sum_{k \in K} \mathbb{I}\{\Delta_k \geq \varepsilon'\} \leq |K| \leq \frac{8CASH^r}{\varepsilon'^r} \ln\left(\frac{me}{w_{\min}}\right)\left(\text{llnp}\left(C'^2 + 3C'D\right) + D\right).$$

Since $\ln\left(\frac{me}{w_{\min}}\right)\left(\text{llnp}\left(C'^2 + 3C'D\right) + D\right)$ is $\text{polylog}(S, A, H, \delta^{-1}, \varepsilon'^{-1})$, the proof is complete. $\qquad\square$

**Lemma E.4** (Conditional Rate Lemma). *Let $r \geq 1$ fix and $C > 0$ which can depend polynomially on the relevant quantities and $\varepsilon' > 0$ and let $D \geq 1$ which can depend poly-logarithmically on the relevant quantities. Further $T \subset [H]$ is a subset of time-indices with $u < t$ for all $t \in T$. Then*

$$\sum_{t \in T} \sum_{s, a \in L_k^{ut}} w_{uk}^t(s, a | s', a')\left(\frac{C(\text{llnp}(n_{tk}(s, a)) + D)}{n_{tk}(s, a)}\right)^{1/r} \leq \varepsilon'\left(\frac{\text{llnp}(n_{uk}(s', a')) + D + 1}{n_{uk}(s', a')}\right)^{1/r}$$

*on all but at most*

$$\frac{8CAS|T|^r}{\varepsilon'^r}\text{polylog}(S, A, H, \delta^{-1}, \varepsilon'^{-1}).$$

*friendly episodes E.*

*Proof.* The proof follows mainly the structure of Lemma E.3. For the sake of completeness, we still present all steps here. Define

$$\Delta_k = \sum_{t \in T} \sum_{s, a \in L_k^{ut}} w_{uk}^t(s, a | s', a')\left(\frac{C(\text{llnp}(n_{tk}(s, a)) + D)}{n_{tk}(s, a)}\right)^{1/r}$$

$$= \sum_{t \in T} \sum_{s, a \in L_k^{ut}} w_{uk}^t(s, a | s', a')^{1-1/r}\left(w_{uk}^t(s, a | s', a')\frac{C(\text{llnp}(n_{tk}(s, a)) + D)}{n_{tk}(s, a)}\right)^{1/r}.$$

We first bound using Hölder's inequality

$$\Delta_k \leq \left(\sum_{t \geq u} \sum_{s, a \in L_k^{ut}} w_{uk}^t(s, a | s', a')\frac{C|T|^{r-1}(\text{llnp}(n_{tk}(s, a)) + D)}{n_{tk}(s, a)}\right)^{\frac{1}{r}}$$

Using the property in Lemma E.1 of friendly episodes as well as the fact that $w_{uk}^t(s, a | s', a') \leq 1$ and $\sum_{i < k} w_{ui}^t(s, a | s', a') \geq 4 \ln \frac{S^2 A^2 H^2}{\delta'} \geq 4 \ln(2) \geq 2$, we bound

$$n_{tk}(s, a) \geq \frac{1}{4}n_{uk}(s', a')\sum_{i < k} w_{ui}^t(s, a | s', a') \geq \frac{1}{8}n_{uk}(s', a')\sum_{i \leq k} w_{ui}^t(s, a | s', a').$$

The function $\frac{\mathrm{llnp}(x)+D}{x}$ is monotonically decreasing in $x \geq 0$ since $D \geq 1$ (see Lemma E.6). This allows us to bound

$$\Delta_k^r \leq \sum_{t \in T} \sum_{s,a \in L_k^{ut}} w_{uk}^t(s,a|s',a') \frac{C|T|^{r-1}(\mathrm{llnp}(n_{tk}(s,a)) + D)}{n_{tk}(s,a)}$$

$$\leq 8C|T|^{r-1} \sum_{t \in T} \sum_{s,a \in L_k^{ut}} \frac{w_{uk}^t(s,a|s',a')(\mathrm{llnp}\left(\frac{1}{8} n_{uk}(s',a') \sum_{i \leq k} w_{ui}^t(s,a|s',a')\right) + D)}{n_{uk}(s',a') \sum_{i \leq k} w_{ui}^t(s,a|s',a')}$$

$$\leq 8C|T|^{r-1} \sum_{t \in T} \sum_{s,a \in L_k^{ut}} \frac{w_{uk}^t(s,a|s',a')(\mathrm{llnp}\left(\sum_{i \leq k} w_{ui}^t(s,a|s',a')\right) + \mathrm{llnp}(n_{uk}(s',a')) + D + 1)}{n_{uk}(s',a') \sum_{i \leq k} w_{ui}^t(s,a|s',a')},$$

where for the last line we used the first and last property in Lemma E.6. For notational convenience, we will use $D' = D + 1 + \mathrm{llnp}(n_{uk}(s',a'))$. Assume now $\Delta_k > \varepsilon' \left(\frac{D'}{n_{uk}(s',a')}\right)^{1/r}$. In this case the right-hand side of the inequality above is also larger than $\varepsilon'^r \left(\frac{D'}{n_{uk}(s',a')}\right)$ and there is at least one $(s,a,t)$ with $w_{uk}^t(s,a|s',a') > w_{\min}$ and

$$\frac{8CSA|T|^r \left(\mathrm{llnp}\left(\sum_{i \leq k} w_{ui}^t(s,a|s',a')\right) + D'\right)}{\sum_{i \leq k} w_{ui}^t(s,a|s',a')} > D'\varepsilon'^r$$

$$\Leftrightarrow \frac{\left(\mathrm{llnp}\left(\sum_{i \leq k} w_{ui}^t(s,a|s',a')\right) + D'\right)}{\sum_{i \leq k} w_{ui}^t(s,a|s',a')} > \frac{D'\varepsilon'^r}{8CSA|T|^r}.$$

Let us denote $C' = \frac{8CAS|T|^r}{\varepsilon'^r}$. Since $\frac{\mathrm{llnp}(x)+D'}{x}$ is monotonically decreasing and $x = C'^2 + 3C'$ satisfies $\frac{\mathrm{llnp}(x)+D'}{x} \leq \frac{\sqrt{x}+D'}{x} \leq D'\frac{\sqrt{x}+1}{x} \leq \frac{D'}{C'}$, we know that if $\sum_{i \leq k} w_{ui}^t(s,a|s',a') \geq C'^2 + 3C'$ then the above condition cannot be satisfied for $s,a,t$. Since each time the condition is satisfied, it holds that $w_{uk}^t(s,a|s',a') > w_{\min}$ and so $\sum_{i \leq k} w_{ui}^t(s,a|s',a')$ increases by at least $w_{\min}$, it can happen at most

$$m \leq \frac{AS|T|(C'^2 + 3C')}{w_{\min}}$$

times that $\Delta_k > \varepsilon' \left(\frac{D'}{n_{uk}(s',a')}\right)^{1/r}$. Define $K = \left\{k : \Delta_k > \varepsilon' \left(\frac{D'}{n_{uk}(s',a')}\right)^{1/r}\right\} \cap E$ and we know that $|K| \leq m$. Now we consider the sum

$$\sum_{k \in K} \Delta_k^r \leq \sum_{k \in K} 8C|T|^{r-1} \sum_{t \in T} \sum_{s,a \in L_k^{ut}} \frac{w_{uk}^t(s,a|s',a')(\mathrm{llnp}\left(\sum_{i \leq k} w_{ui}^t(s,a|s',a')\right) + D')}{n_{uk}(s',a') \sum_{i \leq k} w_{ui}^t(s,a|s',a')}$$

$$\leq \frac{8C|T|^{r-1}(\mathrm{llnp}\left(C'^2 + 3C'\right) + D')}{n_{uk}(s',a')} \sum_{t \in T} \sum_{s,a \in L_k^{ut}} \sum_{k \in K} \frac{w_{uk}^t(s,a|s',a')}{\sum_{i \leq k} w_{ui}^t(s,a|s',a')}$$

$$\leq \frac{8C|T|^{r-1}D'(\mathrm{llnp}\left(C'^2 + 3C'\right) + 1)}{n_{uk}(s',a')} \sum_{t \in T} \sum_{s,a \in L_k^{ut}} \sum_{k \in K} \frac{w_{uk}^t(s,a|s',a')}{\sum_{i \leq k} w_{ui}^t(s,a|s',a')\mathbb{I}\{w_{ui}^t(s,a|s',a') \geq w_{\min}\}}$$

For every $(s,a,t)$, we consider the sequence of $w_{ui}^t(s,a|s',a') \in [w_{\min}, 1]$ with $i \in I = \{i \in \mathbb{N} : w_{ui}^t(s,a|s',a') \geq w_{\min}\}$ and apply Lemma E.5. This yields that

$$\sum_{k \in K} \frac{w_{uk}^t(s,a|s',a')}{\sum_{i \leq k} w_{ui}^t(s,a|s',a')\mathbb{I}\{w_{ui}^t(s,a|s',a') \geq w_{\min}\}} \leq \ln\left(\frac{me}{w_{\min}}\right)$$

and hence

$$\sum_{k \in K} \Delta_k^r \leq \frac{8CAS|T|^r D'\left(\text{llnp}\left(C'^2 + 3C'\right) + 1\right)}{n_{uk}(s', a')} \ln\left(\frac{me}{w_{\min}}\right)$$

Since each element in $K$ has to contribute at least $\frac{D'\varepsilon'^r}{n_{uk}(s',a')}$ to this bound, we can conclude that

$$\sum_{k \in E} \mathbb{I}\{\Delta_k \geq \varepsilon'\} = \sum_{k \in K} \mathbb{I}\{\Delta_k \geq \varepsilon'\}$$

$$\leq |K| \leq \frac{8CAS|T|^r}{\varepsilon'^r} \ln\left(\frac{me}{w_{\min}}\right)\left(\text{llnp}\left(C'^2 + 3C'\right) + 1\right).$$

Since $\ln\left(\frac{me}{w_{\min}}\right)\left(\text{llnp}\left(C'^2 + 3C'\right) + 1\right)$ is polylog($S, A, H, \delta^{-1}, \varepsilon'^{-1}$), the proof is complete. $\quad\square$

**Lemma E.5.** *Let $a_i$ be a sequence taking values in $[a_{\min}, 1]$ with $a_{\min} > 0$ and $m > 0$, then*

$$\sum_{k=1}^{m} \frac{a_k}{\sum_{i=1}^{k} a_i} \leq \ln\left(\frac{me}{a_{\min}}\right).$$

*Proof.* Let $f$ be a step-function taking value $a_i$ on $[i-1, i)$ for all $i$. We have $F(t) := \int_0^t f(x)dx = \sum_{i=1}^{t} a_i$. By the fundamental theorem of Calculus, we can bound

$$\sum_{k=1}^{m} \frac{a_k}{\sum_{i=1}^{k} a_i} = \frac{a_1}{a_1} + \int_1^m \frac{f(x)}{F(x) - F(0)}dx = 1 + \ln F(m) - \ln F(1)$$

$$\leq 1 + \ln(m) - \ln a_{\min} = \ln\left(\frac{me}{a_{\min}}\right),$$

where the inequality follows from $a_1 \geq a_{\min}$ and $\sum_{i=1}^{m} a_i \leq m$. $\quad\square$

**Lemma E.6** (Properties of llnp). *The following properties hold:*

1. llnp *is continuous and nondecreasing.*

2. $f(x) = \frac{\text{llnp}(nx) + D}{x}$ *with $n \geq 0$ and $D \geq 1$ is monotonically decreasing on $\mathbb{R}_+$.*

3. $\text{llnp}(xy) \leq \text{llnp}(x) + \text{llnp}(y) + 1$ *for all $x, y \geq 0$.*

*Proof.*     1. For $x \leq e$ we have $\text{llnp}(x) = 0$ and for $x \geq e$ we have $\text{llnp}(x) = \ln(\ln(x))$ which is continuous and monotonically increasing and $\lim_{x \searrow e} \ln(\ln(x)) = 0$.

2. The function llnp is continuous as well as $1/x$ on $\mathbb{R}_+$ and therefore so it $f$. Further, $f$ is differentiable except at $x = e/n$. For $x \in [0, e/n)$, we have $f(x) = D/x$ with derivative $-D/x^2 < 0$. Hence $f$ is monotonically decreasing on $x \in [0, e/n)$. For $x > e/n$, we have $f(x) = \frac{\ln(\ln(nx)) + D}{x}$ with derivative

$$-\frac{D + \ln(\ln(nx))}{x^2} + \frac{1}{x^2 \ln(nx)} = \frac{1 - \ln(nx)(D + \ln(\ln(nx)))}{x^2 \ln(nx)}.$$

The denominator is always positive in this range so $f$ is monotonically decreasing if and only if $\ln(nx)(D - \ln(\ln(nx))) \geq 1$. Using $D \geq 1$, we have $\ln(nx)(D + \ln(\ln(nx))) \geq 1(1 + 0) = 1$.

3. First note that for $xy \leq e^e$ we have $\text{llnp}(xy) \leq 1 \leq \text{llnp}(x) + \text{llnp}(y) + 1$ and therfore the statement holds for $x, y \leq e$.

Then consider the case that $x, y \geq e$ and $\text{llnp}(x) + \text{llnp}(y) + 1 - \text{llnp}(xy) = \ln \ln x + \ln \ln y + 1 - \ln(\ln(x) + \ln(y)) = -\ln(a + b) + 1 + \ln(a) + \ln(b)$ where $a = \ln x \geq 1$ and $b = \ln y \geq 1$. The function $g(a, b) = -\ln(a + b) + 1 + \ln(a) + \ln(b)$ is continuous and

differentiable with $\frac{\partial g}{\partial a} = \frac{b}{a(a+b)} > 0$ and $\frac{\partial g}{\partial b} = \frac{a}{b(a+b)} > 0$. Therefore, $g$ attains its minimum on $[1,\infty) \times [1,\infty)$ at $a = 1, b = 1$. Since $g(1,1) = 1 - \ln(2) \geq 0$, the statement also holds for $x, y \geq e$.

Finally consider the case where $x \leq e \leq y$. Then $\mathrm{llnp}(xy) \leq \mathrm{llnp}(ey) = \ln(1 + \ln y) \leq \ln \ln y + 1 \leq \mathrm{llnp}(x) + \mathrm{llnp}(y) + 1$. Due to symmetry this also holds for $y \leq e \leq x$.

$\square$

## E.4  Decomposition of Optimality Gap

In this section we decompose the optimality gap and then bound each term individually. Finally, both rate lemmas presented in the previous section are used to determine a bound on the number of nice / friendly episodes where the optimality gap can be larger than $\varepsilon$. The decomposition in the following lemma is a the simpler version bounding the number of $\varepsilon$-suboptimal nice episodes and eventually lead to the first bound in Theorem 4.

**Lemma E.7** (Optimality Gap Bound On Nice Episodes). *On the good event $F^c$ it holds that $V_1^\star(s_0) - V_1^{\pi_k}(s_0) \leq \varepsilon$ on all nice episodes $k \in N$ except at most*

$$\frac{144(4 + 3H^2 + 4SH^2)ASH^2}{\varepsilon^2} \, \mathrm{polylog}(A, S, H, 1/\varepsilon, 1/\delta)$$

*episodes.*

*Proof.* Using optimism of the algorithm shown in Lemma E.16, we can bound

$$V_1^\star(s_0) - V_1^{\pi_k}(s_0)$$

$$\leq |\tilde{V}_1^{\pi_k}(s_0) - V_1^{\pi_k}(s_0)|$$

$$\leq \sum_{t=1}^H \sum_{s,a} w_{tk}(s,a)|(\tilde{P}_k(s,a,t) - P(s,a,t))^\top \tilde{V}_{t+1}^{\pi_k}| + \sum_{t=1}^H \sum_{s,a} w_{tk}(s,a)|\tilde{r}_k(s,a,t) - r(s,a,t)|$$

$$\leq \sum_{t=1}^H \sum_{s,a \in L_{tk}} w_{tk}(s,a)|(\tilde{P}_k(s,a,t) - P(s,a,t))^\top \tilde{V}_{t+1}^{\pi_k}| + \sum_{t=1}^H \sum_{s,a \in L_{tk}} w_{tk}(s,a)|\tilde{r}_k(s,a,t) - r(s,a,t)|$$

$$+ \sum_{t=1}^H \sum_{s,a \notin L_{tk}} w_{tk}(s,a)|(\tilde{P}_k(s,a,t) - P(s,a,t))^\top \tilde{V}_{t+1}^{\pi_k}| + \sum_{t=1}^H \sum_{s,a \notin L_{tk}} w_{tk}(s,a)|\tilde{r}_k(s,a,t) - r(s,a,t)|$$

$$\leq \sum_{t=1}^H \sum_{s,a \notin L_{tk}} H w_{\min} + \sum_{t=1}^H \sum_{s,a \in L_{tk}} w_{tk}(s,a) \Big[ |(\tilde{P}_k(s,a,t) - \hat{P}_k(s,a,t))^\top \tilde{V}_{t+1}^{\pi_k}|$$

$$+ |(\hat{P}_k(s,a,t) - P(s,a,t))^\top \tilde{V}_{t+1}^{\pi_k}| + |\tilde{r}_k(s,a,t) - r(s,a,t)| \Big] \tag{9}$$

The first term is bounded by $c_\varepsilon \varepsilon = \frac{\varepsilon}{3}$. We now can use Lemma E.9, Lemma E.10 to bound the other terms by

$$\sum_{t=1}^H \sum_{s,a \in L_{tk}} w_{tk}(s,a) \sqrt{\frac{8(H + H\sqrt{S} + 2)^2}{n_{tk}(s,a)} \left( \mathrm{llnp}(n_{tk}(s,a)) + \frac{1}{2} \ln \frac{6SAH'}{\delta} \right)}.$$

We can then apply Lemma E.3 with $r = 2$, $C = 8(H + H\sqrt{S} + 2)^2$, $D = \frac{1}{2} \ln \frac{6SAH}{\delta'}$ ($\geq 1$ for any nontrivial setting) and $\varepsilon' = 2\varepsilon/3$ to bound this term by $\frac{2\varepsilon}{3}$ on all nice episodes except at most

$$\frac{64(H + \sqrt{S}H + 2)^2 ASH^2 3^2}{4\varepsilon^2} \, \mathrm{polylog}(A, S, H, 1/\varepsilon, 1/\delta)$$

$$\leq \frac{144(4 + 3H^2 + 4SH^2)ASH^2}{\varepsilon^2} \, \mathrm{polylog}(A, S, H, 1/\varepsilon, 1/\delta)$$

Hence $V_1^\star(s_0) - V_1^{\pi_k}(s_0) \leq \varepsilon$ holds on all nice episodes except those. $\square$

The lemma below is a refined version of the bound above and uses the stronger concept of friendly episodes to eventually lead to the second bound in Theorem 4.

**Lemma E.8** (Optimality Gap Bound On Friendly Episodes). *On the good event $F^c$ it holds that $p_0^\top (V_1^\star - V_1^{\pi_k}) \le \varepsilon$ on all friendly episodes $E$ except at most*

$$\left( \frac{9216}{\varepsilon} + 417S \right) \frac{ASH^4}{\varepsilon} \, \text{polylog}(S, A, H, 1/\varepsilon, \delta)$$

*episodes if $\delta' \le \frac{3AS^2H}{e^2}$.*

*Proof.* We can further decompose the optimality gap bound in Equation (9) in the proof of Lemma E.7 as

$$\sum_{t=1}^{H} \sum_{s,a \notin L_{tk}} (H+1)w_{\min} + \sum_{t=1}^{H} \sum_{s,a \in L_{tk}} w_{tk}(s,a) \left[ |(\tilde{P}_k(s,a,t) - \hat{P}_k(s,a,t))^\top \tilde{V}_{t+1}^{\pi_k}| + |\tilde{r}_k(s,a,t) - r(s,a,t)| \right.$$

$$\left. + |(\hat{P}_k(s,a,t) - P(s,a,t))^\top V_{t+1}^\star| + |(\hat{P}_k(s,a,t) - P(s,a,t))^\top (V_{t+1}^\star - \tilde{V}_{t+1}^{\pi_k})| \right].$$

$$\le c_\varepsilon \varepsilon + \sum_{t=1}^{H} \sum_{s,a \in L_{tk}} w_{tk}(s,a) \left[ |(\tilde{P}_k(s,a,t) - \hat{P}_k(s,a,t))^\top \tilde{V}_{t+1}^{\pi_k}| + |\tilde{r}_k(s,a,t) - r(s,a,t)| \right.$$

$$+ \quad |(\hat{P}_k(s,a,t) - P(s,a,t))^\top V_{t+1}^\star| \Big]$$

$$+ \sum_{t=1}^{H} \sum_{s,a \in L_{tk}} w_{tk}(s,a)|(\hat{P}_k(s,a,t) - P(s,a,t))^\top (V_{t+1}^\star - \tilde{V}_{t+1}^{\pi_k})|.$$

The second term can be bounded using Lemmas E.11, E.10 and E.9 by

$$\sum_{t=1}^{H} \sum_{s,a \in L_{tk}} w_{tk}(s,a) \sqrt{ \frac{32(H+1)^2}{n_{tk}(s,a)} \left( \text{llnp}(n_{tk}(s,a)) + \frac{1}{2} \ln \frac{6SAH}{\delta'} \right)}.$$

which we bound by $\varepsilon/3$ using Lemma E.3 with $r = 2$, $C = 32(H+1)^2$, $D = \frac{1}{2} \ln \frac{6SAH}{\delta'}$ and $\varepsilon' = \varepsilon/3$ on all friendly episodes except at most

$$\frac{8CASH^2}{\varepsilon'^2} \text{polylog}(S, A, H, 1/\varepsilon, 1/\delta) \le \frac{9216ASH^4}{\text{polylog}}(S, A, H, 1/\varepsilon, 1/\delta).$$

Finally, we apply Lemma E.12 bound to bound the last term in Equation E.4 by $\varepsilon/3$ on all friendly episodes but at most

$$\frac{417AS^2H^4}{\varepsilon} \text{polylog}(S, A, H, 1/\delta, 1/\varepsilon).$$

It hence follows that $p_0^\top (V_1^\star - V_1^{\pi_k}) \le \varepsilon$ on all friendly episodes but at most

$$\left( \frac{9216ASH^4}{\varepsilon^2} + \frac{417AS^2H^4}{\varepsilon} \right) \text{polylog}(S, A, H, 1/\delta, 1/\varepsilon).$$

$\square$

**Lemma E.9** (Algorithm Learns Fast Enough). *It holds for all $s \in \mathcal{S}, a \in \mathcal{A}$ and $t \in [H]$*

$$|(\hat{P}_k(s,a,t) - \tilde{P}_k(s,a,t))^\top \tilde{V}_{t+1}| \le \sqrt{ \frac{2H^2}{n_{tk}(s,a)} \left( \text{llnp}(n_{tk}(s,a)) + \frac{1}{2} \ln \frac{3SAH}{\delta'} \right)}.$$

*Proof.* Using the definition of the constraint in the planning step of the algorithm shown in Lemma D.1 we can bound

$$|(\hat{P}_k(s,a,t) - \tilde{P}_k(s,a,t))^\top \tilde{V}_{t+1}| \leq \sqrt{\frac{H^2}{n_{tk}(s,a)} \left( 2\,\text{llnp}(n_{tk}(s,a)) + \ln\frac{3SAH}{\delta'} \right)}.$$

$$\leq \sqrt{\frac{2H^2}{n_{tk}(s,a)} \left( \text{llnp}(n_{tk}(s,a)) + \frac{1}{2}\ln\frac{3SAH}{\delta'} \right)}.$$

□

**Lemma E.10** (Basic Decompsition Bound). *On the good event $F^c$ it holds for all $s \in \mathcal{S}, a \in \mathcal{A}$ and $t \in [H]$*

$$|(\hat{P}_k(s,a,t) - P(s,a,t))^\top \tilde{V}_{t+1}| \leq \sqrt{\frac{8H^2 S}{n_{tk}(s,a)} \left( \text{llnp}(n_{tk}(s,a)) + \frac{1}{2}\ln\frac{6SAH}{\delta'} \right)}$$

$$|\tilde{r}_k(s,a,t) - r(s,a,t)| \leq \sqrt{\frac{4}{n_{tk}(s,a)} \left( \text{llnp}(n_{tk}(s,a)) + \frac{1}{2}\ln\frac{3SAH}{\delta'} \right)}.$$

*Proof.* On the good event $(F_k^{L1})c$ we have using Hölder's inequality

$$|(\hat{P}_k(s,a,t) - P(s,a,t))^\top \tilde{V}_{t+1}| \leq \|\hat{P}_k(s,a,t) - P(s,a,t))\|_1 \|\tilde{V}_{t+1}\|_\infty$$

$$\leq H\sqrt{\frac{4}{n_{tk}(s,a)} \left( 2\,\text{llnp}(n_{tk}(s,a)) + \ln\frac{3SAH(2^S - 2)}{\delta'} \right)}$$

$$\leq \sqrt{\frac{8H^2 S}{n_{tk}(s,a)} \left( \text{llnp}(n_{tk}(s,a)) + \frac{1}{2}\ln\frac{6SAH}{\delta'} \right)}.$$

Further, on $(F_k^R)^c$ we have

$$|\tilde{r}_k(s,a,t) - r(s,a,t)| \leq |\tilde{r}_k(s,a,t) - r(s,a,t)| + |\tilde{r}_k(s,a,t) - \hat{r}(s,a,t)|$$

$$\leq 2\sqrt{\frac{1}{n_{tk}(s,a)} \left( 2\,\text{llnp}(n_{tk}(s,a)) + \ln\frac{3SAH}{\delta'} \right)}$$

□

**Lemma E.11** (Fixed V Term Confidence Bound). *On the good event $F^c$ it holds for all $s \in \mathcal{S}, a \in \mathcal{A}$ and $t \in [H]$*

$$|(\hat{P}_k(s,a,t) - P(s,a,t))^\top V_{t+1}^\star| \leq \sqrt{\frac{2H^2}{n_{tk}(s,a)} \left( \text{llnp}\, n_{tk}(s,a) + \frac{1}{2}\ln\frac{3SAH}{\delta'} \right)}$$

*Proof.* Since we consider the event $(F_k^V)^c$, we can bound

$$|(\hat{P}_k(s,a,t) - P(s,a,t))^\top V_{t+1}^\star| \leq \sqrt{\frac{2H^2}{n_{tk}(s,a)} \left( \text{llnp}\, n_{tk}(s,a) + \frac{1}{2}\ln\frac{3SAH}{\delta'} \right)}$$

□

**Lemma E.12** (Lower Order Term). *Assume $\delta' \leq \frac{3AS^2 H}{e^2}$. On the good event $F^c$ on all friendly episodes $k \in E$ except at most $\frac{417AS^2 H^4}{\varepsilon}$ polylog$(S, A, H, 1/\delta, 1/\varepsilon)$. it holds that*

$$\sum_{t=1}^{H} \sum_{s,a \in L_{tk}} w_{tk}(s,a)|(\hat{P}_k(s,a,t) - P(s,a,t))^\top (\tilde{V}_{t+1}^{\pi_k} - V_{t+1}^\star)| \leq \frac{\varepsilon}{3}.$$

*Proof.*

$$\sum_{t=1}^{H}\sum_{s,a\in L_{tk}}w_{tk}(s,a)|(\hat{P}_k(s,a,t)-P(s,a,t))^{\top}(\tilde{V}_{t+1}^{\pi_k}-V_{t+1}^{\star})|$$

$$\leq\sum_{t=1}^{H}\sum_{s,a\in L_{tk}}w_{tk}(s,a)\sum_{s'}\sqrt{\frac{2P(s'|s,a,t)}{n_{tk}(s,a)}\left(2\,\mathrm{llnp}(n_{tk}(s,a))+\ln\frac{3S^2AH}{\delta'}\right)}|\tilde{V}_{t+1}^{\pi_k}(s')-V_{t+1}^{\star}(s')|$$

$$+\sum_{t=1}^{H}\sum_{s,a\in L_{tk}}w_{tk}(s,a)\sum_{s'}\frac{1}{n_{tk}(s,a)}\left(2\,\mathrm{llnp}(n_{tk}(s,a))+\ln\frac{3S^2AH}{\delta'}\right)|\tilde{V}_{t+1}^{\pi_k}(s')-V_{t+1}^{\star}(s')|$$

$$\leq\sum_{t=1}^{H}\sum_{s,a\in L_{tk}}w_{tk}(s,a)\sum_{s'}\sqrt{\frac{2P(s'|s,a,t)}{n_{tk}(s,a)}\left(2\,\mathrm{llnp}(n_{tk}(s,a))+\ln\frac{3S^2AH}{\delta'}\right)\left(\tilde{V}_{t+1}^{\pi_k}(s')-V_{t+1}^{\star}(s')\right)^2}$$

$$+\sum_{t=1}^{H}\sum_{s,a\in L_{tk}}\frac{w_{tk}(s,a)HS}{n_{tk}(s,a)}\left(2\,\mathrm{llnp}(n_{tk}(s,a))+\ln\frac{3S^2AH}{\delta'}\right)$$

$$\leq\sum_{t=1}^{H}\sum_{s,a\in L_{tk}}w_{tk}(s,a)\sqrt{\frac{2S}{n_{tk}(s,a)}\left(2\,\mathrm{llnp}(n_{tk}(s,a))+\ln\frac{3S^2AH}{\delta'}\right)P(s,a,t)^{\top}\left(\tilde{V}_{t+1}^{\pi_k}-V_{t+1}^{\star}\right)^2}$$

$$+\sum_{t=1}^{H}\sum_{s,a\in L_{tk}}\frac{w_{tk}(s,a)HS}{n_{tk}(s,a)}\left(2\,\mathrm{llnp}(n_{tk}(s,a))+\ln\frac{3S^2AH}{\delta'}\right)$$

The first inequality follows since we only consider outcomes in the event $(F_k^P)^c$, the second from the fact that value function are in the range $[0,H]$ and the third is an application of the Cauchy-Schwarz inequality. Using of optimism of the algorithm (Lemma E.16), we now bound $P(s,a,t)^{\top}\left(\tilde{V}_{t+1}^{\pi_k}-V_{t+1}^{\star}\right)^2 \leq P(s,a,t)^{\top}\left(\tilde{V}_{t+1}^{\pi_k}-V_{t+1}^{\pi_k}\right)^2$ which we bound by $c_{\varepsilon}\varepsilon+\left(c_{\varepsilon}\varepsilon+\sqrt{\frac{C'^2}{n_{tk}(s,a)S}}\left(\mathrm{llnp}(n_{tk}(s,a)+\frac{1}{2}\ln\frac{3AS^2H\varepsilon^4}{\delta'})\right)\right)^2 \leq c_{\varepsilon}\varepsilon+(c_{\varepsilon}\varepsilon+\frac{C'}{\sqrt{S}}\sqrt{J(s,a,t)})^2$ using Lemma E.13. To keep the notation concise, we use here the shorthand $J(s,a,t) = \frac{1}{n_{tk}(s,a)}\left(\mathrm{llnp}(n_{tk}(s,a))+\frac{1}{2}\ln\frac{3e^4S^2AH}{\delta'}\right)$. This bound holds on all friendly episodes except at most $\left(32ASH^2+48AS^2H^3+AS^2H^4+16AS^2\right)\mathrm{polylog}(S,A,H,1/\delta,1/\varepsilon)$. Plugging this into the bound from above, we get the upper bound

$$\sum_{t=1}^{H}\sum_{s,a\in L_{tk}}w_{tk}(s,a)\sqrt{4SJ(s,a,t)\left(c_{\varepsilon}\varepsilon+(c_{\varepsilon}\varepsilon+C'\sqrt{J(s,a,t)/S})^2\right)}+\sum_{t=1}^{H}\sum_{s,a\in L_{tk}}2w_{tk}(s,a)HSJ(s,a,t)$$

$$\leq\sum_{t=1}^{H}\sum_{s,a\in L_{tk}}w_{tk}(s,a)\sqrt{4SJ(s,a,t)c_{\varepsilon}\varepsilon}+\sum_{t=1}^{H}\sum_{s,a\in L_{tk}}w_{tk}(s,a)\sqrt{4SJ(s,a,t)(c_{\varepsilon}\varepsilon+C'\sqrt{J(s,a,t)/S})^2}$$

$$+\sum_{t=1}^{H}\sum_{s,a\in L_{tk}}2w_{tk}(s,a)HSJ(s,a,t)$$

$$=\sum_{t=1}^{H}\sum_{s,a\in L_{tk}}w_{tk}(s,a)\sqrt{4(c_{\varepsilon}\varepsilon+c_{\varepsilon}^2\varepsilon^2)SJ(s,a,t)}+\sum_{t=1}^{H}\sum_{s,a\in L_{tk}}2w_{tk}(s,a)J(s,a,t)(C'+SH),$$

where we used $\sqrt{a+b} \leq \sqrt{a} + \sqrt{b}$. We now bound the first term using Lemma E.3 with $r = 2, \varepsilon' = \varepsilon/6, D = \frac{1}{2} \ln \frac{3e^4 S^2 AH}{\delta'}, C = 4(c_\varepsilon \varepsilon + c_\varepsilon^2 \varepsilon^2)S$ on all but $\frac{8CASH^2}{\varepsilon'^2}$ polylog$(\dots) = \frac{192c_\varepsilon(1+c_\varepsilon\varepsilon)AS^2H^2}{\varepsilon}$ polylog$(\dots)$ friendly episodes by $\varepsilon/6$.

Applying Lemma E.3 with $r = 1, \varepsilon' = \varepsilon/6, D = \frac{1}{2} \ln \frac{3e^4 S^2 AH}{\delta'}$ and $C = 2(C' + HS)$, we can bound the second term by $\varepsilon/6$ on all but $\frac{8CASH}{\varepsilon'}$ polylog$(\dots) = \frac{96AS(C'+HS)H^2}{\varepsilon}$ polylog$(\dots)$ friendly episodes. Hence, it holds

$$\sum_{t=1}^{H} \sum_{s,a \in L_{tk}} w_{tk}(s,a)|(\hat{P}_k(s,a,t) - P(s,a,t))^\top (\tilde{V}_{t+1}^{\pi_k} - V_{t+1}^\star)| \leq \frac{\varepsilon}{3}$$

on all friendly episodes except at most

$$\left( \frac{96AS(C'+HS)H^2}{\varepsilon} + \frac{192c_\varepsilon(1+c_\varepsilon\varepsilon)AS^2H^2}{\varepsilon} \right.$$

$$\left. + 32ASH^2 + 48AS^2H^3 + AS^2H^4 + 16AS^2 \right) \text{polylog}(S,A,H,1/\delta,1/\varepsilon)$$

episodes. Since $C' = \text{polylog}(S,A,H,1/\delta,1/\varepsilon)$, this simplifies to

$$\left( \frac{96AS}{\varepsilon} + \frac{96AS^2H^3}{\varepsilon} + \frac{64AS^2H^2}{\varepsilon} + 64AS^2H^2 \right.$$

$$\left. + 32ASH^2 + 48AS^2H^3 + AS^2H^4 + 16AS^2 \right) \text{polylog}(S,A,H,1/\delta,1/\varepsilon)$$

$$\leq \left( (64+32+48+1+16)AS^2H^4 + \frac{96+96+64}{\varepsilon}AS^2H^3 \right) \text{polylog}(S,A,H,1/\delta,1/\varepsilon)$$

failure episodes in $E$. We can finally bound the failure episodes by

$$\frac{417AS^2H^4}{\varepsilon} \text{polylog}(S,A,H,1/\delta,1/\varepsilon).$$

$\square$

**Lemma E.13.** *On the good event $F^c$ for any $s \in \mathcal{S}$, $a \in \mathcal{A}$ and $t \in [H]$ with $\delta' \leq \frac{3AS^2H}{e^2}$ it holds*

$$P(s,a,t)^\top (\tilde{V}_{t+1}^{\pi_k} - V_{t+1}^{\pi_k})^2 \leq c_\varepsilon \varepsilon + \left( c_\varepsilon \varepsilon + \sqrt{\frac{1}{n_{tk}(s,a)S} \left( \text{llnp}(n_{tk}(s,a) + \frac{1}{2} \ln \frac{3AS^2H\varepsilon^4}{\delta'} \right)} \right)^2$$

*where $C' = 1 + \sqrt{\frac{1}{2} \ln \frac{3e^2 S^2 AH}{\delta'}}$ on all friendly episodes except for at most*

$$\left( 32ASH^2 + 48AS^2H^3 + AS^2H^4 + 16AS^2 \right) \text{polylog}(S,A,H,1/\delta,1/\varepsilon)$$

*episodes.*

*Proof.* Define $L' = \{s' : w_{tk}^{t+1}(s',a'|s,a) > w'_{\min}\}$ and $J(s') = \frac{\text{llnp} \, n_{t+1k}(s',a') + \frac{1}{2} \ln \frac{3e^2 S^2 AH}{\delta'}}{n_{t+1k}(s',a')}$ where $a' = \pi_k(s',t+1)$ and $C' = 1 + \sqrt{\frac{1}{2} \ln \frac{3e^2 S^2 AH}{\delta'}}$. Using Lemma E.14, we bound

$$P(s,a,t)^\top (\tilde{V}_{t+1}^{\pi_k} - V_{t+1}^{\pi_k})^2 = \sum_{s'} P(s'|s,a,t)(\tilde{V}_{t+1}^{\pi_k}(s') - V_{t+1}^{\pi_k}(s'))^2$$

$$\leq Sw_{\min}H^2 + \sum_{s' \in L'} P(s'|s,a,t) \left( c_\varepsilon \varepsilon + C'\sqrt{J(s')} \right)^2$$

$$\leq c_\varepsilon \varepsilon + C'^2 \sum_{s' \in L'} P(s'|s,a,t)J(s') + c_\varepsilon^2 \varepsilon^2 + 2c_\varepsilon \varepsilon C' \sum_{s' \in L'} P(s'|s,a,t)\sqrt{J(s')}$$

on all friendly episodes except at most $\left(32 + 48SH + SH^2\right)ASH^2\operatorname{polylog}(S,A,H,1/\delta,1/\varepsilon)$. Define now $L'' = \{(s',a') : s' \in L', a' = \pi_k(s',t+1)\}$. We apply Lemma E.4 with $|T| = \{t+1\}, C = 1, D = \frac{1}{2}\ln\frac{3e^2S^2AH}{\delta'} \geq 1, r = 1$ and $\varepsilon' = 1/S$ to

$$\sum_{s' \in L'} P(s'|s,a,t)J(s') = \sum_{s',a' \in L''} \frac{w_{tk}^{t+1}(s',a'|s,a)}{n_{t+1k}(s',a')}\left(\operatorname{llnp} n_{t+1k}(s',a') + \frac{1}{2}\ln\frac{3e^2S^2AH}{\delta'}\right)$$

$$\leq \frac{1}{n_{tk}(s,a)S}\left(\operatorname{llnp}(n_{tk}(s,a)) + \frac{1}{2}\ln\frac{3S^2AHe^4}{\delta'}\right)$$

on all but at most $8AS^2\operatorname{polylog}(A,S,H,1/\delta,1/\varepsilon)$ friendly episodes. Similarly, we bound

$$\sum_{s' \in L'} P(s'|s,a,t)\sqrt{J(s')}$$

$$= \sum_{s',a' \in L''} w_{tk}^{t+1}(s',a'|s,a)\sqrt{\frac{1}{n_{t+1k}(s',a')}\left(\operatorname{llnp} n_{t+1k}(s',a') + \frac{1}{2}\ln\frac{3e^2S^2AH}{\delta'}\right)}$$

$$\leq \sqrt{\frac{1}{n_{tk}(s,a)S}\left(\operatorname{llnp}(n_{tk}(s,a)) + \frac{1}{2}\ln\frac{3S^2AHe^4}{\delta'}\right)}$$

on all but at most $8AS^2\operatorname{polylog}(A,S,H,1/\delta,1/\varepsilon)$ friendly episodes. Hence on all friendly episodes except those failure episodes, we get

$$P(s,a,t)^\top(\tilde{V}_{t+1}^{\pi_k} - V_{t+1}^{\pi_k})^2 \leq c_\varepsilon\varepsilon + \left(c_\varepsilon\varepsilon + \sqrt{\frac{C'^2}{n_{tk}(s,a)S}\left(\operatorname{llnp}(n_{tk}(s,a) + \frac{1}{2}\ln\frac{3AS^2H\varepsilon^4}{\delta'}\right)}\right)^2.$$

$\square$

**Lemma E.14.** *Consider a fix $s' \in S$ and $t \in [H]$, $\delta' \leq \frac{3AS^2H}{e^2}$ and the good event $F^c$. On all but at most*

$$\left(32 + 48SH + SH^2\right)ASH^2\operatorname{polylog}(S,A,H,1/\delta,1/\varepsilon)$$

*friendly episodes $E$ it holds that*

$$V_t^{\pi_k}(s') - \tilde{V}_t^{\pi_k}(s') \leq c_\varepsilon\varepsilon + \left(1 + \sqrt{12\ln\frac{3e^2S^2AH}{\delta'}}\right)\sqrt{\frac{1}{n_{tk}(s',a')}\left(\operatorname{llnp} n_{tk}(s',a') + \frac{1}{2}\ln\frac{3e^2S^2AH}{\delta'}\right)},$$

*where $a' = \pi_k(s',t)$.*

*Proof.* For any $t,s'$ and $a' = \pi_k(s',t)$ we use Lemma E.15 to write the value difference as

$$\tilde{V}_t^{\pi_k}(s') - V_t^{\pi_k}(s') = \sum_{u=t}^{H}\sum_{s,a} w_{tk}^u(s,a|s',a')(\tilde{P}_k(s,a,u) - P(s,a,u))^\top\tilde{V}_{u+1}$$

$$+ \sum_{u=t}^{H}\sum_{s,a} w_{tk}^u(s,a|s',a')(\tilde{r}_k(s,a,u) - r(s,a,u))$$

Let $L_k^{ut} = \{s,a \in S \times A : w_{tk}^u(s,a|s',a') \geq w_{\min}\}$ be the set of state-action pairs for which the conditional probability of observing is sufficiently large. Then we can bound the low-probability differences as

$$\sum_{u=t}^{H}\sum_{s,a \in (L_k^{ut})^c} w_{tk}^u(s,a|s',a')[(\tilde{r}_k(s,a,u) - r(s,a,u)) + (P(s,a,u) - \tilde{P}_k(s,a,u))^\top\tilde{V}_{u+1}]$$

$$\leq \sum_{u=t}^{H}\sum_{s,a \in (L_k^{ut})^c} w_{\min}H \leq w_{\min}H^2S = c_\varepsilon\varepsilon.$$

For the other terms with significant conditional probability, we can leverage the fact that we only consider events in $(F_k^R)^c$ and $(F_k^P)^c$ to bound

$$\sum_{u=t}^{H} \sum_{s,a \in L_k^{ut}} w_{tk}^u(s,a|s',a')(\tilde{r}_k(s,a,u) - r(s,a,u))$$

$$\leq \sum_{u=t}^{H} \sum_{s,a \in L_k^{ut}} w_{tk}^u(s,a|s',a') \sqrt{\frac{32}{n_{tk}(s,a)}\left(\operatorname{llnp}(n_{tk}(s,a)) + \frac{1}{2}\ln\frac{3SAH}{\delta'}\right)}$$

and

$$\sum_{u=t}^{H} \sum_{s,a \in L_k^{ut}} w_{tk}^u(s,a|s',a')(P(s,a,u) - \tilde{P}_k(s,a,u))^\top \tilde{V}_{u+1}$$

$$\leq \sum_{u=t}^{H} \sum_{s,a \in L_k^{ut}} w_{tk}^u(s,a|s',a') \sum_{s''} \tilde{V}_{u+1}(s'') \sqrt{\frac{2P(s''|s,a,u)}{n_{uk}(s,a)}\left(2\operatorname{llnp}(n_{uk}(s,a)) + \ln\frac{3S^2AH}{\delta'}\right)}$$

$$+ \sum_{u=t}^{H} \sum_{s,a \in L_k^{ut}} w_{tk}^u(s,a|s',a') \sum_{s''} \frac{\tilde{V}_{u+1}(s'')}{n_{uk}(s,a)}\left(2\operatorname{llnp}(n_{uk}(s,a)) + \ln\frac{3S^2AH}{\delta'}\right)$$

$$\leq \sum_{u=t}^{H} \sum_{s,a \in L_k^{ut}} w_{tk}^u(s,a|s',a') \sqrt{\frac{2SH^2}{n_{uk}(s,a)}\left(2\operatorname{llnp}(n_{uk}(s,a)) + \ln\frac{3S^2AH}{\delta'}\right)}$$

$$+ \sum_{u=t}^{H} \sum_{s,a \in L_k^{ut}} w_{tk}^u(s,a|s',a') \frac{SH}{n_{uk}(s,a)}\left(2\operatorname{llnp}(n_{uk}(s,a)) + \ln\frac{3S^2AH}{\delta'}\right)$$

where we use Cauchy Schwarz for the last inequality. Combining these individual bounds, we can upper-bound the value difference as

$$\tilde{V}_t^{\pi_k}(s') - V_t^{\pi_k}(s')$$

$$\leq c_\varepsilon \varepsilon + \sum_{u=t}^{H} \sum_{s,a \in L_k^{ut}} w_{tk}^u(s,a|s',a') \sqrt{\frac{(4\sqrt{2}+2\sqrt{S}H)^2}{n_{uk}(s,a)}\left(\operatorname{llnp}(n_{uk}(s,a)) + \frac{1}{2}\ln\frac{3S^2AH}{\delta'}\right)}$$

$$+ \sum_{u=t}^{H} \sum_{s,a \in L_k^{ut}} w_{tk}^u(s,a|s',a') \frac{2SH}{n_{uk}(s,a)}\left(\operatorname{llnp}(n_{uk}(s,a)) + \frac{1}{2}\ln\frac{3S^2AH}{\delta'}\right) \tag{10}$$

We now apply Lemma E.4 with $r=2, D = \frac{1}{2}\ln\frac{3S^2AH}{\delta'}, C = (4\sqrt{2}+2\sqrt{S}H)^2, T = \{t+1, t+2, \ldots H\}$ and $\varepsilon' = 1$ and get that the second term above is bounded by

$$\sqrt{\frac{1}{n_{tk}(s',a')}\left(\operatorname{llnp} n_{tk}(s',a') + \frac{1}{2}\ln\frac{3e^2S^2AH}{\delta'}\right)}$$

on all friendly episodes but at most

$$\frac{8CASH^2}{\varepsilon'^2}\operatorname{polylog}(S,A,H,1/\delta,1/\varepsilon) = (32 + 16\sqrt{2S}H + SH^2)ASH^2\operatorname{polylog}(S,A,H,1/\delta,1/\varepsilon)$$

episodes. We apply Lemma E.4 again to the final term in Equation (10) above with $r=1, D = \frac{1}{2}\ln\frac{3S^2AH}{\delta'} \geq 1, T = \{t+1, t+2, \ldots H\}, C = 2SH$ and $\varepsilon' = 1$. Then the final term is bounded by $\frac{1}{n_{tk}(s',a')}\left(\operatorname{llnp} n_{tk}(s',a') + \frac{1}{2}\ln\frac{3e^2S^2AH}{\delta'}\right)$. on all friendly episodes but

$$\frac{8CASH^2}{\varepsilon'^2}\operatorname{polylog}(S,A,H,1/\delta,1/\varepsilon) = 16AS^2H^3\operatorname{polylog}(S,A,H,1/\delta,1/\varepsilon)$$

many. Combining these bounds, we arrive at

$$V_t^{\pi_k}(s') - \tilde{V}_t^{\pi_k}(s')$$

$$\leq c_\varepsilon \varepsilon + \sqrt{\frac{1}{n_{tk}(s',a')}\left(\text{llnp}\, n_{tk}(s',a') + \frac{1}{2}\ln\frac{3e^2 S^2 AH}{\delta'}\right)}$$

$$+ \frac{1}{n_{tk}(s',a')}\left(\text{llnp}\, n_{tk}(s',a') + \frac{1}{2}\ln\frac{3e^2 S^2 AH}{\delta'}\right)$$

$$\leq c_\varepsilon \varepsilon + \left(1 + \sqrt{\frac{1}{2}\ln\frac{3e^2 S^2 AH}{\delta'}}\right)\sqrt{\frac{1}{n_{tk}(s',a')}\left(\text{llnp}\, n_{tk}(s',a') + \frac{1}{2}\ln\frac{3e^2 S^2 AH}{\delta'}\right)},$$

where we bounded $\sqrt{\frac{1}{n_{tk}(s',a')}\left(\text{llnp}\, n_{tk}(s',a') + \frac{1}{2}\ln\frac{3e^2 S^2 AH}{\delta'}\right)}$ by $\frac{1}{2}\ln\frac{3e^2 S^2 AH}{\delta'}$ since it is decreasing in $n_{tk}(s',a')$ and we therefore can simply use $n_{tk}(s',a') = 1$ (entire bound holds trivially for $n_{tk}(s',a') = 0$). $\qquad\square$

### E.5 Useful Lemmas

**Lemma E.15** (Value Difference Lemma). *For any two MDPs $M'$ and $M''$ with rewards $r'$ and $r''$ and transition probabilities $P'$ and $P''$, the difference in values with respect to the same policy $\pi$ can be written as*

$$V_i'(s) - V_i''(s) = \mathbb{E}''\left[\sum_{t=i}^{H}(r'(s_t,a_t,t) - r''(s_t,a_t,t))\Big|s_i = s\right] + \mathbb{E}''\left[\sum_{t=i}^{H}(P'(s_t,a_t,t) - P''(s_t,a_t,t))^\top V_{t+1}'\Big|s_i = s\right]$$

*where $V_{H+1}' = V_{H+1}'' = \vec{0}$ and the expectation $\mathbb{E}'$ is taken w.r.t to $P'$ and $\pi$ and $\mathbb{E}''$ w.r.t. $P''$ and $\pi$.*

*Proof.* For $i = H + 1$ the statement is trivially true. We assume now it holds for $i + 1$ and show it holds also for $i$. Using only this induction hypothesis and basic algebra, we can write

$$V_i'(s) - V_i''(s)$$
$$= \mathbb{E}_\pi[r'(s_i,a_i,i) + V_{i+1}'^\top P'(s_i,a_i,i) - r''(s_i,a_i,i) - V_{i+1}''^\top P''(s_i,a_i,i)|s_i = s]$$
$$= \mathbb{E}_\pi[r'(s_i,a_i,i) - r''(s_i,a_i,i)|s_i = s] + \mathbb{E}_\pi\left[\sum_{s'\in\mathcal{S}}V_{i+1}'(s')(P'(s'|s_i,a_i,i) - P''(s'|s_i,a_i,i))\Big|s_i = s\right]$$

$$+ \mathbb{E}_\pi\left[\sum_{s'\in\mathcal{S}}P''(s'|s_i,a_i,i)(V_{i+1}'(s') - V_{i+1}''(s'))\Big|s_i = s\right]$$
$$= \mathbb{E}_\pi[r'(s_i,a_i,i) - r''(s_i,a_i,i)|s_i = s] + \mathbb{E}_\pi\left[\sum_{s'\in\mathcal{S}}V_{i+1}'(s')(P'(s'|s_i,a_i,i) - P''(s'|s_i,a_i,i))\Big|s_i = s\right]$$

$$+ \mathbb{E}''\left[V_{i+1}'(s_{i+1}) - V_{i+1}''(s_{i+1}))\Big|s_i = s\right]$$
$$= \mathbb{E}_\pi[r'(s_i,a_i,i) - r''(s_i,a_i,i)|s_i = s] + \mathbb{E}_\pi\left[\sum_{s'\in\mathcal{S}}V_{i+1}'(s')(P'(s'|s_i,a_i,i) - P''(s'|s_i,a_i,i))\Big|s_i = s\right]$$

$$+ \mathbb{E}''\left[\mathbb{E}''\left[\sum_{t=i+1}^{H}(r'(s_t,a_t,t) - r''(s_t,a_t,t))\Big|s_{i+1}\right] + \mathbb{E}''\left[\sum_{t=i+1}^{H}(P'(s_t,a_t,t) - P''(s_t,a_t,t))^\top V_{t+1}'\Big|s_{i+1}\right]\Big|s_i = s\right]$$

$$=\mathbb{E}''\left[\sum_{t=i}^{H}(r'(s_t,a_t,t)-r''(s_t,a_t,t))\Big|s_i=s\right]+\mathbb{E}''\left[\sum_{t=i}^{H}(P'(s_t,a_t,t)-P''(s_t,a_t,t))^\top V'_{t+1}\Big|s_i=s\right]$$

where the last equality follows from law of total expectation $\qquad\square$

**Lemma E.16** (Algorithm ensures optimism). *On the good event $F^c$ it holds that for all episodes $k$, $t\in[H]$, $s\in\mathcal{S}$ that*

$$V_t^{\pi_k}(s)\le V_t^\star(s)\le \tilde{V}_t^{\pi_k}(s).$$

*Proof.* The first inequality follows simply from the definition of the optimal value function $V^\star$.

Since all outcome we consider are in the event $(F_k^V)^c$, we know that the true transition probabilities $P$, the optimal policy $\pi^\star$ and optimal policy $V^\star$ are a feasible solution for the optimistic planning problem in Lemma D.1 that UBEV solves. It therefore follows immediately that $p_0^\top \tilde{V}_1^{\pi_k}\ge p_0^\top V_1^\star$. $\qquad\square$

# F    General Concentration Bounds

**Lemma F.1.** *Let $X_1,X_2,\dots$ be a martingale difference sequence adapted to filtration $\{\mathcal{F}_t\}_{t=1}^\infty$ with $X_t$ conditionally $\sigma^2$-subgaussian so that $\mathbb{E}[\exp(\lambda(X_t-\mu))|\mathcal{F}_{t-1}]\le\exp(\lambda^2\sigma^2/2)$ almost surely for all $\lambda\in\mathbb{R}$. Then with $\hat\mu_t=\frac{1}{t}\sum_{i=1}^t X_i$ we have for all $\delta\in(0,1]$*

$$\mathbb{P}\left(\exists t:|\hat\mu_t-\mu|\ge\sqrt{\frac{4\sigma^2}{t}\left(2\,\mathrm{llnp}(t)+\ln\frac{3}{\delta}\right)}\right)\le 2\delta.$$

*Proof.* Let $S_t=\sum_{s=1}^t(X_s-\mu)$. Then

$$\mathbb{P}\left(\exists t:\hat\mu_t-\mu\ge\sqrt{\frac{4\sigma^2}{t}\left(2\,\mathrm{llnp}(t)+\ln\frac{3}{\delta}\right)}\right)$$

$$\le\mathbb{P}\left(\exists t:S_t\ge\sqrt{4\sigma^2 t\left(2\,\mathrm{llnp}(t)+\ln\frac{3}{\delta}\right)}\right)$$

$$\le\sum_{k=0}^\infty\mathbb{P}\left(\exists t\in[2^k,2^{k+1}]:S_t\ge\sqrt{4\sigma^2 t\left(2\,\mathrm{llnp}(t)+\ln\frac{3}{\delta}\right)}\right)$$

$$\le\sum_{k=0}^\infty\mathbb{P}\left(\exists t\le 2^{k+1}:S_t\ge\sqrt{2\sigma^2 2^{k+1}\left(2\,\mathrm{llnp}(2^k)+\ln\frac{3}{\delta}\right)}\right)$$

We now consider $M_t=\exp(\lambda S_t)$ for $\lambda>0$ which is a nonnegative sub-martingale and use the short-hand $f=\sqrt{2\sigma^2 2^{k+1}\left(2\,\mathrm{llnp}(2^k)+\ln\frac{3}{\delta}\right)}$. Then by Doob's maximal inequality for nonnegative submartingales

$$\mathbb{P}\left(\exists t\le 2^{k+1}:S_t\ge f\right)=\mathbb{P}\left(\max_{t\le 2^{k+1}}M_t\ge\exp(\lambda f)\right)\le\frac{\mathbb{E}[M_{2^{k+1}}]}{\exp(\lambda f)}\le\exp\left(2^{k+1}\frac{\lambda^2\sigma^2}{2}-\lambda f\right).$$

Choosing the optimal $\lambda=\frac{f}{\sigma^2 2^{k+1}}$ we obtain the bound

$$\mathbb{P}\left(\exists t\le 2^{k+1}:S_t\ge f\right)\le\exp\left(-\frac{f^2}{2^{k+2}\sigma^2}\right)=\exp\left(-2\,\mathrm{llnp}(2^k)-\ln\frac{3}{\delta}\right)=\frac{\delta}{3}\exp\left(-2\,\mathrm{llnp}(2^k)\right) \tag{11}$$

$$=\frac{\delta}{3}\exp\left(-\max\{0,2\ln\max\{0,\ln 2^k\}\}\right)=\frac{\delta}{3}\min\left\{1,(k\ln 2)^{-2}\right\}$$

$$\le\frac{\delta}{3}\min\left\{1,\frac{1}{k^2\ln 2}\right\}.$$

Plugging this back in the bound from above, we get

$$\mathbb{P}\left(\exists t: \hat{\mu}_t - \mu \geq \sqrt{\frac{4\sigma^2}{t}\left(2\operatorname{llnp}(t) + \ln\frac{3}{\delta}\right)}\right) \leq \frac{\delta}{3}\sum_{k=0}^{\infty}\min\left\{1, \frac{1}{k^2\ln(2)}\right\}$$

$$= \delta\frac{1}{3}\left(\frac{\pi^2}{6\ln 2} + 2 - 1/\ln(2)\right) \leq \delta. \quad (12)$$

For the other side, the argument follows completely analogously with

$$\mathbb{P}\left(\exists t \leq 2^{k+1}: S_t \leq -f\right) = \mathbb{P}\left(\exists t \leq 2^{k+1}: -S_t \geq f\right)$$

$$= \mathbb{P}\left(\max_{t\leq 2^{k+1}}\exp(-\lambda S_t) \geq \exp(\lambda f)\right)$$

$$\leq \frac{\mathbb{E}[\exp(-\lambda S_{2^{k+1}})]}{\exp(\lambda f)} \leq \exp\left(2^{k+1}\frac{\lambda^2\sigma^2}{2} - \lambda f\right).$$

$\square$

**Lemma F.2.** *Let $X_1, X_2, \ldots$ be a sequence of Bernoulli random variables with bias $\mu \in [0,1]$. Then for all $\delta \in (0,1]$*

$$\mathbb{P}\left(\exists t: |\hat{\mu}_t - \mu| \geq \sqrt{\frac{2\mu}{t}\left(2\operatorname{llnp}(t) + \ln\frac{3}{\delta}\right)} + \frac{1}{t}\left(2\operatorname{llnp}(t) + \ln\frac{3}{\delta}\right)\right) \leq 2\delta$$

*Proof.*

$$\mathbb{P}\left(\exists t: \hat{\mu}_t - \mu \geq \sqrt{\frac{2\mu}{t}\left(2\operatorname{llnp}(t) + \ln\frac{3}{\delta}\right)} + \frac{1}{t}\left(2\operatorname{llnp}(t) + \ln\frac{3}{\delta}\right)\right)$$

$$= \mathbb{P}\left(\exists t: S_t \geq \sqrt{2\mu t\left(2\operatorname{llnp}(t) + \ln\frac{3}{\delta}\right)} + 2\operatorname{llnp}(t) + \ln\frac{3}{\delta}\right)$$

$$\leq \sum_{k=0}^{\infty}\mathbb{P}\left(\exists t \leq 2^{k+1}: S_t \geq \sqrt{2\mu 2^k\left(2\operatorname{llnp}(2^k) + \ln\frac{3}{\delta}\right)} + 2\operatorname{llnp}(2^k) + \ln\frac{3}{\delta}\right)$$

Let $g = 2\operatorname{llnp}(2^k) + \ln\frac{3}{\delta}$ and $f = \sqrt{2^{k+1}\mu g} + g$. Further define $S_t = \sum_{i=1}^{t}X_i - t\mu$ and $M_t = \exp(\lambda S_t)$ which is by construction a nonnegative submartingale. Applying Doob's maximal inequality for nonnegative submartingales, we bound

$$\mathbb{P}\left(\exists t \leq 2^{k+1}: S_t \geq f\right) = \mathbb{P}\left(\max_{i\leq 2^{k+1}}M_i \geq \exp(\lambda f)\right) \leq \frac{\mathbb{E}[M_{2^{k+1}}]}{\exp(\lambda f)} = \exp\left(\ln\mathbb{E}[M_{2^{k+1}}] - \lambda f\right).$$

Since this holds for all $\lambda \in \mathbb{R}$, we can bound

$$\mathbb{P}\left(\exists t \leq 2^{k+1}: S_t \geq f\right) \leq \exp\left(-\sup_{\lambda\in\mathbb{R}}\left(\lambda f - \ln\mathbb{E}[M_{2^{k+1}}]\right)\right)$$

and using Corollary 2.11 by Boucheron et al. [25] (see also note below proof of Corollary 2.11) bound that by

$$\exp\left(-\frac{f^2}{2(2^{k+1}\mu + f/3)}\right)$$

We now argue that this quantity can be upper-bounded by $\exp(-g)$. This is equivalent to

$$-\frac{f^2}{2(2^{k+1}\mu + f/3)} \leq -g$$

$$f^2 \geq 2g(2^{k+1}\mu + f/3) = \frac{2}{3}gf + \frac{2^{k+2}}{3}\mu g$$

$$g^2 + 2\sqrt{2^{k+1}\mu g}g + 2^{k+1}\mu g \geq \frac{2}{3}g^2 + \frac{2}{3}\sqrt{2^{k+1}\mu g}g + \frac{2^{k+2}}{3}\mu g$$

$$\frac{1}{3}g^2 + \frac{4}{3}\sqrt{2^{k+1}\mu g}g + \frac{1}{3}2^{k+1}\mu g \geq 0.$$

Each line is an equivalent inequality since $g, f \geq 0$ and each term on the left in the final inequality is nonnegative. Hence, we get $\mathbb{P}\left(\exists t \leq 2^{k+1} : S_t \geq f\right) \leq \exp(-g)$. Following now the arguments from the proof of Lemma F.1 in Equations (11)–(12), we obtain that

$$\mathbb{P}\left(\exists t : \hat{\mu}_t - \mu \geq \sqrt{\frac{2\mu}{t}\left(2\,\text{llnp}(t) + \ln\frac{3}{\delta}\right)} + \frac{1}{t}\left(2\,\text{llnp}(t) + \ln\frac{3}{\delta}\right)\right) \leq \delta.$$

For the other direction, we proceed analogously to above and arrive at

$$\mathbb{P}\left(\exists t \leq 2^{k+1} : -S_t \geq f\right) \leq \exp\left(-\sup_{\lambda \in \mathbb{R}}\left(-\lambda f - \ln\mathbb{E}[M_{2^{k+1}}]\right)\right)$$

which we bound similarly to above by

$$\exp\left(-\frac{f^2}{2(2^{k+1}\mu - f/3)}\right) \leq \exp\left(-\frac{f^2}{2(2^{k+1}\mu + f/3)}\right) \leq \exp(-g).$$

$\square$

**Lemma F.3** (Uniform L1-Deviation Bound for Empirical Distribution). *Let $X_1, X_2, \ldots$ be a sequence of i.i.d. categorical variables on $[U]$ with distribution $P$. Then for all $\delta \in (0, 1]$*

$$\mathbb{P}\left(\exists t \ : \ \|\hat{P}_t - P\|_1 \geq \sqrt{\frac{4}{t}\left(2\,\text{llnp}(t) + \ln\frac{3(2^U - 2)}{\delta}\right)}\right) \leq \delta$$

*where $\hat{P}_t$ is the empirical distribution based on samples $X_1 \ldots X_t$.*

*Proof.* We use the identity $\|Q - P\|_1 = 2\max_{B \subseteq \mathcal{B}} Q(B) - P(B)$ which holds for all distributions $P, Q$ defined on the finite set $\mathcal{B}$ to bound

$$\mathbb{P}\left(\exists t \ : \ \|\hat{P}_t - P\|_1 \geq \sqrt{\frac{4}{t}\left(2\,\text{llnp}(t) + \ln\frac{3(2^U - 2)}{\delta}\right)}\right)$$

$$= \mathbb{P}\left(\max_{t, B \subseteq [U]}\hat{P}_t(B) - P(B) \geq \frac{1}{2}\sqrt{\frac{4}{t}\left(2\,\text{llnp}(t) + \ln\frac{3(2^U - 2)}{\delta}\right)}\right)$$

$$\leq \sum_{B \subseteq [U]}\mathbb{P}\left(\max_t \hat{P}_t(B) - P(B) \geq \sqrt{\frac{1}{t}\left(2\,\text{llnp}(t) + \ln\frac{3(2^U - 2)}{\delta}\right)}\right).$$

Define now $S_t = \sum_{i=1}^{t} \mathbb{I}\{X_1 \in B\} - tP(B)$ which is a martingale sequence. Then the last line above is equivalent to

$$\sum_{B \subseteq [U]} \mathbb{P}\left(\max_t S_t \geq \sqrt{t\left(2\,\text{llnp}(t) + \ln\frac{3(2^U - 2)}{\delta}\right)}\right)$$

$$\leq \sum_{B \subseteq [U]} \mathbb{P}\left(\max_{k \in \mathbb{N}, t \in [2^k, 2^{k+1}]} S_t \geq \sqrt{t\left(2\,\text{llnp}(t) + \ln\frac{3(2^U - 2)}{\delta}\right)}\right)$$

$$\leq \sum_{B \subseteq [U]} \sum_{k=0}^{\infty} \mathbb{P}\left(\max_{t \in [2^k, 2^{k+1}]} S_t \geq \sqrt{t\left(2\,\text{llnp}(t) + \ln\frac{3(2^U - 2)}{\delta}\right)}\right)$$

$$\leq \sum_{B \subseteq [U]} \sum_{k=0}^{\infty} \mathbb{P}\left(\max_{t \leq 2^{k+1}} S_t \geq \sqrt{2^k\left(2\,\text{llnp}(2^k) + \ln\frac{3(2^U - 2)}{\delta}\right)}\right)$$

$$= \sum_{B \subseteq [U]} \sum_{k=0}^{\infty} \mathbb{P}\left(\max_{t \leq 2^{k+1}} \exp(\lambda S_t) \geq \exp(\lambda f)\right)$$

$$= \sum_{B \subseteq [U], B \neq \emptyset, B \neq [U]} \sum_{k=0}^{\infty} \mathbb{P}\left(\max_{t \leq 2^{k+1}} \exp(\lambda S_t) \geq \exp(\lambda f)\right)$$

where $f = \sqrt{2^k\left(2\,\text{llnp}(2^k) + \ln\frac{3(2^U - 2)}{\delta}\right)}$ and $\lambda \in \mathbb{R}$ and the last equality follows from the fact that for $B = \emptyset$ and $B = [U]$ the difference between the distributions has to be 0. Since $\mathbb{I}\{X_1 \in B\} - tP(B)$ is a centered Bernoulli variable it is $1/2$-subgaussian and so $S_t$ satisfies $\mathbb{E}[\exp(\lambda S_t)] \leq \exp(\lambda^2 t/8)]$. Since $S_t$ is a martingale, $\exp(\lambda S_t)$ is a nonnegative sub-martingale and we can apply the maximal inequality to bound

$$\mathbb{P}\left(\max_{t \leq 2^{k+1}} \exp(\lambda S_t) \geq \exp(\lambda f)\right) \leq \exp\left(\frac{1}{8}\lambda^2 2^{k+1} - \lambda f\right).$$

Choosing $\lambda = \frac{4f}{2^{k+1}}$, we get $\mathbb{P}\left(\max_{t \leq 2^{k+1}} \exp(\lambda S_t) \geq \exp(\lambda f)\right) \leq \exp\left(-\frac{f^2}{2^k}\right)$. Hence, using the same steps as in the proof of Lemma F.1, we get $\mathbb{P}\left(\max_{t \leq 2^{k+1}} \exp(\lambda S_t) \geq \exp(\lambda f)\right) \leq \frac{\delta}{3(2^{[U]}-2)}\min\left\{1, \frac{1}{k^2 \ln 2}\right\}$ and then

$$\mathbb{P}\left(\exists t : \|\hat{P}_t - P\|_1 \geq \sqrt{\frac{4}{t}\left(2\,\text{llnp}(t) + \ln\frac{3(2^U - 2)}{\delta}\right)}\right)$$

$$\leq \sum_{B \subseteq [U], B \neq \emptyset, B \neq [U]} \frac{\delta}{3(2^{[U]} - 2)} \sum_{k=0}^{\infty} \min\left\{1, \frac{1}{k^2 \ln 2}\right\} \leq \sum_{B \subseteq [U], B \neq \emptyset, B \neq [U]} \frac{\delta}{2^{[U]} - 2} = \delta.$$

$\square$

**Lemma F.4.** *Let $\mathcal{F}_i$ for $i = 1\ldots$ be a filtration and $X_1, \ldots X_n$ be a sequence of Bernoulli random variables with $\mathbb{P}(X_i = 1|\mathcal{F}_{i-1}) = P_i$ with $P_i$ being $\mathcal{F}_{i-1}$-measurable and $X_i$ being $\mathcal{F}_i$ measurable. It holds that*

$$\mathbb{P}\left(\exists n : \sum_{t=1}^{n} X_t < \sum_{t=1}^{n} P_t/2 - W\right) \leq e^{-W}$$

*Proof.* $P_t - X_t$ is a Martingale difference sequence with respect to the filtration $\mathcal{F}_t$. Since $X_t$ is nonnegative and has finite second moment, we have for any $\lambda > 0$ that $\mathbb{E}\left[e^{-\lambda(X_t - P_t)}|\mathcal{F}_{t-1}\right] \leq e^{\lambda^2 P_t/2}$ (Exercise 2.9, Boucheron et al. [25]). Hence, we have

$$\mathbb{E}\left[e^{\lambda(P_t - X_t) - \lambda^2 P_t/2}|\mathcal{F}_{t-1}\right] \leq 1$$

and by setting $\lambda = 1$, we see that

$$M_n = e^{\sum_{t=1}^{n}(-X_t + P_t/2)}$$

is a supermartingale. It hence holds by Markov's inequality

$$\mathbb{P}\left(\sum_{t=1}^{n}(-X_t + P_t/2) \geq W\right) = \mathbb{P}\left(M_n \geq e^W\right) \leq e^{-W}\mathbb{E}[M_n] \leq e^{-W}$$

wich gives us the derised result

$$\mathbb{P}\left(\sum_{t=1}^{n} X_t \leq \sum_{t=1}^{n} P_t/2 - W\right) \leq e^{-W}$$

for a fixed $n$. We define now the stopping time $\tau = \min\{t \in \mathbb{N} : M_t > e^W\}$ and the sequence $\tau_n = \min\{t \in \mathbb{N} : M_t > e^W \vee t \geq n\}$. Applying the convergence theorem for nonnegative supermartingales (Theorem 5.2.9 in Durrett [26]), we get that $\lim_{t\to\infty} M_t$ is well-defined almost surely. Therefore, $M_\tau$ is well-defined even when $\tau = \infty$. By the optional stopping theorem for nonnegative supermartingales (Theorem 5.7.6 by Durrett [26]), we have $\mathbb{E}[M_{\tau_n}] \leq \mathbb{E}[M_0] \leq 1$ for all $n$ and applying Fatou's lemma, we obtain $\mathbb{E}[M_\tau] = \mathbb{E}[\lim_{n\to\infty} M_{\tau_n}] \leq \liminf_{n\to\infty} \mathbb{E}[M_{\tau_n}] \leq 1$. Using Markov's inequality, we can finally bound

$$\mathbb{P}\left(\exists n : \sum_{t=1}^{n} X_t < \frac{1}{2}\sum_{t=1}^{n} P_t - W\right) \leq \mathbb{P}(\tau < \infty) \leq \mathbb{P}(M_\tau > e^W) \leq e^{-W}\mathbb{E}[M_\tau] \leq e^{-W}.$$

$\square$

## Footnotes

[4]We here only use $H/2$ timesteps for bandits and the remaining $H/2$ time steps to accumulate a reward of $O(H)$ for each bandit