[Reviews · NeurIPS 2017]

Reviewer 1



The paper defines "Uniform-PAC" where uniformity is over the optimality criterion, eps. It is PAC like in that optimal actions are taken in all but a bounded number of steps. It is also regret like in that the algorithm is eventually good relative to any epsilon---not just one it is told to meet. I liked the paper. I thought the discussion of different performance metrics was thorough and informative. I would have liked more intuition about the iterated logarithm idea and its main properties, but I understand that the highly technical stuff had to be expressed in very limited space. I would also have liked to have seen a more thorough accounting of open problems, as arguing that we have a new algorithmic criterion means that some new problems arise and some older problems are reduced in priority. So, for example, do we need an analysis of the infinite-horizon case? Does it raise new issues or do we essentially replace H by 1/(1-gamma) and things go through? Does the criterion have interesting problems from a bandit perspective? Detailed comments: "UBEV leverages new confidence intervals which the law of iterated logarithm allows us to more tightly bound the probability of failure events, in which the algorithm may not achieve high performance.": Confusing, if not broken. Rewrite. "F_PAC(1/eps, log(1/delta)": Missing paren. Also, F_PAC is used immediately after with several new parameters. "F_UHPR(S,A,H,T,log(1/delta)": Missing paren. "(Delta_k)_k": Isn't that second k incorrect? "guarantees only bounds" -> "guarantees only bound". "criteria overcomes" -> "criterion overcomes"? "that an algorithm optimal" -> "that an algorithm with optimal"? "Uniform-PAC is the bridge" -> "Uniform-PAC is a bridge"? "thet" -> "the"? I'm confused about the implications of Theorem 3 on Theorem 2. Theorem 3b says the algorithm produces an (eps,delta)-PAC bound. Theorem 3c says it has T^1/2 high probability regret. But, Theorem 2a that an algorithm can't have an (eps,delta)-PAC bound and high probability regret better than T^2/3. Theorem 2b is even stronger, saying it can't do better than T. Oh, wait, maybe I see the subtlety here. The issue is not that *no* algorithm with a certain PAC bound can achieve a certain regret bound. It's that knowing that an algorithm satisfies the PAC bound is insufficient to guarantee that it has the regret bound. The UPAC bound is stronger and results in algorithms that have both kinds of guarantees. I think you can make this clearer in the paper. (It's very interesting.) "rng" is used without explanation. I found https://en.wikipedia.org/wiki/Rng_(algebra), but it doesn't seem to use rng as an operator, just a property. So, I think additional background is needed here. "All algorithms are run with setting where their bounds hold.": Reword? "increase. . One": Extra period. "the discounted for which": Missing word? Note: I was very impressed with the comments of Assigned_Reviewer_3, but they didn't significantly dampen my enthusiasm for the paper. To me, the interesting contribution is the definition of a criterion that implies both PAC and regret bounds. The analysis of the finite-horizon case was just a small downpayment on using this concept. That being said, I would be much happier if an infinite horizon analysis was presented and I do not think your claim that it is "even not clear how regret should be defined here" is very convincing. There are several concepts used in the literature that would seem worth considering. Something related to the Strehl/Littman concept raised by Assigned_Reviewer_1 seems like it could work.

Reviewer 2



SUMMARY. This paper considers the setting of finite horizon episodic learning MDPs, and studies the performance of learning algorithms in terms of PAC and regret guarantees. The paper introduces a novel algorithm called UBEV, together with a finite time regret and PAC analysis. DETAILED COMMENTS. I initially thought this would be a very promising article; In the end I am quite disappointed. A) I find Section 2 a bit spurious. The whole discussion seems to be motivated only by the fact that you make use of a refined concentration inequality based on a peeling argument instead of the crude union bound that is sometimes used in MDP analysis. Further, the UBEV algorithm is nothing but a regret minimization algorithm for undiscounted MDPs, in the spirit of UCRL, KL-UCRL and so on, with the key notable difference that it applies to the episodic setting with fixed horizon H. I recommend removing this section 2 entirely. B) Finite Horizon H: The assumption of an episodic setting with a fixed, known horizon H should really be discussed. I find it very misleading that in page 5, several differences are mentioned with other setting but this knowledge of the horizon is not even mentioned. This vastly simplifying assumption is actually the key element that enables to obtain a simplified analysis. In particular, it naturally gives a known upper bound on the diameter (H) of the MDP, and more importantly it enables the use of Dynamic Programming, as opposed to Value Iteration, for which it is simple to enforce constrained/regularization on the MDP. In stark contrast, computing an optimistic value while enforcing constrained on the output value in the context of an infinite horizon is a key challenge; See algorithms such as UCRL, KL-UCRL or REGAL that consider infinite horizon (they do not resort to DP due to the infinite horizon), plus discussions there-in regarding the difficulty of enforcing constrains on the MDP in practice. C) Scaling of the regret: The leading term of the regret in Theorem 4 scales with sqrt{S} but it also scales with H^2 with the horizon. In this context where H upper bounds the diameter D of the MDP, this dependency seems to be loose compared to the regret of other algorithms that scale with D, not D^2. Thus it seems in the analysis you just improve on one factor but loose on another one. Once again, the fact this "detail" is not even discussed is misleading. D) Concentration inequalities: The main building block of this paper, apart from the greatly simplifying assumption of having a finite horizon H, is to resort to an improved concentration inequality for the empirical transition distribution, in order to properly handle the random stopping time (number of observations of a state-action pair). Indeed the naive way to obtain concentration inequalities for the empirical mean reward or transition probability at time t, with N_t(s,a) observations, is to apply a crude union bound on all possible values of N_t(s,a), that is t values, thus creating a factor t in the log terms (log(t\delta)). However, improved techniques based on a peeling argument (where the values of N_t(s,a) are localized) leads to better scaling essentially moving from a t term to a log(t) factor, up to loosing some constant factors. Such argument has been used in several analyses of bandit: see e.g. the analysis of KL-UCB by [Cappe et al.], or tools by Aurelien Garivier relating this indeed to the LIL. One can also see a similar type of argument in the PhD of Sebastien Bubeck, or in even earlier papers by Tze Leung Lai. More generally, the peeling technique is largely used is mathematical statistics, and is not novel. Now, it is an easy exercise to show that the same peeling argument leads to concentration inequalities valid not for a single t, but uniformly over all t (actually for possibly unbounded random stopping time, and thus in particular for the random number of pulls) as well; thus I do not see much novelty either. Let us go a little more in depth in the statements (e.g. G.1) regarding tightness of the bounds that is provided here, one may first question the use of a peeling with a geometric factor 2, that is arguably the most naive and sub-optimal one, and could be improved (see e.g. analysis of KL-UCB in [Cappe et al.]). Second, at least for sub-Gaussian random variables, one may instead resort to the Laplace method: this method leads to a \sqrt{t} term in the logarithm instead of a \log(t) and thus might be considered sub-optimal. However, the constant factors that we are used when applying the peeling argument, even after optimization of all terms (which is not done here), is still larger than the one we loose with the Laplace method. As a result, the bound resulting from the peeling argument term becomes only tighter than the one resulting from the Laplace method asymptotically for huge values of t (this actually makes sense as the regime of the LIL is also a quite asymptotic regime). Thus, I feel that the discussion regarding the llnp(t) term is incomplete and misses an important block of the literature, once again. E) The only possibly interesting part of the paper is Lemma D.1; but as mentioned, being able to enforce the constraints on the MDP is made easy by the fact that a finite horizon is considered. F) Minor point: the appendix seems too long for a NIPS paper. DECISION. The paper does not enough put in perspective its results with the existing literature, especially regarding the horizon H and the concentration inequalities. Likewise, the scaling of the regret bounds is not discussed against other results. These missing points tend to give the feeling that the authors oversell their contribution, and hide the more debatable aspects of their results. This is also extremely misleading for the less expert readers that may not be able to see the loose aspects of this work. Perhaps it is just that the paper was just written in a rush, but this is anyway bad practice. Points B,C,D should be improved/fixed. Reject. FINAL DECISION I read the authors' rebuttal. Their answer to point C) (scaling with H) should be included in the paper, as this is an important clarification of their result. Even though the paper is restricted to the simpler finite horizon setting, I agree that this setting is also relevant and deserves attention. I still believe that the selling point of the paper should not be section 2 and that much more credit should be given to previous works: indeed each separate idea presented in the paper existed before, so that the paper does not introduce a novel idea; what is novel however, is the combination of these. We should thus judge how this combination is executed, and I consider this is well done. Overall, given that the claims and proofs look correct (even though I haven't checked everything in detail), and that the paper is globally well executed, I accept to raise my score in order to meet a consensus.

Reviewer 3



This paper presents a novel framework for providing performance guarantees to reinforcement learning. I am somewhat familiar with some of the earlier results, but by no means an expert in this more theoretical corner of RL, so let me take some time to explain my understanding. Hopefully the authors might be able to use some of this to provide a bit more guidance for non-experts. In theoretical RL, there have been a number of frameworks to provide guarantees. Several approaches have extended the notion of PAC learning to RL, leading (amongst others) to the notion of "PAC-MDP", which seems to correspond to what the current paper calls (eps,delta)-PAC. A method is PAC-MDP if *for any epsilon and delta*, there is a polynomial function F(S,A,1/eps,1/delta) that expresses the sample complexity of exploration (=the number of mistakes N_\epsilon) with prob. 1-delta. This is nice, but many (all?) of the PAC-MDP algorithms actually take \epsilon as input: I.e., they do learning for a particular (s,a)-pair until they can guarantee (with high prob) that the accuracy is within \epsilon. Then they stop learning for this particular (s,a). This paper proposes the "\delta-uniform PAC framework". In contrast to the above, a \delta-uniform PAC algorithm only takes \delta as its input, which means that it keeps learning for all (s,a) pairs, and which means that it will converge to an optimal policy. In addition, the authors show that if an algorithm has a uniform PAC bound that directly implies a tighter regret bound than what (eps,delta)-PAC would imply. Questions: -In how far is the proposed framework limited to the episodic setting with known horizon? -If I correctly understand, the current approach still gives \delta to the algorithm. An obvious questions is whether this dependence can also be removed? -While not formalized, as in this paper, I think that many of the proposed PAC-MDP methods in fact support continue learning. E.g., Strehl&Littman'08 write "important to note that for each state-action pair (s, a), MBIE uses only the first m experiences of (s, a)" with the following footnote: "This property of the algorithm is mainly enforced for our analysis. [...] However, in experiments we have found that setting m to be very large or infinite actually improves the performance of the algorithm when computation is ignored." This gives rise to 2 questions: -in how far is the "uniform PAC framework" really novel? (don't we automatically get this when taking away the 'stop learning' switches from current PAC-MDP approaches?) -in how far are the empirical results of the baselines included in the paper really representative of their best performance? -The paper claims that the uniform PAC framework is the first to formally connect PAC and regret. I have two reservations: 1) Theorem 2a seems to directly indicate a relation between the previous PAC framework and regret? 2) Strehl&Littman'08 prove a relation to "average loss" which seems very close to regret? Details: -"Algorithms with small regret may be arbitrarily bad infinitely often." ->This is not clear to me. Seems that if they are arbitrarily bad (say, -infty) just once that would give a regret of -infty...? -display formula below 54 mixes up 't' vs 'i' -would be good to better emphasize the episodic setup, and the fact that 'T' means episode (not time step, this confused me) -line 63, FPAC has wrong arguments and bracket problem. -in the algorithm there are some strange remainders of comments...? -line 163, what does 'rng' mean exactly? -line 171 "as in c) above" - where do we find c) ? Is this Theorem 3c? (that does not seem to make sense?)